# OFF-POLICY AVERAGE REWARD ACTOR-CRITIC WITH DETERMINISTIC POLICY SEARCH

## ABSTRACT

The average reward criterion is relatively less explored as most existing works in the Reinforcement Learning literature consider the discounted reward criterion. There are few recent works that present on-policy average reward actor-critic algorithms, but average reward off-policy actor-critic is relatively less explored. In this paper, we present both on-policy and off-policy deterministic policy gradient theorems for the average reward performance criterion. Using these theorems, we also present an Average Reward Off-Policy Deep Deterministic Policy Gradient (ARO-DDPG) Algorithm. We show a finite time analysis of the resulting three-timescale stochastic approximation scheme with linear function approximator and obtain an $\epsilon$-optimal stationary policy with a sample complexity of $\Omega(\epsilon^{-2.5})$. We compare the average reward performance of our proposed algorithm and observe better empirical performance compared to state-of-the-art on-policy average reward actor-critic algorithms over MuJoCo based environments.

## 1 INTRODUCTION

The reinforcement learning (RL) paradigm has shown significant promise for finding solutions to decision making problems that rely on a reward-based feedback from the environment. Here one is mostly concerned with the long-term reward acquired by the algorithm. In the case of infinite horizon problems, the discounted reward criterion has largely been studied because of its simplicity. Major recent development in the context of RL in continuous state-action spaces has considered the discounted reward criterion (Schulman et al., 2015; 2017; Lillicrap et al., 2016; Haarnoja et al., 2018). However, there are very few works which focus on the average reward performance criterion in the continuous state-action setting (Zhang & Ross, 2021; Ma et al., 2021).

The average reward criterion has started receiving attention in recent times and there are papers that discuss the benefits of using this criterion over the discounted reward (Dewanto & Gallagher, 2021; Naik et al., 2019). One of the reasons being, average reward criteria only considers recurrent states and it happens to be the most selective optimization criterion in recurrent Markov Decision Processes (MDPs) according to n-discount optimality criterion. Please refer Mahadevan (1996) for more details on n-discount optimality criterion. Further, optimization in average reward setting is not dependent on the initial state distribution. Moreover, the discrepancy between the objective function and the evaluation metric, that exists for discounted reward setting, is resolved by opting for the average reward criterion. We encourage the readers to go through Dewanto & Gallagher (2021); Naik et al. (2019) for better understanding of the benefits mentioned.

There are very few algorithms in literature that optimize the average reward and all of them happen to be on-policy algorithms (Zhang & Ross, 2021; Ma et al., 2021). It has been demonstrated several times that on-policy algorithms are less sample efficient than off-policy algorithms Lillicrap et al. (2016); Haarnoja et al. (2018); Fujimoto et al. (2018) for the discounted reward criterion. In this paper we try to find whether the same is true for the average reward criterion. We try to overcome the research gap in development of off-policy average reward algorithms for continuous state and action spaces by proposing an Average Reward Off-Policy Deep Deterministic Policy Gradient (ARO-DDPG) Algorithm.

The policy evaluation step in the case of the average reward algorithm is equivalent to finding the solution to the Poisson equation (i.e., the Bellman equation for a given policy). Poisson equation, because of its form, does not admit a unique solution but only solutions that are unique up to a constant

term. Further, the policy evaluation step in this case consists of finding not just the Differential Q-value function but also the average reward. Thus, because of the required estimation of two quantities instead of one, the role of the optimization algorithm and the target network increases here. Therefore we implement the proposed ARO-DDPG algorithm by using target network and by carefully selecting the optimization algorithm.

The following are the broad contributions of our paper:

- We provide both on-policy and off-policy deterministic policy gradient theorems for the average reward performance metric.
- We present our Average Reward Off-Policy Deep Deterministic Policy Gradient (ARO-DDPG) algorithm.
- We perform non-asymptotic convergence analysis and provide a finite time analysis of our three timescale stochastic approximation based actor-critic algorithm using a linear function approximator.
- We show the results of implementations using our algorithm with other state-of-the-art algorithms in the literature.

The rest of the paper is structured as follows: In Section 2, we present the preliminaries on the MDP framework, the basic setting as well as the policy gradient algorithm. Section 3 presents the deterministic policy gradient theorem and our algorithm. Section 4 then presents the main theoretical results related to the finite time analysis. Section 5 presents the experimental results. In Section 6, we discuss other related work and Section 7 presents the conclusions. The detailed proofs for the finite time analysis are available in the Appendix.

## 2 PRELIMINARIES

Consider a Markov Decision Process (MDP) $M = \{S, A, R, P, \pi\}$ where $S \subset \mathbb{R}^n$ is the (continuous) state space, $A \subset \mathbb{R}^m$ is the (continuous) action space, $R : S \times A \mapsto \mathbb{R}$ denotes the reward function with $R(s, a)$ being the reward obtained under state $s$ and action $a$. Further, $P(\cdot|s, a)$ denotes the state transition function defined as $P : S \times A \mapsto \mu(\cdot)$, where $\mu : \mathcal{B}(S) \mapsto [0, 1]$ is a probability measure. Deterministic policy $\pi$ is defined as $\pi : S \mapsto A$. In the above, $\mathcal{B}(S)$ represents the Borel sigma algebra on $S$. Stochastic policy $\pi_r$ is defined as $\pi_r : S \mapsto \mu'(\cdot)$, where $\mu' : \mathcal{B}(A) \mapsto [0, 1]$ and $\mathcal{B}(A)$ is the Borel sigma algebra on A.

**Assumption 1.** *The Markov process obtained under any policy $\pi$ is ergodic.*

Assumption 1 is necessary to ensure existence of steady state distribution of Markov process.

### 2.1 DISCOUNTED REWARD MDPS

In discounted reward MDPs, discounting is controlled by $\gamma \in (0, 1)$. The following performance metric is optimized with respect to the policy:

$$\eta(\pi) = \mathbb{E}^\pi[\sum_{t=0}^{\infty} \gamma^t R(s_t, a_t)] = \int_S \rho_0(s) V^\pi(s) \, ds. \tag{1}$$

Here, $\rho_0$ is the initial state distribution and $V^\pi$ is the value function. $V_\pi(s)$ denotes the long term reward acquired when starting in the state $s$.

$$V^\pi(s_t) = \mathbb{E}^\pi[R(s_t, a_t) + \gamma V^\pi(s_{t+1})|s_t]. \tag{2}$$

### 2.2 AVERAGE REWARD MDPS

The performance metric in the case of average reward MDPs is the long-run average reward $\rho(\pi)$ defined as follows:

$$\rho(\pi) = \lim_{N \to \infty} \frac{1}{N} \mathbb{E}^\pi[\sum_{t=0}^{N-1} R(s_t, a_t)] = \int_S d^\pi(s) R^\pi(s) \, ds, \tag{3}$$

where $R^\pi(s) \triangleq R(s, \pi(s))$. The limit in the first equality in equation 3 exists because of Assumption 1. The quantity $d^\pi(s)$ in the second equality in equation 3 corresponds to the steady state probability of the Markov process being in state $s \in S$ and it exists and is unique given $\pi$ from Assumption 1 as well.

$V^\pi_{diff}$ is the differential value function corresponding to the policy $\pi$ and is defined in (4). Further, the differential Q-value or action-value function $Q^\pi_{diff}$ is defined in (5).

$$V^\pi_{diff}(s_t) = \mathbb{E}^\pi[\sum_{k=t}^{\infty} R(s_k, a_k) - \rho(\pi)|s_t]. \tag{4}$$

$$Q^\pi_{diff}(s_t, a_t) = \mathbb{E}^\pi[\sum_{k=t}^{\infty} R(s_k, a_k) - \rho(\pi)|s_t, a_t]. \tag{5}$$

**Lemma 1.** *There exists a unique constant $k(= \rho(\pi))$ which satisfies the following equation for differential value function $V_{diff}$ :*

$$V^\pi_{diff}(s_t) = \mathbb{E}^\pi[R(s_t, a_t) - k + V^\pi_{diff}(s_{t+1})|s_t] \tag{6}$$

*Proof.* See appendix for the proof. $\qquad\square$

### 2.3 POLICY GRADIENT THEOREM

Unlike in Q-learning where we try to find the optimal Q-value function and then infer the policy from it, the policy gradient theorem (Sutton et al., 1999; Silver et al., 2014; Degris et al., 2012) allows us to directly optimize the performance metric via its gradient with respect to the policy parameters. Q-learning can be visualized to be a value iteration scheme while an algorithm based on the policy gradient theorem can be seen as mimicking policy iteration. Sutton et al. (1999) provided the policy gradient theorem for on-policy optimization of both the discounted reward and the average reward algorithms, see (7)-(8), respectively.

$$\nabla_\theta \eta(\pi) = \int_S \omega^\pi(s) \int_A \nabla_\theta \pi_r(a|s, \theta) Q^\pi(s, a) \, da \, ds. \tag{7}$$

$$\nabla_\theta \rho(\pi) = \int_S d^\pi(s) \int_A \nabla_\theta \pi_r(a|s, \theta) Q^\pi_{diff}(s, a) \, da \, ds. \tag{8}$$

In (7) $\omega^\pi$ denotes the long term discounted state visitation probability density which is defined in equation 9 while $d^\pi(s) = \lim_{t\to\infty} P^\pi_t(s)$ is the steady state probability density on states. $P^\pi$ denotes the transition probability kernel for the Markov chain induced by policy $\pi$ and $P^\pi_t$ is the state distribution at instant $t$ given by (10).

$$\omega^\pi(s) = (1 - \gamma) \sum_{t=0}^{\infty} \gamma^t P^\pi_t(s). \tag{9}$$

$$P^\pi_t(s) = \int_{S \times S...} \rho_0(s_0) \prod_{k=0}^{t-1} P^\pi(s_{k+1}|s_k) \, ds_0 \ldots ds_{t-1}. \tag{10}$$

The policy gradient theorem in Sutton et al. (1999) is only valid for on-policy algorithms. Degris et al. (2012) proposed an approximate off-policy policy gradient theorem for stochastic policies, see (11), where $d^\mu$ stands for the steady state density function corresponding to the policy $\mu$.

$$\nabla_\theta \eta(\pi) \approx \int_S d^\mu(s) \int_A \nabla_\theta \pi_r(a|s, \theta) Q^\pi(s, a) \, da \, ds. \tag{11}$$

Silver et al. (2014) came up with the deterministic policy gradient theorem, see (12), which eventually led to the development of very successful Deep Deterministic Policy Gradient (DDPG) (Lillicrap et al., 2016) algorithm and Twin Delayed DDPG (TD3) algorithm (Fujimoto et al., 2018).

$$\nabla_\theta \eta(\pi) = \int_S \omega^\pi(s) \nabla_a Q^\pi(s, a)|_{a=\pi(s)} \nabla_\theta \pi(s, \theta) \, ds. \tag{12}$$

## 3 PROPOSED AVERAGE REWARD ALGORITHM

We now propose the deterministic policy gradient theorem for the average reward criterion. The policy gradient estimator has to be derived separately for both the on-policy and off-policy settings. Obtaining the on-policy deterministic policy gradient estimator is straight forward but dealing with the off-policy gradient estimates involves an approximate gradient (Degris et al., 2012).

### 3.1 ON-POLICY POLICY GRADIENT THEOREM

We cannot directly use the second equality of (3) to derive the policy gradient theorem because of the inability to take the derivative of steady state density function. Therefore one needs to use (6) to obtain the average reward deterministic policy gradient theorem.

**Theorem 1.** *The gradient of $\rho(\pi)$ with respect to policy parameter $\theta$ is given as follows:*

$$\nabla_\theta \rho(\pi) = \int_S d^\pi(s) \nabla_a Q^\pi_{diff}(s,a)|_{a=\pi(s)} \nabla_\theta \pi(s,\theta) \, ds. \tag{13}$$

*Proof.* See appendix for the proof. □

### 3.2 COMPATIBLE FUNCTION APPROXIMATION

The result in this section is mostly inspired from Silver et al. (2014). Recall that $Q^\pi_{diff}(s,a)$ is the 'true' differential $Q$-value of the state-action tuple $(s,a)$ under the parameterized policy $\pi$. Now let $Q^w_{diff}(s,a)$ denote the approximate differential $Q$-value of the $(s,a)$-tuple when function approximation with parameter $w$ is used. Lemma 2 says that when the function approximator satisfies a compatibility condition (cf. (14,15)), then the gradient expression in (13,) is also satisfied by $Q^w_{diff}$ in place of $Q^\pi_{diff}$.

**Lemma 2.** *Assume that the differential Q-value function (5) satisfies the following:*

$$1. \nabla_w \nabla_a Q^w_{diff}(s,a) = \nabla_\theta \pi(s,\theta). \tag{14}$$

*2. Differential Q-value function parameter $w = w^*_\epsilon$ optimizes the following error function:*

$$\zeta(\theta, w) = \frac{1}{2} \int_S d^\pi(s) \|\nabla_a Q^\pi_{diff}(s,a)|_{a=\pi(s)} - \nabla_a Q^w_{diff}(s,a)|_{a=\pi(s)}\|^2 \, ds. \tag{15}$$

*Then,*

$$\int_S d^\pi(s) \nabla_a Q^\pi_{diff}(s,a)|_{a=\pi(s)} \nabla_\theta \pi(s,\theta) \, ds = \int_S d^\pi(s) \nabla_a Q^w_{diff}(s,a)|_{a=\pi(s)} \nabla_\theta \pi(s,\theta) \, ds. \tag{16}$$

*Further, in the case when a linear function approximator is used, we obtain*

$$\nabla_a Q^w_{diff}(s,a) = \nabla_\theta \pi(s,\theta)^\intercal w. \tag{17}$$

*Proof.* See the appendix for a proof. □

An important implication of lemma 2 also is that the dimension of the matrix on the left hand side and the right hand side of (14) should be the same. Hence the dimensions of the parameters $\theta$ (used in the parameterized policy) and $w$ (used to approximate the differential Q-value function) are the same. Lemma 2 shows that the compatible function approximation theorem has the same form in the average reward setting as the discounted reward setting.

### 3.3 OFF-POLICY POLICY GRADIENT THEOREM

In order to derive off-policy policy gradient theorem it is not possible to use the direction adopted by Degris et al. (2012) for off-policy stochastic policy gradient theorem for the discounted reward setting. We first mention our proposed approximate off-policy deterministic policy gradient theorem and then explain why some alternatives would not have worked.

**Assumption 2.** *For the Markov chain obtained from the policy $\pi$, let $K(\cdot|\cdot)$ be the transition kernel and $S^\pi$ the steady state measure. Then there exists $a > 0$ and $\kappa \in (0, 1)$ such that*

$$D_{TV}(K^t(\cdot|s), S^\pi(\cdot)) \leq a\kappa^t, \forall t, \forall s \in S.$$

Assumption 2 states that Markov chain generated by a policy $\pi$ follows uniform ergodicity property. This assumption is necessary to get an upper bound on the total variation norm of steady state probability distribution of two policies. This assumption is used in Lemma 12, which in turn is used for Theorem 2.

**Theorem 2.** *The approximate gradient of the average reward $\rho(\pi)$ with respect to the policy parameter $\theta$ is given by the following expression:*

$$\widehat{\nabla_\theta \rho}(\pi) = \int_S d^\mu(s) \nabla_a Q_{diff}^\pi(s, a)|_{a=\pi(s)} \nabla_\theta \pi(s, \theta) \, ds. \tag{18}$$

*Further, the approximation error is $\mathcal{E}(\pi, \mu) = \|\nabla_\theta \rho(\pi) - \widehat{\nabla_\theta \rho}(\pi)\|$, where $\mu$ represents the behaviour policy. $\mathcal{E}$ satisfies*

$$\mathcal{E}(\pi, \mu) \leq Z\|\theta^\pi - \theta^\mu\|. \tag{19}$$

*Here, $Z = 2^{m+1}C(\lceil \log_\kappa a^{-1} \rceil + 1/\kappa)L_t$ with $L_t$ being the Lipchitz constant for the transition probability density function (Assumption 9). Constants $a$ and $\kappa$ are from Assumption 2, $m$ is the dimension of action space, and $C = \max_s \|\nabla_a Q_{diff}^\pi(s, a)|_{a=\pi(s)} \nabla_\theta \pi(s, \theta)\|$.*

*Proof.* See the appendix for a proof. $\qquad\square$

Theorem 2 suggests that the approximation error in the gradient increases as the difference between the target policy $\pi$ and the behaviour policy $\mu$ increases.

### 3.4 OFF-POLICY ALTERNATIVES

In this section we will talk about what alternatives could be thought of in place of what is suggested in section 3.3 and why those alternatives would not work.

1. One can possibly take inspiration from Degris et al. (2012) and define an objective function, $\bar{\rho}(\pi)$, as in (20), which is a naive off-policy version of (3).

$$\rho_{new}(\pi) = \int_S d^\mu(s) R^\pi(s) \, ds. \tag{20}$$

   If, however, we take the derivative of $\rho_{new}(\pi)$ defined above, we get the policy update rule as in (21).

$$\nabla_\theta \rho_{new}(\pi) = \int_S d^\mu(s) \nabla_a R(s, a)|_{a=\pi(s)} \nabla_\theta \pi(s, \theta) \, ds. \tag{21}$$

   The update rule (21) only considers the reward function and not the transition dynamics of the MDP. In (18), the derivative of the objective function includes the differential Q-value function which encapsulates both the information of the reward function and the transition dynamics of the MDP and hence is valid derivative.

2. A lot of work in the off-policy setting relies on importance sampling ratios. Recently a few works devised a method to estimate the steady state probability density ratio of the target and behavior policies (Zhang et al., 2020a;b; Liu et al., 2018; Nachum et al., 2019). The ratio of steady state densities could be used for deterministic policy optimization but there are certain issues which prohibit its usage, see (22).

$$\nabla_\theta \rho(\pi) = \int_S d^\mu(s) \tau(s) \nabla_a Q_{diff}^\pi(s, a)|_{a=\pi(s)} \nabla_\theta \pi(s, \theta) \, ds. \tag{22}$$

   Here, $\tau(s)$ is the steady state probability density ratio defined as $d^\pi(s)/d^\mu(s)$. In order to calculate $\tau(s)$ we need information about $(\pi(a|s), \mu(a|s)$ and $P(s'|s, a))$. We need the ratio

$\pi(a|s)/\mu(a|s)$ and for deterministic policies the ratio would be $\delta(a - \pi(s))/\delta(a - \mu(s))$, where $\delta(\cdot)$ is the Dirac-Delta function:

$$\frac{\delta(a - \pi(s))}{\delta(a - \mu(s))} = \begin{cases} 0 & \text{if } a = \mu(s) \\ \infty & \text{if } a = \pi(s) \\ \frac{0}{0} & \text{otherwise.} \end{cases} \tag{23}$$

From (23), it is clear that the ratio $\delta(a - \pi(s))/\delta(a - \mu(s))$ will be undefined for almost all actions $a \in A$. Thus, we cannot use this ratio for deterministic policies. Otherwise, we need $P(s'|s, \pi(a))$ and $P(s'|s, \mu(a))$. It is possible to get the information about $P(s'|s, \mu(a))$ by sampling from the Markov process generated by the policy $\mu$ but obtaining this information about $P(s'|s, \pi(a))$ is impossible as in the off-policy setting data from $\pi$ is assumed to be simply unavailable.

## 3.5 ACTOR-CRITIC UPDATE RULE

**Assumption 3.** $\alpha_t, \beta_t,$ and $\gamma_t$ are the step sizes for critic, target estimator, and actor parameter updates respectively.

$$\alpha_t = \frac{C_\alpha}{(1+t)^\sigma} \quad \beta_t = \frac{C_\beta}{(1+t)^u} \quad \gamma_t = \frac{C_\gamma}{(1+t)^v}$$

*Here, $C_\alpha, C_\beta, C_\gamma > 0$ and $0 < \sigma < u < v < 1$. $\alpha_t$ is at the fastest timescale, $\beta_t$ is at slower timescale and $\gamma_t$ is at the slowest timescale.*

The critic and average reward parameters are estimated using the TD(0) update rule but use target estimators. We are using target estimators to ensure stability of the iterates of the algorithm. Let $\{s_i, a_i, s'_i\}_{i=0}^{n-1}$ denote the batch of sampled data from the replay buffer.

$$\xi_t^j = \frac{1}{2} \sum_{i=0}^{n-1} \left( R(s_i, a_i) - \overline{\rho_t} - Q_{diff}^{w_i}(s_i, a_i) + \min(Q_{diff}^{\overline{w_1}}, Q_{diff}^{\overline{w_2}})(s'_i, \pi(s'_i, \overline{\theta_t})) \right)^2 j \in \{1, 2\} \tag{24}$$

$$\xi_t^3 = \frac{1}{2} \sum_{i=0}^{n-1} \left( R(s_i, a_i) - \rho_t - \min(Q_{diff}^{\overline{w_1}}, Q_{diff}^{\overline{w_2}})(s_i, a_i) \right) + \min(Q_{diff}^{\overline{w_1}}, Q_{diff}^{\overline{w_2}})(s'_i, \pi(s'_i, \overline{\theta_t})) \right)^2 \tag{25}$$

Equation 24 and 25 are the bellman error for differential Q-value function approximator and average reward estimator respectively. Note here we are using double Q-value function approximator.

$$w_{t+1}^i = w_t^i - \alpha_t \nabla_{w_i} \xi_t^i \quad i \in \{1, 2\} \tag{26}$$

$$\rho_{t+1} = \rho_t - \alpha_t \nabla_p \xi_t^3 \tag{27}$$

The bellman errors 24 is used to update Q-value function approximator parameters $w_t^i$ using 26 and the bellman average in 25 is used to update average reward estimator $\rho_t$ using 27.

$$\nu_i = \nabla_a min(Q_{diff}^{w_1}, Q_{diff}^{w_2})(s_i, a)|_{a=\pi(s_i)} \nabla_\theta \pi(s_i, \theta_t) \tag{28}$$

$$\theta_{t+1} = \theta_t + \gamma_t \left( \sum_{i=0}^{n-1} \nu_i \right) \tag{29}$$

Actor update is performed using theorem 2. Actor parameter, $\theta_t$, is updated using empirical estimate (28) of the gradient in 18.

$$\overline{w_{t+1}^i} = \overline{w_t^i} + \beta_t(w_{t+1}^i - \overline{w_t^i}) \quad i \in \{1, 2\} \tag{30}$$

$$\overline{\rho_{t+1}} = \overline{\rho_t} + \beta_t(\rho_{t+1} - \overline{\rho_t}) \tag{31}$$

$$\overline{\theta_{t+1}} = \overline{\theta_t} + \beta_t(\theta_{t+1} - \overline{\theta_t}) \tag{32}$$

Equation 30-32 are used to update the target Q-value function approximator $\overline{w_t^i}$, target average reward estimator $\overline{\rho_t}$ and target actor parameter $\overline{\theta_t}$.

## 4 FINITE TIME ANALYSIS

In this section we present the finite time analysis of the on-policy and off-policy average reward actor critic algorithm with linear function approximators. First we mention the assumptions taken to perform the finite time analysis followed by the main results.

**Assumption 4.** $\phi^\pi(s)\big(=\phi(s,\pi(s))$ denotes the feature vector of state s and satisfies $\|\phi^\pi(s)\| \leq 1$.

The assumption above is just taken for the sake of convenience.

**Assumption 5.** *The reward function is uniformly bounded, viz.,* $|R^\pi(s)| \leq C_r < \infty$.

Assumption 5 is required to make sure that the average reward objective function is bounded from above.

**Assumption 6.** $Q^w_{diff}(s,a)$ *is Lipchitz continuous w.r.t to a. Thus,* $\forall w \quad \|Q^w_{diff}(s,a_1) - Q^w_{diff}(s,a_2)\| \leq L_a\|a_1 - a_2\|$.

Continuity of approximate Q-value function w.r.t action is enforced using Assumption 6. Without the continuity property approximate differential Q-values will not generalize for unseen action values.

**Assumption 7.** *Parameterised policy* $\pi(s,\theta)$ *is Lipchitz continuous w.r.t* $\theta$. *Thus,* $\|\pi(s,\theta_1) - \pi(s,\theta_2)\| \leq L_\pi\|\theta_1 - \theta_2\|$.

Assumption 7 is a common regularity assumption for convergence of actor. It can be found in Wu et al. (2020), Xiong et al. (2022) and Zou et al. (2019).

**Assumption 8.** *The state feature mapping* $(\phi^\pi(s) = \phi(s,\pi(s)))$ *defined for a policy* $\pi$ *with parameter* $\theta$ *is Lipschitz continuous w.r.t* $\theta$. *Thus,* $\max_s \|\phi^{\pi_1}(s) - \phi^{\pi_2}(s)\| \leq L_\phi\|\theta_1 - \theta_2\|$.

Continuity of state action feature w.r.t action is required to ensure generalisation of Q-values to unseen action values. Using this continuity of state action feature with Assumption 7 we can satisfy Assumption 8.

### 4.1 ON-POLICY ANALYSIS

In this section we present the theorem for finite time analysis of the on-policy version of the algorithm with linear function approximator and target estimator for the critic and average reward.

**Theorem 3.** *The on-policy average reward actor critic algorithm (Algorithm 2) obtains an* $\epsilon$-*accurate optimal point with sample complexity of* $\Omega(\epsilon^{-2.5})$. *We obtain*

$$\min_{0 \leq t \leq T-1} E\|\nabla_\theta \rho(\theta_t)\|^2 = \mathcal{O}\left(\frac{1}{T^{0.4}}\right) + \mathcal{O}(1),$$
$$\leq \epsilon + \mathcal{O}(1).$$

*Proof.* See the appendix for a proof. ∎

We want to reach as close as possible to a value of $\theta$ such that $\|\nabla_\theta \rho(\theta)\| = 0$, which indicates we have found a local maxima. $\mathcal{O}(1)$ term is present in the bound because of using linear function approximation and will not reduce as time increases. However, if the $\mathcal{O}(1)$ term is small enough, the bound in Theorem 3 shows that as T is increases, the algorithm will get close to the local maxima of the objective function(3). A similar $\mathcal{O}(1)$ term is present in (Xiong et al., 2022). Xiong et al. claims the term will be small upon using neural network for critic.

### 4.2 OFF-POLICY ANALYSIS

In this section we present the theorem for finite time analysis of off-policy version of the algorithm with linear function approximator and target estimator for the critic and average reward.

**Theorem 4.** *The off-policy average reward actor critic algorithm (Algorithm 3) with behavior policy* $\mu$ *obtains an* $\epsilon$-*accurate optimal point with sample complexity of* $\Omega(\epsilon^{-2.5})$. *Here* $\theta_\mu$ *refers to the behavior policy parameter and* $\theta_t$ *refers to the target or current policy parameter. We obtain*

$$\min_{0 \le t \le T-1} E\|\widehat{\nabla_\theta \rho}(\theta_t)\|^2 = \mathcal{O}\left(\frac{1}{T^{0.4}}\right) + \mathcal{O}(1) + \mathcal{O}(W_\theta^2)$$

$$\le \epsilon + \mathcal{O}(1) + \mathcal{O}(W_\theta^2)$$

$$where\ W_\theta := \max_t \|\theta_\mu - \theta_t\|.$$

*Proof.* See the appendix for a proof. ☐

The significance of finding a bound on $\|\widehat{\nabla_\theta \rho}(\theta_t)\|$ is same as explained above for Theorem 3. The error bound in the off-policy algorithm has an extra term $\mathcal{O}(W_\theta^2)$. The extra term denotes the error induced because of not using the samples from the current policy for performing updates. $W_\theta^2$ will be small when replay buffer is used because replay buffer contains data from policies similar to the current policy. This explains why Theorem 2 can be used with replay buffer.

## 5 EXPERIMENTAL RESULTS

We conducted experiments on six different environments using the DeepMind control suite (Tassa et al., 2018) and found the performance of ARO-DDPG to be superior than the other algorithms. All the environments selected are infinite horizon tasks. Maximum reward per time step is 1.None of the tasks have a goal reaching nature. We performed all the experiments using 10 different seeds. We show here performance comparisons with two state-of-the-art algorithms: the Average Reward TRPO (ATRPO) (Zhang & Ross, 2021) and the Average Policy Optimization (APO) (Ma et al., 2021) respectively. In general for the average reward performance, not many algorithms are available in the literature. We implemented the ATRPO algorithm using the instructions available in the original paper. We used the original hyper-parameters suggested by the author for ATRPO.

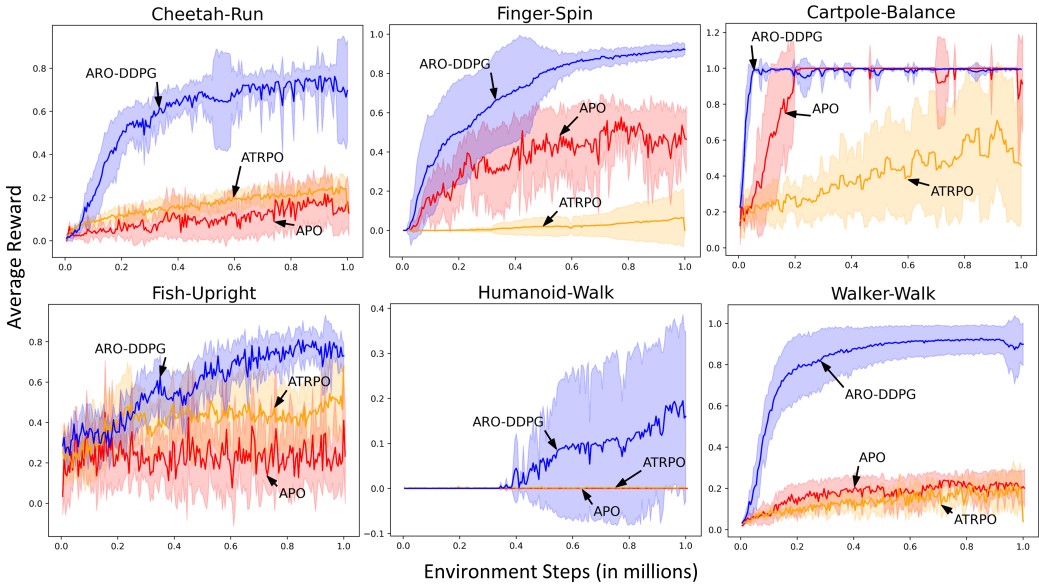

Figure 1: Comparison of performance of different average reward algorithms

For our proposed algorithm we trained the agent for 1 million time steps and evaluated the agent after every 5,000 time steps in the concerned environment. The length of each episode for the training phase was taken to be 1,000 and for the evaluation phase it was taken to be 10,000. The reason for taking longer episode length for evaluation phase was to compare the long term average reward performance of the algorithms. We also tried using episode length of 10,000 for training phase and found that to be giving poor average reward performance. We do not reset the agent if it lands in a

state before completing 10,000 steps from where it is unable to escape of its own, while continuing to give a penalty for the remaining length of the episode. That way the cost of failure is very high. While training we updated the actor after performing a fixed number of environment steps. We updated the critic neural network with more frequency as compared to the actor neural network. We used target actor and critic networks along with target estimator of the average reward parameter for stability while using bootstrapping updates. We updated the target network using polyak averaging. We tried to enforce multiple timescales in our algorithm by using different update frequency for actor, critic and polyak averaging for target networks. We also borrowed the double Q-network trick from Fujimoto et al. (2018). Complete information regarding the set of hyper-parameters used is provided in the appendix.

# 6 RELATED WORK

Actor-Critic algorithms for average reward performance criterion is much less studied compared to discounted reward performance criterion. One of the earliest works on the average reward criterion is Mahadevan (1996). In this paper, Mahadevan compares the performance of R-learning with that of Q-learning and concludes that fine tuning is required to get better results from R-learning. R-learning is the average reward version of Q-learning. Later in 1999, Sutton et al. derived the policy gradient theorem for both discounted and average reward criteria (Sutton et al., 1999), which formed the bedrock for development of the average reward actor-critic algorithms. The first proof of asymptotic convergence of average reward actor-critic algorithms with function approximation appeared in Konda & Tsitsiklis (2003). In 2007, Bhatnagar et al. proposed incremental natural policy gradient algorithms for the average reward setting and provided the asymptotic convergence proof of these.

Recently, Wan et al. presented a Differential Q-learning algorithm and claimed that their algorithm is able to find the exact differential value function without an offset. Further, Wan et al. provided an extension of the options framework from the discounted setting to the average reward setting and demonstrated the performance of the algorithm in the Four-Room domain task. One of the major contributions in off-policy policy evaluation is made by Zhang et al. (2021a). Here Zhang et al. gave a convergent off-policy evaluation scheme inspired from the gradient temporal difference learning algorithms but involving a primal-dual formulation making the policy evaluation step feasible for a neural network implementation. Zhang et al. (2021b) provided another convergent off-policy evaluation algorithm using target network and $l_2$-regularisation. In our work we use the same policy evaluation update.

Our work in this paper is actually an extension of the work of Silver et al. (2014) from the discounted to the average reward setting. In Xiong et al. (2022), a finite time analysis for deterministic policy gradient algorithm was done for the discounted reward setting. We performed the finite time analysis for the average reward deterministic policy gradient algorithm and in particular obtain the same sample complexity for our algorithm as reported by Wu et al. (2020) for stochastic policies.

# 7 CONCLUSION AND FUTURE WORK

In this paper we presented a deterministic policy gradient theorem for both on-policy and off-policy settings. We then proposed the Average Reward Off-policy Deep Deterministic Policy Gradient(ARO-DDPG) algorithm using neural network and replay buffer for high dimensional MuJoCo based environments. We observed superior performance of ARO-DDPG over existing average reward algorithms (ATRPO and PPO). At the end we provided a finite time analysis for the on-policy and off-policy algorithms obtained from the proposed policy gradient theorem and obtained a sample complexity of $\Omega(\epsilon^{-2.5})$. Lastly to extend the current line of work, one could try using natural gradient descent based update rule for deterministic policy. Further in the current work we tried optimizing the average reward performance (gain optimality). In the literature, optimizing the differential value function for all the states is mentioned as part of achieving Blackwell optimality. Hence actor-critic algorithms could be designed that not only optimize average reward performance but also differential value function (bias optimality).

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

## A APPENDIX

### A.1 ADDITIONAL ASSUMPTIONS, PROOFS OF LEMMAS AND THEOREMS

We make the following additional assumptions.

**Assumption 9.** *The transition probability density function for a policy $\pi$ with parameter $\theta$ is Lipschitz continuous w.r.t $\theta$. Thus, $max_{s',s}|P^{\pi_1}(s'|s) - P^{\pi_2}(s'|s)| \leq L_t\|\theta_1 - \theta_2\|$.*

The above assumption is a standard assumption in theoretical studies in literature. Reference for those assumptions can be found in Xiong et al. (2022); Bertsekas (1975); Chow & Tsitsiklis (1991) and Dufour & Prieto-Rumeau (2015).

**Assumption 10.** *The reward function for a policy $\pi$ with parameter $\theta$ is Lipschitz continuous w.r.t $\theta$. Thus, $max_s|R^{\pi_1}(s) - R^{\pi_2}(s)| \leq L_r\|\theta_1 - \theta_2\|$.*

The above assumption can be satified by using a well defined reward function to ensure Lipchitz continuity of reward function w.r.t action and then evoking Assumption 7.

**Assumption 11.** *The initial value of target estimators is bounded. Thus, $\|\bar{w}_0\| \leq C_w$ and $\|\bar{\rho}_0\| \leq (Cr + 2C_w)$.*

Assumption 11 is used to enforce the stability of the iterates of target estimators.

**Assumption 12.** *Let* $A(\theta) = \int d^\pi(s)(\phi^\pi(s)(\int P^\pi(s'|s)\phi^\pi(s')\,ds' - \phi^\pi(s))^\intercal - \eta I)\,ds.$ $\lambda_{min}$ *is the lower bound on the minimum eigenvalue of* $A(\theta)$ *for all values of* $\theta$.

The assumption above is used in Lemma 6 to prove the Lipchitz continuity of optimal critic parameter $w^*$ for a particular value of policy parameter $\theta$ with respect to $\theta$.

**Assumption 13.** *Let* $A'(\theta) = \int d^\pi(s)(\phi^\pi(s)(\int P^\pi(s'|s)\phi^\pi(s')\,ds' - \phi^\pi(s))^\intercal)\,ds.$ $\lambda_{max}^{all}$ *is the upper bound on maximum eigenvalue of* $(A'(\theta) + A'(\theta)^\intercal)/2$ *for all values of* $\theta$.

Assumption 13 is used to prove the negative definiteness of the matrix $A_\theta$ (defined in Assumption 12) in Lemma 11.

**Assumption 14.** *Let* $H_\theta = \int_S d^\pi(s)\nabla_\theta\pi(s,\theta)\nabla_\theta\pi(s,\theta)^\intercal\,ds.$ $\lambda_{min}^\epsilon > 0$ *is the lower bound on the minimum eigenvalues of* $H_\theta$ *for all values of* $\theta$.

The above assumption is used in Lemma 13 to make sure $H_\theta$ is invertible and optimal critic parameter $w_\epsilon^*$ according to compatible function approximation lemma (Lemma 2) can be obtained. Similar assumption is present in (Xiong et al., 2022).

**Assumption 15.** *Let* $A_{off}^\mu{}'(\theta) = \int d^\mu(s)(\phi^\pi(s)(\int P^\pi(s'|s)\phi^\pi(s')\,ds' - \phi^\pi(s))^\intercal)\,ds.$ $\chi_{max}^{all}$ *is the upper bound on maximum eigenvalue of* $(A_{off}^\mu{}'(\theta) + A_{off}^\mu{}'(\theta)^\intercal)/2$ *for behaviour policy* $\mu$ *and all values of* $\theta$.

Assumption 15 is used to prove the negative definiteness of the matrix $A_\theta$ (defined in Lemma 15) in Lemma 16.

**Lemma 1.** *There exists a unique constant* $k(= \rho(\pi))$ *which satisfies the following equation for differential value function* $V_{diff}$ :

$$V_{diff}^\pi(s_t) = \mathbb{E}^\pi[R(s_t, a_t) - k + V_{diff}^\pi(s_{t+1})|s_t].$$

*Proof.*

$$V_{diff}^\pi(s_t) = R(s_t, \pi(s_t)) - k + \int_S P^\pi(s_{t+1}|s_t)V_{diff}^\pi(s_{t+1})\,ds_{t+1}$$

$$\implies V_{diff}^\pi(s_t) - \int_S P^\pi(s_{t+1}|s_t)V_{diff}^\pi(s_{t+1})\,ds_{t+1} = R(s_t, \pi(s_t)) - k$$

$$\implies \sum_{t=0}^{T-1}\left(V_{diff}^\pi(s_t) - \int_S P^\pi(s_{t+1}|s_t)V_{diff}^\pi(s_{t+1})\,ds_{t+1}\right) = \sum_{t=0}^{T-1} R(s_t, \pi(s_t)) - kT$$

Integrating w.r.t the stationary distribution $d^\pi$ of policy $\pi$ :

$$\sum_{t=0}^{T-1}\int_S d^\pi(s_t)\left(V_{diff}^\pi(s_t) - \int_S P^\pi(s_{t+1}|s_t)V_{diff}^\pi(s_{t+1})\,ds_{t+1}\right)ds_t$$

$$= \sum_{t=0}^{T-1}\int_S d^\pi(s_t)R(s_t, \pi(s_t))\,ds - kT$$

$$\sum_{t=0}^{T-1}\left(\int_S d^\pi(s_t)V_{diff}^\pi(s_t)\,ds_t - \int_S d^\pi(s_{t+1})V_{diff}^\pi(s_{t+1})\,ds_{t+1}\right)$$

$$= \sum_{t=0}^{T-1}\int_S d^\pi(s_t)R(s_t, \pi(s_t))\,ds_t - kT$$

Note: $\left(\int_S d^\pi(s_t)V_{diff}^\pi(s_t)\,ds_t - \int_S d^\pi(s_{t+1})V_{diff}^\pi(s_{t+1})\,ds_{t+1}\right) = 0.$

$$\implies k = \frac{1}{T} \sum_{t=0}^{T-1} \int_S d^\pi(s_t) R(s_t, \pi(s_t)) \, ds_t$$

$$\implies k = \lim_{T \to \infty} \frac{1}{T} \sum_{t=0}^{T-1} \int_S d^\pi(s_t) R(s_t, \pi(s_t)) \, ds_t$$

$$\implies k = \rho(\pi) \quad \text{(using (3))}.$$

$\square$

**Theorem 1.** *The gradient of $\rho(\pi)$ with respect to the policy parameter $\theta$ is given as follows:*

$$\nabla_\theta \rho(\pi) = \int_S d^\pi(s) \nabla_a Q_{diff}^\pi(s, a)|_{a=\pi(s)} \nabla_\theta \pi(s, \theta) \, ds.$$

*Proof.* Using Lemma 1:

$$V_{diff}^\pi(s_t) = R(s_t, \pi(s_t)) - \rho(\pi) + \int_S P^\pi(s_{t+1}|s_t) V_{diff}^\pi(s_{t+1}) \, ds_{t+1}$$

$$\implies Q_{diff}^\pi(s_t, \pi(s_t)) = R(s_t, \pi(s_t)) - \rho(\pi) + \int_S P^\pi(s_{t+1}|s_t) Q_{diff}^\pi(s_{t+1}, \pi(s_{t+1})) \, ds_{t+1}$$

Differentiating w.r.t $\theta$, we obtain

$$\begin{aligned}
\nabla_\theta Q_{diff}^\pi(s_t, \pi(s_t)) &= \nabla_\theta R(s_t, \pi(s_t)) - \nabla_\theta \rho(\pi) \\
&\quad + \nabla_\theta \left( \int_S P^\pi(s_{t+1}|s_t) Q_{diff}^\pi(s_{t+1}, \pi(s_{t+1})) \, ds_{t+1} \right) \\
&= \nabla_a R(s_t, a)|_{a=\pi(s_t)} \nabla_\theta \pi(s_t) - \nabla_\theta \rho(\pi) \\
&\quad + \int_S \nabla_a P^\pi(s_{t+1}|s_t, a)|_{a=\pi(s_t)} \nabla_\theta \pi(s_t) Q_{diff}^\pi(s_{t+1}, \pi(s_{t+1})) \, ds_{t+1} \\
&\quad + \int_S P^\pi(s_{t+1}|s_t) \nabla_\theta Q_{diff}^\pi(s_{t+1}, \pi(s_{t+1})) \, ds_{t+1}.
\end{aligned}$$

Note: $\nabla_a \rho(\pi) = \nabla_a \left( \int_S d^\pi(s) R^\pi(s) \, ds \right) = 0$.

$$\begin{aligned}
\implies \nabla_\theta Q_{diff}^\pi(s_t, \pi(s_t)) &= \nabla_a Q_{diff}^\pi(s_t, a)|_{a=\pi(s_t)} \nabla_\theta \pi(s_t) - \nabla_\theta \rho(\pi) \\
&\quad + \int_S P^\pi(s_{t+1}|s_t) \nabla_\theta Q_{diff}^\pi(s_{t+1}, \pi(s_{t+1})) \, ds_{t+1}.
\end{aligned}$$

Integrating w.r.t stationary distribution $d^\pi(\cdot)$ of policy $\pi$:

$$\begin{aligned}
\int_S d^\pi(s_t) \nabla_\theta Q_{diff}^\pi(s_t, \pi(s_t)) ds_t &= \int_S d^\pi(s_t) \nabla_a Q_{diff}^\pi(s_t, a)|_{a=\pi(s_t)} \nabla_\theta \pi(s_t) ds_t - \nabla_\theta \rho(\pi) \\
&\quad + \int_S d^\pi(s_t) \int_S P^\pi(s_{t+1}|s_t) \nabla_\theta Q_{diff}^\pi(s_{t+1}, \pi(s_{t+1})) \, ds_{t+1} \, ds_t.
\end{aligned}$$

Note: $\int_S d^\pi(s) P^\pi(s'|s) \, ds = d^\pi(s')$. Thus,

$$\begin{aligned}
\nabla_\theta \rho(\pi) &= \int_S d^\pi(s_t) \nabla_a Q_{diff}^\pi(s_t, a)|_{a=\pi(s_t)} \nabla_\theta \pi(s_t) ds_t \\
&\quad + \int_S d^\pi(s_{t+1}) \nabla_\theta Q_{diff}^\pi(s_{t+1}, \pi(s_{t+1})) \, ds_{t+1} \\
&\quad - \int_S d^\pi(s_t) \nabla_\theta Q_{diff}^\pi(s_t, \pi(s_t)) ds_t.
\end{aligned}$$

$$\nabla_\theta \rho(\pi) = \int_S d^\pi(s) \nabla_a Q_{diff}^\pi(s,a)|_{a=\pi(s)} \nabla_\theta \pi(s) \, ds.$$

$\square$

**Lemma 2.** *Assume that the differential Q-value function (5) satisfies the following:*

    *1.*

$$\nabla_w \nabla_a Q_{diff}^w(s,a) = \nabla_\theta \pi(s,\theta).$$

    *2. The differential Q-value function parameter $w = w_\epsilon^*$ optimizes the following error function:*

$$\zeta(\theta,w) = \frac{1}{2} \int_S d^\pi(s) \|\nabla_a Q_{diff}^\pi(s,a)|_{a=\pi(s)} - \nabla_a Q_{diff}^w(s,a)|_{a=\pi(s)}\|^2 \, ds.$$

*Then,*

$$\int_S d^\pi(s) \nabla_a Q_{diff}^\pi(s,a)|_{a=\pi(s)} \nabla_\theta \pi(s,\theta) \, ds = \int_S d^\pi(s) \nabla_a Q_{diff}^w(s,a)|_{a=\pi(s)} \nabla_\theta \pi(s,\theta) \, ds.$$

*Further,*

$$\nabla_a Q_{diff}^w(s,a) = \nabla_\theta \pi(s,\theta)^\intercal w \quad \text{(for linear function approximator).}$$

*Proof.* Let $\mathcal{E}(\theta,w,s) = \nabla_a Q_{diff}^\pi(s,a)|_{a=\pi(s)} - \nabla_a Q_{diff}^w(s,a)|_{a=\pi(s)}$,

$$\zeta(\theta,w) = \frac{1}{2} \int_S d^\pi(s) \mathcal{E}(\theta,w,s)^\intercal \mathcal{E}(\theta,w,s) \, ds.$$

Differentiating w.r.t the critic parameter $w$, we obtain:

$$\begin{aligned}
\nabla_w \zeta(\theta,w) &= \int_S d^\pi(s) \nabla_w \mathcal{E}(\theta,w,s) \mathcal{E}(\theta,w,s) \, ds \\
&= -\int_S d^\pi(s) \nabla_w \nabla_a Q_{diff}^w(s,a)|_{a=\pi(s)} \Big( \nabla_a Q_{diff}^\pi(s,a)|_{a=\pi(s)} \\
&\quad - \nabla_a Q_{diff}^w(s,a)|_{a=\pi(s)} \Big) \, ds = 0.
\end{aligned}$$

Letting $\nabla_w \nabla_a Q_{diff}^w(s,a)|_{a=\pi(s)} = \nabla_\theta \pi(s)$, we obtain

$$\int_S d^\pi(s) \nabla_a Q_{diff}^\pi(s,a)|_{a=\pi(s)} \nabla_\theta \pi(s,\theta) \, ds = \int_S d^\pi(s) \nabla_a Q_{diff}^w(s,a)|_{a=\pi(s)} \nabla_\theta \pi(s,\theta) \, ds.$$

Let us consider the case of linear function approximator with parameter $w$, i.e., $Q_{diff}^w(s,\pi(s)) = \phi^\pi(s,\pi(s))^\intercal w$.

We know from above,

$$\begin{aligned}
\nabla_w \nabla_a Q_{diff}^w(s,a)|_{a=\pi(s)} &= \nabla_\theta \pi(s) \\
\implies \nabla_a \phi^\pi(s,a)|_{a=\pi(s)} &= \nabla_\theta \pi(s).
\end{aligned} \tag{A.1}$$

Thus,

$$\begin{aligned}
Q_{diff}^w(s,\pi(s)) &= \phi^\pi(s,\pi(s))^\intercal w \\
\implies \nabla_a Q_{diff}^w(s,a)|_{a=\pi(s)} &= \nabla_a \phi^\pi(s,a)|_{a=\pi(s)}^\intercal w \\
\implies \nabla_a Q_{diff}^w(s,a)|_{a=\pi(s)} &= \nabla_\theta \pi(s)^\intercal w \quad \text{(using } (A.1)).
\end{aligned}$$

$\square$

**Theorem 2.** *The approximate gradient of the average reward $\rho(\pi)$ with respect to the policy parameter $\theta$ is given by the following expression:*

$$\widehat{\nabla_\theta \rho}(\pi) = \int_S d^\mu(s) \nabla_a Q^\pi_{diff}(s,a)|_{a=\pi(s)} \nabla_\theta \pi(s,\theta)\, ds.$$

*Further, the approximation error is $\mathcal{E}(\pi,\mu) = \|\nabla_\theta \rho(\pi) - \widehat{\nabla_\theta \rho}(\pi)\|$, where $\mu$ represents the behaviour policy. $\mathcal{E}$ satisfies*

$$\mathcal{E}(\pi,\mu) \leq Z\|\theta^\pi - \theta^\mu\|.$$

*Here, $Z = 2^{m+1} C(\lceil \log_\kappa a^{-1} \rceil + 1/\kappa) L_t$ with $L_t$ being the Lipchitz constant for the transition probability density function (Assumption 9). Constants $a$ and $\kappa$ are from Assumption 2, $m$ is the dimension of action space, and $C = \max_s \|\nabla_a Q^\pi_{diff}(s,a)|_{a=\pi(s)} \nabla_\theta \pi(s,\theta)\|.$*

*Proof.*

$$\mathcal{E}(\pi,\mu) = \|\nabla_\theta \rho(\pi) - \widehat{\nabla_\theta \rho}(\pi)\|$$

$$= \| \int_S d^\pi(s) \nabla_a Q^\pi_{diff}(s,a)|_{a=\pi(s)} \nabla_\theta \pi(s,\theta)\, ds$$

$$- \int_S d^\mu(s) \nabla_a Q^\pi_{diff}(s,a)|_{a=\pi(s)} \nabla_\theta \pi(s,\theta)\, ds\|$$

$$\leq \int_S |d^\pi(s) - d^\mu(s)| \|\nabla_a Q^\pi_{diff}(s,a)|_{a=\pi(s)} \nabla_\theta \pi(s,\theta)\|\, ds$$

$$\leq C \int_S |d^\pi(s) - d^\mu(s)|\, ds.$$

Here, $C = \max_s \|\nabla_a Q^\pi_{diff}(s,a)|_{a=\pi(s)} \nabla_\theta \pi(s,\theta)\|$. Thus,

$$\mathcal{E}(\pi,\mu) \leq C L_d \|\theta^\pi - \theta^\mu\| = Z\|\theta^\pi - \theta^\mu\| \text{ (using } Lemma12).$$

Here, $Z = 2^{m+1} C(\lceil \log_\kappa a^{-1} \rceil + 1/\kappa) L_t$.

$\square$

**Lemma 3.** *Let the cumulative error of on-policy actor be $\sum_{t=0}^{T-1} E\|\nabla_\theta \rho(\theta_t)\|^2$ and cumulative error of critic be $\sum_{t=0}^{T-1} E\|\Delta w_t\|^2$. $\theta_t$ and $w_t$ are the actor and linear critic parameter at time t.Bound on the cumulative error of on-policy actor is proven using cumulative error of critic as follows:*

$$\frac{1}{T} \sum_{t=0}^{T-1} E\|\nabla_\theta \rho(\theta_t)\|^2 \leq 2\frac{C_r}{C_\gamma} T^{v-1} + 3C_\pi^4 (\frac{1}{T} \sum_{t=0}^{T-1} E\|\Delta w_t\|^2) + 3C_\pi^4 (\tau^2 + \frac{4}{M} C_{w_\epsilon^*}^2),$$

$$+ \frac{C_\gamma L_J G_\theta^2}{1-v} T^{-v}$$

*Here, $C_r$ is the upper bound on rewards (Assumption 5), $C_\gamma$, v are constants used for step size $\gamma_t$ (Assumption 3, $\|\nabla_\theta \pi(s)\| \leq C_\pi$ (Assumption 7), $\Delta w_t = w_t - w_t^*$, $\tau = \max_t \|w_t^* - w_{\epsilon,t}^*\|$, $w_\epsilon^*$ is the optimal critic parameter according to Lemma 2. $w_t^*$ is the optimal parameters given by TD(0) algorithm corresponding to policy parameter $\theta_t$. Constant $C_{w_\epsilon^*}$ is defined in Lemma 13. $L_J$ is the coefficient used in smoothness condition of the non convex function $\rho(\theta)$. Constant $G_\theta$ is defined in Lemma 7. M is the size of batch of samples used to update parameters.*

*Proof.* By $[-L_J, L_J]$-smoothness of non-convex function we have:

$$E[\rho(\theta_{t+1})] \geq E[\rho(\theta_t)] + E\langle \nabla_\theta \rho(\theta_t), \theta_{t+1} - \theta_t \rangle - \frac{L_J}{2} E\|\theta_{t+1} - \theta_t\|^2. \tag{A.2}$$

Now,

$$h(B_t, w_t, \theta_t) = \frac{1}{M} \sum_i \nabla_a Q^\pi(s_{t,i}, a)|_{a=\pi(s_{t,i})} \nabla_\theta \pi(s_{t,i}).$$

$$
\begin{aligned}
E\langle \nabla_\theta \rho(\theta_t), \theta_{t+1} - \theta_t \rangle &= \gamma_t E\langle \nabla_\theta \rho(\theta_t), h(B_t, w_t, \theta_t) \rangle \\
&= \gamma_t E\langle \nabla_\theta \rho(\theta_t), h(B_t, w_t, \theta_t) - \nabla_\theta \rho(\theta_t) \rangle + \gamma_t E\|\nabla_\theta \rho(\theta_t)\|^2.
\end{aligned}
\tag{A.3}
$$

From (A.3), we have

$$
E\langle \nabla_\theta \rho(\theta_t), h(B_t, w_t, \theta_t) - \nabla_\theta \rho(\theta_t) \rangle \geq -\frac{1}{2} E\|\nabla_\theta \rho(\theta_t)\|^2 - \frac{1}{2} E\|h(B_t, w_t, \theta_t) - \nabla_\theta \rho(\theta_t)\|^2
$$
$$
(\because x^\intercal y \geq -\|x\|^2/2 - \|y\|^2/2).
$$
$$\tag{A.4}$$

From (A.4):
$$
\begin{aligned}
&E\|h(B_t, w_t, \theta_t) - \nabla_\theta \rho(\theta_t)\|^2 \\
&= E\|h(B_t, w_t, \theta_t) - h(B_t, w_t^*, \theta_t) + h(B_t, w_t^*, \theta_t) - h(B_t, w_{\epsilon,t}^*, \theta_t) + h(B_t, w_{\epsilon,t}^*, \theta_t) - \nabla_\theta \rho(\theta_t)\|^2 \\
&\leq 3(E\|h(B_t, w_t, \theta_t) - h(B_t, w_t^*, \theta_t)\|^2 \quad \text{①} \\
&\quad + E\|h(B_t, w_t^*, \theta_t) - h(B_t, w_{\epsilon,t}^*, \theta_t)\|^2 \quad \text{②} \\
&\quad + E\|h(B_t, w_{\epsilon,t}^*, \theta_t) - \nabla_\theta \rho(\theta_t)\|^2 \quad \text{③}
\end{aligned}
$$
$$\tag{A.5}$$

From (A.5):

①:

$$
\begin{aligned}
&E\|h(B_t, w_t, \theta_t) - h(B_t, w_t^*, \theta_t)\|^2 \\
&= \frac{1}{M}\|\sum_{i=0} \nabla_a Q^{w_t}(s_{t,i}, a)|_{a=\pi(s_{t,i})} \nabla_\theta \pi(s_{t,i}) - \sum_{i=0} \nabla_a Q^{w_t^*}(s_{t,i}, a)|_{a=\pi(s_{t,i})} \nabla_\theta \pi(s_{t,i})\|^2.
\end{aligned}
$$

Here, by compatible function approximation lemma 2: $\nabla_a Q^{w_t^*}(s_i, a)|_{a=\pi(s_i)} = \nabla_\theta \pi(s)^\intercal w$.

$$
\begin{aligned}
E\|h(B_t, w_t, \theta_t) - h(B_t, w_t^*, \theta_t)\|^2 &= E\|\frac{1}{M} \sum_{i=0} \nabla_\theta \pi(s_{t,i}) \nabla_\theta \pi(s_{t,i})^\intercal (w_t - w_t^*)\|^2 \\
&\leq C_\pi^4 E\|w_t - w_t^*\|^2.
\end{aligned}
$$

② is similar as ①:

$$
\begin{aligned}
E\|h(B_t, w_t^*, \theta_t) - h(B_t, w_{\epsilon,t}^*, \theta_t)\|^2 &\leq C_\pi^4 E\|w_t^* - w_{\epsilon,t}^*\|^2 \\
&\leq C_\pi^4 \tau^2.
\end{aligned}
$$

③ :

- By compatible function approximation lemma 2: $\nabla_\theta \rho(\theta_t) = \int_S d(s, \pi(\theta_t)) \nabla_\theta \pi(s) \nabla_\theta \pi(s)^\intercal w_{\epsilon,t}^* \, ds = E[h(B_t, w_{\epsilon,t}^*, \theta_t)]$

- By lemma 4 (Xiong et al., 2022), if $E[\hat{Y}] = \bar{Y}, \|\hat{Y}\|, \|\bar{Y}\| \leq C_Y$ then,

$$
E\|\frac{1}{M} \sum_{i=0}^{M-1} \hat{Y}_i - \bar{Y}\| \leq 4\frac{C_Y^2}{M}.
$$

Using above two bullet points:

$$E||h(B_t, w^*_{\epsilon,t}, \theta_t) - \nabla_\theta \rho(\theta_t)||^2 \leq \frac{4}{M}||\nabla_\theta \pi(s) \nabla_\theta \pi(s)^\intercal w^*_{\epsilon,t}||^2$$
$$\leq \frac{4 C_\pi^4 C_{w_\epsilon}^2}{M}.$$

Combining ①,② and ③ and using in (A.5):

$$E||h(B_t, w_t, \theta_t) - \nabla_\theta \rho(\theta_t)||^2 \leq 3 C_\pi^4 (E||w_t - w^*_t||^2 + \tau^2 + \frac{4 C_{w_\epsilon}^2}{M}). \tag{A.6}$$

Using (A.6) in (A.4):

$$E\langle \nabla_\theta \rho(\theta_t), h(B_t, w_t, \theta_t) - \nabla_\theta \rho(\theta_t) \rangle \geq -\frac{1}{2} E||\nabla_\theta \rho(\theta_t)||^2$$
$$- \frac{3}{2} C_\pi^4 (E||w_t - w^*_t||^2 + \tau^2 + \frac{4 C_{w_\epsilon}^2}{M}). \tag{A.7}$$

Using (A.7) in (A.3):

$$E\langle \nabla_\theta \rho(\theta_t), \theta_{t+1} - \theta_t \rangle \geq \frac{\gamma_t}{2} E||\nabla_\theta \rho(\theta_t)||^2$$
$$- \frac{3 \gamma_t}{2} C_\pi^4 (E||w_t - w^*_t||^2 + \tau^2 + \frac{4 C_{w_\epsilon}^2}{M}). \tag{A.8}$$

Using (A.8) in (A.2):

$$E[\rho(\theta_{t+1})] - E[\rho(\theta_t)] \geq \frac{\gamma_t}{2} E||\nabla_\theta \rho(\theta_t)||^2 - \frac{L_J}{2} E||\theta_{t+1} - \theta_t||^2$$
$$- \frac{3 \gamma_t}{2} C_\pi^4 (E||w_t - w^*_t||^2 + \tau^2 + \frac{4 C_{w_\epsilon}^2}{M})$$

$$\implies E||\nabla_\theta \rho(\theta_t)||^2 \geq \frac{2}{\gamma_t} E[\rho(\theta_{t+1})] - E[\rho(\theta_t)] + 3 C_\pi^4 (E||w_t - w^*_t||^2)$$
$$+ 3 C_\pi^4 (\tau^2 + \frac{4 C_{w_\epsilon}^2}{M}) + L_J \gamma_t G_\theta^2 \quad \text{(using lemma 7)}$$

$$\implies \sum_{t=0}^{T-1} E||\nabla_\theta \rho(\theta_t)||^2 \geq \sum_{t=0}^{T-1} \frac{2}{\gamma_t} E[\rho(\theta_{t+1})] - E[\rho(\theta_t)] \quad ①$$
$$+ \sum_{t=0}^{T-1} 3 C_\pi^4 (E||w_t - w^*_t||^2) \quad ②$$
$$+ \sum_{t=0}^{T-1} 3 C_\pi^4 (\tau^2 + \frac{4 C_{w_\epsilon}^2}{M}) \quad ③$$
$$+ \sum_{t=0}^{T-1} L_J \gamma_t G_\theta^2 \quad ④ \quad \text{(using lemma 7)} \tag{A.9}$$

From equation A.9

①:

$$\sum_{t=0}^{T-1} \frac{2}{\gamma_t} E[\rho(\theta_{t+1})] - E[\rho(\theta_t)] = 2\bigg(\sum_{t=0}^{T-1}\Big(\frac{1}{\gamma_t} - \frac{1}{\gamma_{t-1}}\Big) E[\rho(\theta_t)] + \frac{E[\rho(\theta_0)]}{\gamma_0} - \frac{E[\rho(\theta_T)]}{\gamma_{T-1}}\bigg)$$

$$\leq 2\bigg(\sum_{t=0}^{T-1}\Big(\frac{1}{\gamma_t} - \frac{1}{\gamma_{t-1}}\Big) E[\rho(\theta_t)] + \frac{E[\rho(\theta_0)]}{\gamma_0}\bigg)$$

$$\leq 2\bigg(\sum_{t=0}^{T-1}\Big(\frac{1}{\gamma_t} - \frac{1}{\gamma_{t-1}}\Big) + \gamma_0\bigg)C_r$$

$$\leq \frac{2C_r}{\gamma_{T-1}} = \frac{2C_r T^v}{C_\gamma}$$

②:

$$\sum_{t=0}^{T-1} 3C_\pi^4(E||w_t - w_t^*||^2) = \sum_{t=0}^{T-1} 3C_\pi^4(E||\Delta w_t||^2)$$

④:

$$\sum_{t=0}^{T-1} L_J \gamma_t G_\theta^2 \leq L_J G_\theta^2 C_\gamma \frac{T^{1-v}}{1-v} \quad \Big(\because \sum_{t=0}^{T-1}\frac{1}{1+t^v} \leq \int_0^T \frac{1}{t^v}\, dt = \frac{T^{1-v}}{1-v}\Big)$$

Using ①-④ and dividing equation A.9 by T:

$$\frac{1}{T}\sum_{t=0}^{T-1} E||\nabla_\theta \rho(\theta_t)||^2 \leq 2\frac{C_r}{C_\gamma}T^{v-1} + 3C_\pi^4\Big(\frac{1}{T}\sum_{t=0}^{T-1} E||\Delta w_t||^2\Big) + 3C_\pi^4\Big(\tau^2 + \frac{4}{M}C_{w_\epsilon}^2\Big)$$

$$+ \frac{C_\gamma L_J G_\theta^2}{1-v}T^{-v}$$

$\square$

**Lemma 4.** *Let the cumulative error of linear critic be $\sum_{t=0}^{T-1}\mathbb{E}||\Delta w_t||^2$ and cumulative error of average reward estimator be $\sum_{t=0}^{T-1}\mathbb{E}||\Delta\rho_t||^2$. $w_t$ and $\rho_t$ are linear critic parameter and average reward estimator at time t respectively. Bound on the cumulative error of critic is proven using cumulative error of average reward estimator as follows:*

$$\frac{1}{T}\sum_{t=0}^{T-1}\mathbb{E}||\Delta w_t||^2 \leq 2\bigg(\sqrt{\frac{2C_w^2}{\lambda C_\alpha}T^{\sigma-1} + \frac{C_g C_\alpha}{1-\sigma}T^{-\sigma} + }$$

$$\frac{L_w G_\theta}{\lambda}\Big(\frac{1}{T}\sum\Big(\frac{\gamma_t}{\alpha_t}\Big)^2\Big)^{1/2} + \frac{2(C_r + 3C_w)}{\lambda}\bigg)^2 + \frac{2}{\lambda^2}\frac{1}{T}\sum_{t=0}^{T-1}\mathbb{E}||\Delta\rho_t||^2$$

*Here, $\Delta w_t = w_t - w_t^*$, $\Delta\rho_t = \rho_t - \rho_t^*$. $w_t^*$ and $\rho_t^*$ are the optimal parameters given by TD(0) algorithm corresponding to policy parameter $\theta_t$. $C_\alpha$, $\sigma$ are constants and $\gamma_t, \alpha_t$ are step-sizes defined in Assumption 3, $\|w_t\| \leq C_w$ (Algorithm 2, step 8), $C_r$ is the upper bound on rewards (Assumption 5), Constant $G_\theta$ is defined in Lemma 7, $C_g = \frac{L_w^2}{\lambda}\max_t \frac{\gamma_t^2}{\alpha_t^2}G_\theta^2 + \frac{C_\delta^2}{\lambda}$, $C_\delta = 2C_r + (4+\eta)C_w$. $\eta$ is the l2-regularisation coefficient from Algorithm 2 and $\eta > \lambda_{max}^{all}$, where $\lambda_{max}^{all}$ is defined in Lemma 11. $\lambda$ is defined in Lemma 11. $L_w$ is defined in Lemma 6.*

*Proof.*

$$w_{t+1} = w_t + \alpha_t \frac{1}{M} \sum_{i=0}^{M-1} \left( R^\pi(s_{t,i}) - \bar{\rho}_t + \phi^\pi(s'_{t,i})^\intercal \bar{w}_t - \phi^\pi(s_{t,i})^\intercal w_t \right) \phi^\pi(s_{t,i}) - \alpha_t \eta w_t$$

$$\implies w_{t+1} - w^*_{t+1} = w_t - w^*_t + w^*_t - w^*_{t+1} \quad \textcircled{1}$$

$$+ \alpha_t \frac{1}{M} \sum_{i=0}^{M-1} \left( R^\pi(s_{t,i}) - \rho^*_t + \phi^\pi(s'_{t,i})^\intercal \bar{w}_t - \phi^\pi(s_{t,i})^\intercal w_t \right) \phi^\pi(s_{t,i}) - \alpha_t \eta w_t \quad \textcircled{2}$$

$$+ \alpha_t \frac{1}{M} \sum_{i=0}^{M-1} \left( \rho^*_t - \rho_t \right) \phi^\pi(s_{t,i}) \quad \textcircled{3}$$

$$+ \alpha_t \frac{1}{M} \sum_{i=0}^{M-1} \left( \rho_t - \bar{\rho}_t \right) \phi^\pi(s_{t,i}) \quad \textcircled{4}$$

$$\tag{A.10}$$

From equation A.10:
$\textcircled{2}$:

$$\frac{1}{M} \sum_{i=0}^{M-1} \left( R^\pi(s_{t,i}) - \rho^*_t + \phi^\pi(s'_{t,i})^\intercal \bar{w}_t - \phi^\pi(s_{t,i})^\intercal w_t \right) \phi^\pi(s_{t,i}) - \eta w_t$$

$$= \frac{1}{M} \sum_{i=0}^{M-1} \left( R^\pi(s_{t,i}) - \rho^*_t + \phi^\pi(s'_{t,i})^\intercal w_t - \phi^\pi(s_{t,i})^\intercal w_t \right) \phi^\pi(s_{t,i}) - \eta w_t$$

$$+ \frac{1}{M} \sum_{i=0}^{M-1} \phi^\pi(s_{t,i}) \phi^\pi(s'_{t,i})^\intercal (\bar{w}_t - w_t)$$

$$= \frac{1}{M} \sum_{i=0}^{M-1} \phi^\pi(s_{t,i}) \phi^\pi(s'_{t,i})^\intercal (\bar{w}_t - w_t) + g(B_t, w_t, \theta_t) - \bar{g}(w_t, \theta_t)$$

$$+ \bar{g}(w_t, \theta_t) - \bar{g}(w^*_t, \theta_t)$$

$$\tag{A.11}$$

Let $g(B_t, w_t, \theta_t) := \frac{1}{M} \sum_{i=0}^{M-1} \left( R^\pi(s_{t,i}) - \rho^*_t \right) \phi^\pi(s_{t,i}) + \frac{1}{M} \sum_{i=0}^{M-1} \left( \phi^\pi(s_{t,i})(\phi^\pi(s'_{t,i}) - \phi^\pi(s_{t,i}))^\intercal - \eta I \right) w_t$

Let $\bar{g}(w_t, \theta_t) := \int d(s, \pi(\theta_t)) \phi^\pi(s) \left( r^\pi(s) - \rho^*_t + \int \rho^\pi(s'|s) \phi^\pi(s')^\intercal w_t \, ds' - \phi^\pi(s)^\intercal w_t \right) ds$

Using equation A.11 in equation A.10:

$$w_{t+1} - w^*_{t+1} = w_t - w^*_t + w^*_t - w^*_{t+1} +$$

$$+ \alpha_t \frac{1}{M} \sum_{i=0}^{M-1} (\rho^*_t - \rho_t) \phi^\pi(s_{t,i})$$

$$+ \alpha_t \frac{1}{M} \sum_{i=0}^{M-1} (\rho_t - \bar{\rho}_t) \phi^\pi(s_{t,i})$$

$$+ \alpha_t \frac{1}{M} \sum_{i=0}^{M-1} \phi^\pi(s_{t,i}) \phi^\pi(s'_{t,i})^\intercal (\bar{w}_t - w_t)$$

$$+ \alpha_t (g(B_t, w_t, \theta_t) - \bar{g}(w_t, \theta_t))$$

$$+ \alpha_t (\bar{g}(w_t, \theta_t) - \bar{g}(w^*_t, \theta_t))$$

$$\text{Let,} \quad f(B_t, w_t, \theta_t) := \frac{1}{M} \sum_{i=0}^{M-1} (\rho_t^* - \rho_t) \phi^\pi(s_{t,i})$$

$$+ \frac{1}{M} \sum_{i=0}^{M-1} (\rho_t - \bar{\rho}_t) \phi^\pi(s_{t,i})$$

$$+ \frac{1}{M} \sum_{i=0}^{M-1} \phi^\pi(s_{t,i}) \phi^\pi(s'_{t,i})^\mathsf{T} (\bar{w}_t - w_t)$$

$$+ g(B_t, w_t, \theta_t) - \bar{g}(w_t, \theta_t)$$
$$+ \bar{g}(w_t, \theta_t) - \bar{g}(w_t^*, \theta_t)$$

$$||w_{t+1} - w_{t+1}^*||^2 = ||(w_t - w_t^*) + (w_t^* - w_{t+1}^*) + \alpha_t f(B_t, w_t, \theta_t)||^2$$
$$= ||w_t - w_t^*||^2 + ||w_t^* - w_{t+1}^*||^2$$
$$+ \alpha_t^2 ||f(B_t, w_t, \theta_t)||^2$$
$$+ 2\langle \Delta w_t, w_t^* - w_{t+1}^* \rangle + 2\alpha_t \langle \Delta w_t, f(B_t, w_t, \theta_t) \rangle$$
$$+ 2\alpha_t \langle w_t^* - w_{t+1}^*, f(B_t, w_t, \theta_t) \rangle$$

$$\mathbb{E}||w_{t+1} - w_{t+1}^*||^2 \leq \mathbb{E}||\Delta w_t||^2 + 2\mathbb{E}||w_t^* - w_{t+1}^*||^2$$
$$+ 2\alpha_t^2 E||f(B_t, w_t, \theta_t)||^2$$
$$+ 2\mathbb{E}\langle \Delta w_t, w_t^* - w_{t+1}^* \rangle$$
$$+ 2\alpha_t \mathbb{E}\langle \Delta w_t, f(B_t, w_t, \theta_t) \rangle$$
$$= \mathbb{E}||\Delta w_t||^2 + 2\mathbb{E}||w_t^* - w_{t+1}^*||^2 \quad ①$$
$$+ 2\alpha_t^2 \mathbb{E}||f(B_t, w_t, \theta_t)||^2 \quad ②$$
$$+ 2\mathbb{E}\langle \Delta w_t, w_t^* - w_{t+1}^* \rangle \quad ③$$
$$+ 2\alpha_t \mathbb{E}\langle \Delta w_t, \frac{1}{M} \sum_{i=0}^{M-1} (\rho_t^* - \rho_t) \phi^\pi(s_{t,i}) \rangle \quad ④$$
$$+ 2\alpha_t \mathbb{E}\langle \Delta w_t, \frac{1}{M} \sum_{i=0}^{M-1} (\rho_t - \bar{\rho}_t) \phi^\pi(s_{t,i}) \rangle \quad ⑤$$
$$+ 2\alpha_t \mathbb{E}\langle \Delta w_t, \frac{1}{M} \sum_{i=0}^{M-1} \phi^\pi(s_{t,i}) \phi^\pi(s'_{t,i})^\mathsf{T} (\bar{w}_t - w_t) \rangle \quad ⑥$$
$$+ 2\alpha_t \mathbb{E}\langle \Delta w_t, g(B_t, w_t, \theta_t) - \bar{g}(w_t, \theta_t) \rangle \quad ⑦$$
$$+ 2\alpha_t \mathbb{E}\langle \Delta w_t, \bar{g}(w_t, \theta_t) - \bar{g}(w_t^*, \theta_t) \rangle \quad ⑧$$

(A.12)

From equation A.12:

①:

$$\mathbb{E}||w_t^* - w_{t+1}^*||^2 \leq L_w^2 \mathbb{E}||\theta_{t+1} - \theta_t||^2 \quad \text{(using lemma 6)}$$
$$\leq L_w^2 \gamma_t^2 G_\theta^2 \quad \text{(using lemma 7)}$$

②:

$$\mathbb{E}||f(B_t, w_t, \theta_t)||^2$$
$$= \mathbb{E}||\frac{1}{M} \sum_{i=0}^{M-1} (R^\pi(s_{t,i}) - \bar{\rho}_t + \phi^\pi(s'_{t,i})^\mathsf{T} \bar{w}_t - \phi^\pi(s_{t,i})^\mathsf{T} w_t) \phi^\pi(s_{t,i}) - \eta w_t||^2$$
$$\leq \mathbb{E}\big(||\frac{1}{M} \sum_{i=0}^{M-1} (R^\pi(s_{t,i}) - \bar{\rho}_t + \phi^\pi(s'_{t,i})^\mathsf{T} \bar{w}_t - \phi^\pi(s_{t,i})^\mathsf{T} w_t) \phi^\pi(s_{t,i})|| + \eta ||w_t||\big)^2$$

Here,

$$||\phi^\pi(s)|| < 1 \quad \text{(Assumption 4)}$$
$$|R^\pi(s)| \leq C_r \quad \text{(Assumption 5)}$$
$$||w_t|| \leq C_w \quad \text{(Algorithm 2, step 8)}$$
$$|\rho_t| \leq C_r + 2C_w \quad \text{(lemma 8)}$$
$$||\bar{w}_t|| \leq C_w \quad \text{(lemma 9)}$$
$$||\bar{\rho}_t|| \leq C_r + 2C_w \quad \text{(lemma 10)}$$

$$\leq \mathbb{E}\Big(\frac{1}{M}\sum_{i=0}^{M-1}||(R^\pi(s_{t,i}) - \bar{\rho}_t + \phi^\pi(s'_{t,i})^\mathsf{T}\bar{w}_t - \phi^\pi(s_{t,i})^\mathsf{T}w_t)\phi^\pi(s_{t,i})|| + \eta||w_t||\Big)^2$$
$$\leq \mathbb{E}(C_r + C_r + 2C_w + 2C_w + \eta C_w)^2$$
$$\leq \mathbb{E}(C_\delta)^2 \quad (C_\delta = 2C_r + (4+\eta)C_w)$$
$$\leq C_\delta^2$$

③:

$$\mathbb{E}\langle \Delta w_t, w_t^* - w_{t+1}^*\rangle \leq \mathbb{E}||\Delta w_t||\,||w_t^* - w_{t+1}^*||$$
$$\leq L_w \mathbb{E}||\Delta w_t||\,||\theta_{t+1} - \theta_t|| \quad \text{(using Lemma 6)}$$

④:

$$\mathbb{E}[\langle \Delta w_t, \frac{1}{M}\sum_{i=0}^{M-1}(\rho_t^* - \rho_t)\phi^\pi(s_{t,i})\rangle] = \mathbb{E}[\frac{1}{M}\sum_{i=0}^{M-1}\langle \Delta w_t, \phi^\pi(s_{t,i})\rangle(\rho_t^* - \rho_t)]$$
$$\leq \mathbb{E}[\frac{1}{M}\sum_{i=0}^{M-1}||\Delta w_t||\,||\phi^\pi(s_{t,i})||\,||(\rho_t^* - \rho_t)||]$$
$$\leq \mathbb{E}||\Delta w_t||\,|\rho_t^* - \rho_t|$$
$$= \mathbb{E}||\Delta w_t||\,|\Delta \rho_t|$$

⑤:

$$\mathbb{E}[\langle \Delta w_t, \frac{1}{M}\sum_{i=0}^{M-1}(\rho_t - \bar{\rho}_t)\phi^\pi(s_{t,i})\rangle] = \mathbb{E}[\frac{1}{M}\sum_{i=0}^{M-1}\langle \Delta w_t, \phi^\pi(s_{t,i})\rangle(\rho_t - \bar{\rho}_t)]$$
$$\leq \mathbb{E}[\frac{1}{M}\sum_{i=0}^{M-1}||\Delta w_t||\,||\phi^\pi(s_{t,i})||\,|\rho_t - \bar{\rho}_t|]$$
$$\leq \mathbb{E}[||\Delta w_t||\,|\rho_t - \bar{\rho}_t|]$$
$$\leq \mathbb{E}[||\Delta w_t||\,(|\rho_t| + |\bar{\rho}_t|)]$$
$$\leq \mathbb{E}[||\Delta w_t||]2(C_r + 2C_w) \quad \text{(using Lemma 8, 10)}$$

⑥:

$$\mathbb{E}\langle \Delta w_t, \frac{1}{M}\sum_{i=0}^{M-1}\phi^\pi(s_{t,i})\phi^\pi(s'_{t,i})^\mathsf{T}(\bar{w}_t - w_t)\rangle \leq \mathbb{E}||\Delta w_t||\,||\frac{1}{M}\sum_{i=0}^{M-1}\phi^\pi(s_{t,i})\phi^\pi(s'_{t,i})^\mathsf{T}(\bar{w}_t - w_t)||$$
$$\leq \mathbb{E}||\Delta w_t||\,||\bar{w}_t - w_t||$$
$$\leq 2C_w\mathbb{E}||\Delta w_t|| \quad \text{(using algorithm 2)}$$

⑦:

$$\mathbb{E}[\langle \Delta w_t, g(B_t, w_t, \theta_t) - \bar{g}(w_t, \theta_t)\rangle] = \mathbb{E}[\langle \Delta w_t, \mathbb{E}[g(B_t, w_t, \theta_t) - \bar{g}(w_t, \theta_t)|\Delta w_t]\rangle]$$
Note: $\quad \mathbb{E}[g(B_t, w_t, \theta_t) - \bar{g}(w_t, \theta_t)] = 0$
Hence, $\quad \mathbb{E}[\langle \Delta w_t, g(B_t, w_t, \theta_t) - \bar{g}(w_t, \theta_t)\rangle] = 0$

⑧:

$$\mathbb{E}[\langle \Delta w_t, \bar{g}(w_t, \theta_t) - \bar{g}(w_t^*, \theta_t) \rangle]$$

$$A(\theta_t) = \int_S d^\pi(s, \theta_t)(\phi^\pi(s)(\mathbb{E}[\phi^\pi(s')] - \phi^\pi(s)^\intercal - \eta I)ds$$

$$b(\theta_t) = \int_S d^\pi(s, \theta_t)r^\pi(s)\phi^\pi(s)ds$$

$$\bar{g}(w_t, \theta_t) - \bar{g}(w_t^*, \theta_t) = b(\theta_t) + A(\theta_t)w_t - b(\theta_t) - A(\theta_t)w_t^*$$

$$= A(\theta_t)(w_t - w_t^*)$$

$$\text{Now,} \quad \mathbb{E}[\langle \Delta w_t, \bar{g}(w_t, \theta_t) - \bar{g}(w_t^*, \theta_t) \rangle] = \mathbb{E}[\langle \Delta w_t, A(\theta_t)\Delta w_t \rangle]$$

$$= \mathbb{E}[\Delta w_t^\intercal A(\theta_t)\Delta w_t]$$

$$\le -\lambda \mathbb{E}||\Delta w_t||^2 \qquad \text{(Lemma 11)}$$

Combining ① - ⑧ into equation A.12:

$$\mathbb{E}||w_{t+1} - w_{t+1}^*||^2 \le (1 - 2\lambda\alpha_t)\mathbb{E}||\Delta w_t||^2 + 2L_w^2\gamma_t^2 G_\theta^2 + 2\alpha_t^2 C_\delta^2$$

$$+ 2L_w\mathbb{E}||\Delta w_t||||\theta_{t+1} - \theta_t|| + 2\alpha_t\mathbb{E}||\Delta w_t|||\Delta\rho_t|$$

$$+ 4\alpha_t\mathbb{E}||\Delta w_t||(2C_w + C_r) + 4\alpha_t C_w\mathbb{E}||\Delta w_t||$$

$$\implies 2\lambda\alpha_t\mathbb{E}||\Delta w_t||^2 \le \mathbb{E}[||\Delta w_t||^2] - \mathbb{E}||\Delta w_{t+1}||^2 + 2L_w^2\gamma_t^2 G_\theta^2 + 2\alpha_t^2 C_\delta^2$$

$$+ 2L_w\gamma_t G_\theta\mathbb{E}||\Delta w_t|| + 2\alpha_t\mathbb{E}||\Delta w_t|||\Delta\rho_t|$$

$$+ 4\alpha_t(C_r + 3C_w)\mathbb{E}||\Delta w_t||$$

$$\implies \mathbb{E}||\Delta w_t||^2 \le \frac{1}{2\lambda\alpha_t}(\mathbb{E}||\Delta w_t||^2 - \mathbb{E}||w_{t+1}||^2)$$

$$+ \Big(\frac{L_w^2\gamma_t^2}{\lambda\alpha_t}G_\theta^2 + \frac{\alpha_t}{\lambda}C_\delta^2\Big)$$

$$+ \frac{L_w}{\lambda}\frac{\gamma_t}{\alpha_t}G_\theta\mathbb{E}||\Delta w_t||$$

$$+ \frac{\mathbb{E}||\Delta w_t|||\Delta\rho_t|}{\lambda}$$

$$+ \frac{2}{\lambda}(C_r + 3C_w)\mathbb{E}||\Delta w_t||$$

$$\implies \sum_{t=0}^{T-1}\mathbb{E}||\Delta w_t||^2 \le \sum_{t=0}^{T-1}\frac{1}{2\lambda\alpha_t}(\mathbb{E}||\Delta w_t||^2 - \mathbb{E}||\Delta w_{t+1}||^2) \quad ①$$

$$+ \sum_{t=0}^{T-1}\Big(\frac{L_w}{\lambda}\frac{\gamma_t^2}{\alpha_t}G_\theta^2 + \frac{\alpha_t}{\lambda}C_\delta^2\Big) \quad ②$$

$$+ \sum_{t=0}^{T-1}\frac{L_w}{\lambda}\frac{\gamma_t}{\alpha_t}G_\theta\mathbb{E}||\Delta w_t|| \quad ③ \qquad\qquad \text{(A.13)}$$

$$+ \sum_{t=0}^{T-1}\frac{\mathbb{E}||\Delta w_t|||\Delta\rho_t|}{\lambda} \quad ④$$

$$+ \sum_{t=0}^{T-1}\frac{2}{\lambda}(C_r + 3C_w)\mathbb{E}||\Delta w_t|| \quad ⑤$$

From equation A.13:

①:

$$\frac{1}{2\lambda}\sum_{t=0}^{T-1}(\mathbb{E}||\Delta w_t||^2 - \mathbb{E}||\Delta w_{t+1}||^2)\frac{1}{\alpha_t} = \frac{1}{2\lambda}\Big(\sum_{t=1}^{T-1}\Big(\frac{1}{\alpha_t} - \frac{1}{\alpha_{t-1}}\Big)\mathbb{E}||\Delta w_t||^2 + \frac{1}{\alpha_0}\mathbb{E}||\Delta w_0||^2 - \frac{1}{\alpha_{T-1}}\mathbb{E}||\Delta w_T||^2\Big)$$

$$\leq \frac{1}{2\lambda}\Big(\sum_{t=1}^{T-1}\Big(\frac{1}{\alpha_t} - \frac{1}{\alpha_{t-1}}\Big) + \frac{1}{\alpha_0}\Big)4C_w^2$$

$$\leq \frac{4C_w^2}{2\lambda\alpha_{T-1}} = \frac{C_w^2}{\lambda C_\alpha}T^\sigma \quad (\because \alpha_t = \frac{C_\alpha}{(1+t)^\alpha})$$

②:

$$\sum_{t=0}^{T-1}\Big(\frac{L_w^2}{\lambda}\frac{\gamma_t^2}{\alpha_t}G_\theta^2 + \frac{\alpha_t}{\lambda}C_\delta^2\Big) = \sum_{t=0}^{T-1}\Big(\frac{L_w^2}{\lambda}\frac{\gamma_t^2}{\alpha_t^2}G_\theta^2 + \frac{C_\delta^2}{\lambda}\Big)\alpha_t$$

$$\leq \sum_{t=0}^{T-1}\Big(\frac{L_w^2}{\lambda}\max_t\frac{\gamma_t^2}{\alpha_t^2}G_\theta^2 + \frac{C_\delta^2}{\lambda}\Big)\alpha_t$$

$$= \sum_{t=0}^{T-1}C_g\alpha_t = \frac{C_g C_\alpha}{1-\sigma}T^{1-\sigma} \quad \Big(C_g = \frac{L_w^2}{\lambda}\max_t\frac{\gamma_t^2}{\alpha_t^2}G_\theta^2 + \frac{C_\delta^2}{\lambda}\Big)$$

③:

$$\sum_{t=0}^{T-1}\frac{L_w}{\lambda}\frac{\gamma_t}{\alpha_t}G_\theta\mathbb{E}||\Delta w_t|| = \frac{L_w}{\lambda}G_\theta\sum_{t=0}^{T-1}\frac{\gamma_t}{\alpha_t}\mathbb{E}||\Delta w_t||$$

$$\leq \frac{L_w}{\lambda}G_\theta\Big(\sum_{t=0}^{T-1}\Big(\frac{\gamma_t}{\alpha_t}\Big)^2\Big)^{\frac{1}{2}}\Big(\sum_{t=0}^{T-1}(\mathbb{E}||\Delta w_t||)^2\Big)^{\frac{1}{2}}$$

(Using Cauchy Schwartz inequality)

$$\leq \frac{L_w}{\lambda}G_\theta\Big(\sum_{t=0}^{T-1}\Big(\frac{\gamma_t}{\alpha_t}^2\Big)\Big)^{\frac{1}{2}}\Big(\sum_{t=0}^{T-1}\mathbb{E}||\Delta w_t||^2\Big)^{\frac{1}{2}}$$

(Using Jensen's inequality)

④:

$$\frac{1}{\lambda}\sum_{t=0}^{T-1}\mathbb{E}||\Delta w_t|||\Delta\rho_t| \leq \frac{1}{\lambda}\Big(\sum_{t=0}^{T-1}(\mathbb{E}||\Delta w_t||)^2\Big)^{\frac{1}{2}}\Big(\sum_{t=0}^{T-1}(\mathbb{E}|\Delta\rho_t|)^2\Big)^{\frac{1}{2}}$$

$$\leq \frac{1}{\lambda}\Big(\sum_{t=0}^{T-1}\mathbb{E}||\Delta w_t||^2\Big)^{\frac{1}{2}}\Big(\sum_{t=0}^{T-1}\mathbb{E}|\Delta\rho_t|^2\Big)^{\frac{1}{2}}$$

⑤:

$$\sum_{t=0}^{T-1}\frac{2(C_r + 3C_w)}{\lambda}\mathbb{E}||\Delta w_t|| \leq \frac{2(C_r + 3C_w)}{\lambda}\Big(\sum_{t=0}^{T-1}\mathbb{E}||\Delta w_t||^2\Big)^{\frac{1}{2}}\Big(\sum_{t=0}^{T-1}1\Big)^{\frac{1}{2}}$$

$$\leq \frac{2(C_r + 3C_w)}{\lambda}T^{\frac{1}{2}}\Big(\sum_{t=0}^{T-1}\mathbb{E}||\Delta w_t||^2\Big)^{\frac{1}{2}}$$

Combining ① - ⑤ into equation A.13:

$$\frac{1}{T}\sum_{t=0}^{T-1}\mathbb{E}||\Delta w_t||^2 \leq \frac{2C_w^2}{\lambda C_\alpha}T^{\sigma-1} + \frac{C_gC^\alpha}{1-\sigma}T^{-\sigma}$$

$$+ \frac{L_wG_\theta}{\lambda}\left(\frac{1}{T}\sum_{t=0}^{T-1}\left(\frac{\gamma_t}{\alpha_t}\right)^2\right)^{\frac{1}{2}}\left(\frac{1}{T}\sum_{t=0}^{T-1}\mathbb{E}||\Delta w_t||^2\right)^{\frac{1}{2}}$$

$$+ \frac{1}{\lambda}\left(\frac{1}{T}\sum_{t=0}^{T-1}\mathbb{E}||\Delta w_t||^2\right)^{\frac{1}{2}}\left(\frac{1}{T}\sum_{t=0}^{T-1}\mathbb{E}||\Delta\rho_t||^2\right)^{\frac{1}{2}}$$

$$+ \frac{2(C_r+3C_w)}{\lambda}\left(\frac{1}{T}\sum_{t=0}^{T-1}\mathbb{E}||\Delta w_t||^2\right)^{\frac{1}{2}}$$

Let,

$$M(T) = \frac{1}{T}\sum_{t=0}^{T-1}\mathbb{E}||\Delta w_t||^2$$

$$N(T) = \frac{1}{T}\sum_{t=0}^{T-1}\mathbb{E}|\Delta\rho_t|^2$$

$$M(T) \leq K_1 + K_2\sqrt{M(T)} + K_3\sqrt{M(T)}\sqrt{N(T)}$$

$$K_1 := \frac{2C_w^2}{\lambda C_\alpha}T^{\sigma-1} + \frac{C_gC_\alpha}{1-\sigma}T^{-\sigma}$$

$$K_2 := \frac{L_wG_\theta}{\lambda}\left(\frac{1}{T}\sum_{t=0}^{T-1}\left(\frac{\gamma_t}{\alpha_t}\right)^2\right)^{\frac{1}{2}} + \frac{2(C_r+3C_w)}{\lambda}$$

$$K_3 := \frac{1}{\lambda}$$

$$M(T) - 2\frac{K_2}{2}\sqrt{M(T)} - 2\frac{K_3}{2}\sqrt{M(T)}\sqrt{N(T)} + 2\frac{K_2}{2}\frac{K_3}{2}\sqrt{N(T)}$$

$$+ \left(\frac{K_2}{2}\right)^2 + \left(\frac{K_3}{2}\sqrt{N(T)}\right)^2 \leq K_1 + \left(\frac{K_2}{2}\right)^2 + \left(\frac{K_3}{2}\sqrt{N(T)}\right)^2 + 2\frac{K_2}{2}\frac{K_3}{2}\sqrt{N(T)}$$

$$\implies \left(\sqrt{M(T)} - \frac{K_2}{2} - \frac{K_3}{2}\sqrt{N(T)}\right)^2 \leq K_1 + \left(\frac{K_2}{2} + \frac{K_3}{2}\sqrt{N(T)}\right)^2$$

$$\implies \sqrt{M(T)} - \frac{K_2}{2} - \frac{K_3}{2}\sqrt{N(T)} \leq \sqrt{K_1} + \frac{K_2}{2} + \frac{K_3}{2}\sqrt{N(T)}$$

$$\implies \sqrt{M(T)} \leq \sqrt{K_1} + K_2 + K_3\sqrt{N(T)}$$

$$\implies M(T) \leq 2(\sqrt{K_1} + K_2)^2 + 2K_3^2 N(T)$$

$$\frac{1}{T}\sum_{t=0}^{T-1}\mathbb{E}||\Delta w_t||^2 \leq 2\left(\sqrt{\frac{2C_w^2}{\lambda C_\alpha}T^{\sigma-1} + \frac{C_gC_\alpha}{1-\sigma}T^{-\sigma}} + \frac{L_wG_\theta}{\lambda}\left(\frac{1}{T}\sum_{t=0}^{T-1}\left(\frac{\gamma_t}{\alpha_t}\right)^2\right)^{\frac{1}{2}} + \frac{2(C_r+3C_w)}{\lambda}\right)^2$$

$$+ \frac{2}{\lambda^2}\frac{1}{T}\sum_{t=0}^{T-1}\mathbb{E}||\Delta\rho_t||^2$$

$\square$

**Lemma 5.** *Let the cumulative error of linear critic be $\sum_{t=0}^{T-1} \mathbb{E}||\Delta w_t||^2$ and cumulative error of average reward estimator be $\sum_{t=0}^{T-1} \mathbb{E}||\Delta \rho_t||^2$. $w_t$ and $\rho_t$ are linear critic parameter and average reward estimator at time t respectively. Bound on the cumulative error of average reward estimator is proven using cumulative error of critic as follows:*

$$\frac{1}{T} \sum_{t=0}^{T-1} \mathbb{E}|\Delta \rho_t|^2 \leq 2\left( \sqrt{\frac{2(C_r + 2C_w)^2 T^{\sigma-1}}{C_\alpha} + \frac{C_s C_\alpha T^{-\sigma}}{1-\sigma}} \right.$$

$$\left. + L_p G_\theta \left( \frac{1}{T} \sum_{t=0}^{T-1} \left( \frac{\gamma_t}{\alpha_t} \right)^2 \right)^{\frac{1}{2}} + 4C_w \right)^2$$

$$+ \frac{8}{T} \sum_{t=0}^{T-1} \mathbb{E}||\Delta w_t||^2$$

*Here, $\Delta \rho_t = \rho_t - \rho_t^*$, $\Delta w_t = w_t - w_t^*$. $w_t^*$ and $\rho_t^*$ are the optimal parameters given by TD(0) algorithm corresponding to policy parameter $\theta_t$. $C_\alpha$, $\sigma$ are constants and $\gamma_t$, $\alpha_t$ are step-sizes defined in Assumption 3, $||w_t|| \leq C_w$ (Algorithm 2, step 8), $C_r$ is the upper bound on rewards (Assumption 5), Constant $G_\theta$ is defined in Lemma 7. $C_s = L_p^2 G_\theta^2 \max_t \frac{\gamma_t^2}{\alpha_t^2} + 4(C_r + 2C_w)^2$. $L_p$ is Lipchitz constant defined in Lemma 14.*

*Proof.*

$$\rho_{t+1} = \rho_t + \alpha_t \frac{1}{M} \sum_{i=0}^{M-1} \left( R^\pi(s_{t,i}) - \rho_t + \phi^\pi(s'_{t,i})^\mathsf{T} \bar{w}_t - \phi^\pi(s_{t,i})^\mathsf{T} \bar{w}_t \right)$$

$$\rho_{t+1} - \rho_{t+1}^* = \rho_t - \rho_t^* + \rho_t^* - \rho_{t+1}^*$$

$$+ \alpha_t \frac{1}{M} \sum_{i=0}^{M-1} \left( R^\pi(s_{t,i}) - \rho_t + \phi^\pi(s'_{t,i})^\mathsf{T} \bar{w}_t - \phi^\pi(s_{t,i})^\mathsf{T} \bar{w}_t \right)$$

$$= \rho_t - \rho_t^* + \rho_t^* - \rho_{t+1}^*$$

$$+ \alpha_t \frac{1}{M} \sum_{i=0}^{M-1} \left( R^\pi(s_{t,i}) - \rho_t^* + \phi^\pi(s'_{t,i})^\mathsf{T} \bar{w}_t - \phi^\pi(s_{t,i})^\mathsf{T} \bar{w}_t \right)$$

$$+ \alpha_t(\rho_t^* - \rho_t)$$

$$\rho_{t+1} - \rho_{t+1}^* = \rho_t - \rho_t^* + \rho_t^* - \rho_{t+1}^*$$

$$+ \alpha_t(\rho_t^* - \rho_t)$$

$$+ \alpha_t \left( \frac{1}{M} \sum_{i=0}^{M-1} (\phi^\pi(s'_{t,i}) - \phi^\pi(s_{t,i}))^\mathsf{T} (\bar{w}_t - w_t) \right)$$

$$+ \alpha_t \left( \frac{1}{M} \sum_{i=0}^{M-1} (R^\pi(s_{t,i}) - \rho_t^* + \phi^\pi(s'_{t,i})^\mathsf{T} w_t - \phi^\pi(s_{t,i})^\mathsf{T} w_t) \right)$$

$$= \rho_t - \rho_t^* + \rho_t^* - \rho_{t+1}^*$$

$$+ \alpha_t(\rho_t^* - \rho_t)$$

$$+ \alpha_t \left( \frac{1}{M} \sum_{i=0}^{M-1} (\phi^\pi(s'_{t,i}) - \phi^\pi(s_{t,i}))^\mathsf{T} (\bar{w}_t - w_t) \right)$$

$$+ \alpha_t \left( l(B_t, w_t, \theta_t) - \bar{l}(w_t, \theta_t) \right)$$

$$+ \alpha_t \left( \bar{l}(w_t, \theta_t) - \bar{l}(w_t^*, \theta_t) \right)$$

Here,

$$l(B_t, w_t, \theta_t) := \Big( \frac{1}{M} \sum_{i=0}^{M-1} (R^\pi(s_{t,i}) - \rho_t^* + \phi^\pi(s_{t,i}')^\intercal w_t - \phi^\pi(s_{t,i})^\intercal w_t) \Big)$$

$$\bar{l}(w_t, \theta_t) := \int_S d^\pi\big(s, \pi(\theta_t)\big)\big(R^\pi(s) - \rho(\pi(\theta_t)) + \phi^\pi(s')^\intercal w_t - \phi^\pi(s)^\intercal w_t\big)\, ds$$

$$\bar{l}(B_t, \rho_t, w_t, \theta_t) := (\rho_t^* - \rho_t)$$
$$+ \Big( \frac{1}{M} \sum_{i=0}^{M-1} (\phi^\pi(s_{t,i}') - \phi^\pi(s_{t,i}))^\intercal (\bar{w}_t - w_t) \Big)$$
$$+ \big( l(B_t, w_t, \theta_t) - \bar{l}(w_t, \theta_t) \big)$$
$$+ \big( \bar{l}(w_t, \theta_t) - \bar{l}(w_t^*, \theta_t) \big)$$

$$\begin{aligned}
||\Delta\rho_{t+1}||^2 &= ||\Delta\rho_t + \rho_t^* + \alpha_t l(B_t, w_t, \rho_t, \theta_t)||^2 \\
&= ||\Delta\rho_t||^2 + ||\rho_t^* - \rho_{t+1}^*||^2 + \alpha_t^2 ||l(B_t, \bar{w}_t, \rho_t, \theta_t)||^2 \\
&\quad + 2\langle \Delta\rho_t, \rho_t^* - \rho_{t+1}^* \rangle \\
&\quad + 2\alpha_t \langle \Delta\rho_t, l(B_t, \bar{w}_t, \rho_t, \theta_t) \rangle \\
&\quad + 2\alpha_t \langle \rho_t^* - \rho_{t+1}^*, l(B_t, \rho_t, \bar{w}_t, \theta_t) \rangle \\
&\leq ||\Delta\rho_t||^2 + 2||\rho_t^* - \rho_{t+1}^*||^2 + 2\alpha_t^2 ||l(B_t, \bar{w}_t, \rho_t, \theta_t)||^2 \\
&\quad + 2\langle \Delta\rho_t, \rho_t^* - \rho_{t+1}^* \rangle \\
&\quad + 2\alpha_t \langle \Delta\rho_t, l(B_t, \bar{w}_t, \rho_t, \theta_t) \rangle
\end{aligned}$$

$$\begin{aligned}
\mathbb{E}||\Delta\rho_{t+1}||^2 &\leq \mathbb{E}||\Delta\rho_t||^2 + 2\mathbb{E}||\rho_t^* - \rho_{t+1}^*||^2 \quad \text{①} \\
&\quad + 2\alpha_t^2 \mathbb{E}||l(B_t, \bar{w}_t, \rho_t, \theta_t)||^2 \quad \text{②} \\
&\quad + 2\mathbb{E}\langle \Delta\rho_t, \rho_t^* - \rho_{t+1}^* \rangle \quad \text{③} \\
&\quad + 2\alpha_t \mathbb{E}\langle \Delta\rho_{t1} - \Delta\rho_t \rangle \quad \text{④} \\
&\quad + 2\alpha_t \mathbb{E}\langle \Delta\rho_t, \frac{1}{M} \sum_{i=0}^{M-1} (\phi^\pi(s_{t,i}') - \phi^\pi(s_{t,i}))^\intercal (\bar{w}_t - w_t) \rangle \quad \text{⑤} \\
&\quad + 2\alpha_t \mathbb{E}\langle \Delta\rho_t, l(B_t, w_t, \theta_t) - \bar{l}(w_t, \theta_t) \rangle \quad \text{⑥} \\
&\quad + 2\alpha_t \mathbb{E}\langle \Delta\rho_t, \bar{l}(w_t, \theta_t) - \bar{l}(w_t^*, \theta_t) \rangle \quad \text{⑦}
\end{aligned}$$

(A.14)

From equation A.14:

①:
$$\mathbb{E}||\rho_t^* - \rho_{t+1}^*||^2 \leq L_p^2 \mathbb{E}||\theta_{t+1} - \theta_t||^2 \text{(Lemma 14)}$$

②:
$$\begin{aligned}
\mathbb{E}||l(B_t, \rho_t, \bar{w}_t, \theta_t)||^2 &= \mathbb{E}||\frac{1}{M} \sum_{i=0}^{M-1} \big( R^\pi(s_{t,i}) - \rho_t + \big(\phi^\pi(s_{t,i}') - \phi^\pi(s_{t,i})\big)^\intercal \bar{w}_t \big)||^2 \\
&\leq \mathbb{E}\Big( \frac{1}{M} \sum_{i=0}^{M-1} (C_r + C_r + 2C_w + 2C_w) \Big)^2 \\
&= 4(C_r + 2C_2)^2
\end{aligned}$$

③:
$$\begin{aligned}
\mathbb{E}\langle \Delta\rho_t, \rho_t^* - \rho_{t+1}^* \rangle &\leq \mathbb{E}||\Delta\rho_t||\,|\rho_t^* - \rho_{t+1}^*| \\
&\leq L_p \mathbb{E}|\Delta\rho_t|\,||\theta_{t+1} - \theta_t||
\end{aligned}$$

④:
$$\mathbb{E}\langle \Delta\rho_t, -\Delta\rho_t \rangle = -\mathbb{E}|\Delta\rho_t|^2$$

⑤:

$$\mathbb{E}\langle\Delta\rho_t, \frac{1}{M}\sum_{i=0}^{M-1}\big(\phi^\pi(s'_{t,i})^\intercal - \phi^\pi(s_{t,i})^\intercal\big)(\bar{w}_t - w_t)\rangle$$

$$\leq \mathbb{E}\Big[\frac{1}{M}\sum_{i=0}^{M-1}||\phi^\pi(s'_{t,i}) - \phi^\pi(s_{t,i})||\,||\bar{w}_t - w_t||\,|\Delta\rho_t|\Big]$$

$$\leq 4C_w\mathbb{E}|\Delta\rho_t|$$

⑥:

$$\mathbb{E}\langle\Delta\rho_t, l(B_t, w_t, \theta_t) - \bar{l}(w_t, \theta_t)\rangle = \mathbb{E}\langle\Delta\rho_t, \mathbb{E}[l(B_t, w_t, \theta_t) - \bar{l}(w_t, \theta_t)|\Delta\rho_t]\rangle$$

Note: $\mathbb{E}[l(B_t, w_t, \theta_t) - \bar{l}(w_t, \theta_t)|\Delta\rho_t]$

$$\mathbb{E}\langle\Delta\rho_t, l(B_t, w_t, \theta_t) - \bar{l}(w_t, \theta_t)\rangle = 0$$

⑦:

$$\mathbb{E}[\langle\Delta\rho_t, \bar{l}(w_t, \theta_t) - \bar{l}(w_t^*, \theta_t)\rangle] = \mathbb{E}[\langle\Delta\rho_t, \big(\mathbb{E}[\phi^\pi(s')] - \phi^\pi(s)\big)^\intercal(w_t - w_t^*)\rangle]$$

$$\leq \mathbb{E}[\langle\phi^\pi(s') - \phi^\pi(s), \Delta w_t\rangle|\Delta\rho_t|]$$

$$\leq \mathbb{E}[||\phi^\pi(s') - \phi^\pi(s)||\,||\Delta w_t||\,|\Delta\rho_t|]$$

$$\leq 2\mathbb{E}(||\Delta w_t||\,|\Delta\rho_t|)$$

Combining ①-⑦ into equation A.14:

$$\mathbb{E}||\Delta\rho_{t+1}||^2 \leq (1 - 2\alpha_t)\mathbb{E}||\Delta\rho_t||^2 + 2L_p^2\mathbb{E}||\theta_{t+1} - \theta_t||^2$$
$$+ 8\alpha_t^2(C_r + 2C_w)^2 + 2L_p\mathbb{E}|\Delta\rho_t|\,||\theta_{t+1} - \theta_t||$$
$$+ 8\alpha_t C_w\mathbb{E}|\Delta\rho_t| + 4\alpha_t\mathbb{E}||\Delta w_t||\,|\Delta\rho_t|$$

$$\begin{aligned}\Longrightarrow \sum_{t=0}^{T-1}\mathbb{E}||\Delta\rho_t||^2 \leq{}& \sum_{t=0}^{T-1}\frac{1}{2\alpha_t}\Big(\mathbb{E}||\Delta\rho_t||^2 - \mathbb{E}||\Delta\rho_{t+1}||^2\Big)\ \text{①}\\
&+ \sum_{t=0}^{T-1}\Big(\frac{L_p^2\gamma_t^2}{\alpha_t}G_\theta^2 + 4\alpha_t(C_r + 2C_w)^2\Big)\ \text{②}\\
&+ \sum_{t=0}^{T-1}\Big(L_p\frac{\gamma_t}{\alpha_t}G_\theta + 4C_w\Big)\mathbb{E}|\Delta\rho_t|\ \text{③}\\
&+ \sum_{t=0}^{T-1}2\mathbb{E}||\Delta w_t||\,|\Delta\rho_t|\ \text{④}\end{aligned}$$

(A.15)

From equation A.15:
①:

$$\begin{aligned}\frac{1}{2}\sum_{t=0}^{T-1}\frac{1}{\alpha_t}(\mathbb{E}||\Delta\rho_t||^2 - \mathbb{E}||\Delta\rho_{t+1}||^2) ={}& \frac{1}{2}\Bigg(\sum_{t=0}^{T-1}\Big(\frac{1}{\alpha_t} - \frac{1}{\alpha_{t-1}}\Big)\mathbb{E}|\Delta\rho_t|^2 + \frac{1}{\alpha_0}\mathbb{E}|\Delta\rho_0|^2 - \frac{1}{\alpha_{T-1}}\mathbb{E}|\Delta\rho_t|^2\Bigg)\\
&\leq \frac{1}{2}\Bigg(\sum_{t=0}^{T-1}\Big(\frac{1}{\alpha_t} - \frac{1}{\alpha_{t-1}}\Big) + \frac{1}{\alpha_0}\Bigg)4(C_r + 2C_w)^2\\
&\leq \frac{2(C_r + 2C_2)^2}{C_\alpha}T^\sigma\end{aligned}$$

②:

$$\sum_{t=0}^{T-1}\Big(L_p^2G_\theta^2\frac{\gamma_t^2}{\alpha_t}+4\alpha_t(C_r+2C_w)^2\Big)\le\sum_{t=0}^{T-1}\Big(L_p^2G_\theta^2\max_t\frac{\gamma_t^2}{\alpha_t^2}+4(C_r+2C_w)^2\Big)\alpha_t$$

$$\le\sum_{t=0}^{T-1}C_s\alpha_t\quad(C_s=L_p^2G_\theta^2\max_t\frac{\gamma_t^2}{\alpha_t^2}+4(C_r+2C_w)^2)$$

$$\le\frac{C_sC_\alpha}{1-\sigma}T^{1-\sigma}$$

③:

$$\sum_{t=0}^{T-1}\Big(L_pG_\theta\frac{\gamma_t}{\alpha_t}+4C_w\Big)\mathbb{E}||\Delta\rho_t||=\sum_{t=0}^{T-1}L_pG_\theta\frac{\gamma_t}{\alpha_t}\mathbb{E}||\Delta\rho_t||+4C_w\sum_{t=0}^{T-1}\mathbb{E}||\Delta\rho_t||$$

$$\le L_pG_\theta\bigg(\sum_{t=0}^{T-1}\Big(\frac{\gamma_t}{\alpha_t}\Big)^2\bigg)^{\frac{1}{2}}\bigg(\sum_{t=0}^{T-1}\mathbb{E}|\Delta\rho_t|^2\bigg)^{\frac{1}{2}}+4C_w\bigg(\sum_{t=0}^{T-1}\mathbb{E}|\Delta\rho_t|^2\bigg)^{\frac{1}{2}}T^{\frac{1}{2}}$$

(using cauchy schwarz inequality)

④:

$$2\sum_{t-0}^{T-1}\mathbb{E}||\Delta w_t||\,|\Delta\rho_t|\le2(\sum_{t=0}^{T-1}\mathbb{E}||\Delta w_t||^2)^{\frac{1}{2}}(\sum_{t=0}^{T-1}\mathbb{E}|\Delta\rho_t|^2)^{\frac{1}{2}}$$

(using cauchy schwarz inequality)

Combining ①-④ into equation A.15

$$\frac{1}{T}\sum_{t=0}^{T-1}\mathbb{E}||\Delta\rho_t||^2\le\frac{2(C_r+2C_w)^2T^{\sigma-1}}{C_\alpha}+\frac{C_sC_\alpha T^{-\sigma}}{1-\sigma}$$

$$+L_pG_\theta\bigg(\frac{1}{T}\sum_{t=0}^{T-1}\Big(\frac{\gamma_t}{\alpha_t}\Big)^2\bigg)^{\frac{1}{2}}\bigg(\frac{1}{T}\sum_{t=0}^{T-1}\mathbb{E}|\Delta\rho_t|^2\bigg)^{\frac{1}{2}}$$

$$+4C_w\bigg(\frac{1}{T}\sum_{t=0}T-1\mathbb{E}|\Delta\rho_t|^2\bigg)^{\frac{1}{2}}$$

$$+2\bigg(\frac{1}{T}\sum_{t=0}^{T-1}\mathbb{E}||\Delta w_t||^2\bigg)^{\frac{1}{2}}\bigg(\frac{1}{T}\sum_{t=0}^{T-1}\mathbb{E}|\Delta\rho_t|^2\bigg)^{\frac{1}{2}}$$

$$M(T)=\frac{1}{T}\sum_{t=0}^{T-1}\mathbb{E}||\Delta\rho_t||^2$$

$$N(T)=\frac{1}{T}\sum_{t=0}^{T-1}\mathbb{E}||\Delta w_t||^2$$

$$M(T)\le K_1+K_2\sqrt{M(T)}+K_3\sqrt{M(T)}\sqrt{N(T)}$$

Here,

$$K_1=\frac{2(C_r+2C_2)^2T^{\sigma-1}}{C_\alpha}+\frac{C_sC_\alpha T^{-\sigma}}{1-\sigma}$$

$$K_2=L_pG_\theta\bigg(\frac{1}{T}\sum_{t=0}^{T-1}\Big(\frac{\gamma_t}{\alpha_t}\Big)^2\bigg)^{\frac{1}{2}}+4C_w$$

$$K_3=2$$

From Lemma 4, we know that

$$M(T) \leq 2(\sqrt{K_1} + K_2)^2 + 2K_3^2 N(T)$$

Hence,

$$\frac{1}{T} \sum_{t=0}^{T-1} \mathbb{E}|\Delta \rho_t|^2 \leq 2\Big(\sqrt{\frac{2(C_r + 2C_w)^2}{C_\alpha}T^{\sigma-1} + \frac{C_s C_\alpha}{1-\sigma}T^{-\sigma}}$$

$$+ L_p G_\theta \Big(\frac{1}{T} \sum_{t=0}^{T-1} \Big(\frac{\gamma_t}{\alpha_t}\Big)^2\Big)^{\frac{1}{2}} + 4C_w\Big)^2$$

$$+ 8\frac{1}{T} \sum_{t=0}^{T-1} \mathbb{E}||\Delta w_t||^2$$

$$\square$$

**Theorem 3.** *The on-policy average reward actor critic algorithm obtain $\epsilon$-accurate optimal point with sample complexity of $\Omega(\epsilon^{-2.5})$.*

$$\min_{0 \leq t \leq T-1} E||\nabla_\theta \rho(\theta_t)||^2 = \mathcal{O}\Big(\frac{1}{T^{0.4}}\Big) + \mathcal{O}(1)$$

$$\min_{0 \leq t \leq T-1} E||\nabla_\theta \rho(\theta_t)||^2 \leq \epsilon + \mathcal{O}(1)$$

*Proof.* Using lemma 4 and lemma 5 we obtain,

$$\frac{1}{T} \sum_{t=0}^{T-1} E||\Delta w_t||^2 \leq 2\Big(\sqrt{\frac{2C_w^2}{\lambda C_\alpha}T^{\sigma-1} + \frac{C_g C_\alpha}{1-\sigma}T^{-\sigma}} + \frac{L_w G_\theta}{\lambda}\Big(\frac{1}{T} \sum_{t=0}^{T-1}(\frac{\gamma_t}{\alpha_t})^2\Big)^{1/2} + \frac{2(C_r + 3C_w)}{\lambda}\Big)^2$$

$$+ \frac{4}{\lambda^2}\Big(\sqrt{\frac{2(C_r + 2C_w)^2}{C_\alpha}T^{\sigma-1} + \frac{C_s C_\alpha}{1-\sigma}T^{-\sigma}} + L_p G_\theta\Big(\frac{1}{T} \sum_{t=0}^{T-1}(\frac{\gamma_t}{\alpha_t})^2\Big)^{1/2} + 4C_w\Big)^2$$

$$+ \frac{16}{\lambda^2 T} \sum_{t=0}^{T-1} E||\Delta w_t||^2$$

$$\implies \frac{1}{T} \sum_{t=0}^{T-1} E||\Delta w_t||^2 \leq \frac{2\lambda^2}{\lambda^2 - 16}\Big(\sqrt{\frac{2C_w^2}{\lambda C_\alpha}T^{\sigma-1} + \frac{C_g C_\alpha}{1-\sigma}T^{-\sigma}}$$

$$+ \frac{L_w G_\theta}{\lambda}\Big(\frac{1}{T} \sum_{t=0}^{T-1}(\frac{\gamma_t}{\alpha_t})^2\Big)^{1/2} + \frac{2(C_r + 3C_w)}{\lambda}\Big)^2 \quad \text{①}$$

$$+ \frac{4}{\lambda^2 - 16}\Big(\sqrt{\frac{2(C_r + 2C_w)^2}{C_\alpha}T^{\sigma-1} + \frac{C_s C_\alpha}{1-\sigma}T^{-\sigma}}$$

$$+ L_p G_\theta\Big(\frac{1}{T} \sum_{t=0}^{T-1}(\frac{\gamma_t}{\alpha_t})^2\Big)^{1/2} + 4C_w\Big)^2 \quad \text{②}$$

$$(A.16)$$

From equation A.16

①:

$$\frac{1}{T} \sum_{t=0}^{T-1}(\frac{\gamma_t}{\alpha_t})^2 \leq \frac{1}{T} \sum_{t=0}^{T-1}(\frac{C_\gamma}{C_\alpha})^2 \frac{1}{(1+t)^{2(v-\sigma)}} \leq \frac{T^{-2(v-\sigma)}}{1 - 2(v-\sigma)}$$

$$\Big(\because \sum_{t=0}^{T-1} \frac{1}{1+t^v} \leq \int_0^T \frac{1}{t^v}\, dt = \frac{T^{1-v}}{1-v}\Big)$$

$$\left( \sqrt{\frac{2C_w^2}{\lambda C_\alpha}T^{\sigma-1} + \frac{C_g C_\alpha}{1-\sigma}T^{-\sigma}} + \frac{L_w G_\theta}{\lambda}\left(\frac{1}{T}\sum_{t=0}^{T-1}(\frac{\gamma_t}{\alpha_t})^2\right)^{1/2} + \frac{2(C_r + 3C_w)}{\lambda}\right)^2$$

$$\leq \left( \sqrt{\frac{2C_w^2}{\lambda C_\alpha}T^{\sigma-1} + \frac{C_g C_\alpha}{1-\sigma}T^{-\sigma}} + \frac{L_w G_\theta}{\lambda}\left(\frac{T^{-2(v-\sigma)}}{1-2(v-\sigma)}\right)^{1/2} + \frac{2(C_r + 3C_w)}{\lambda}\right)^2$$

$$\leq 3\left( \frac{2C_w^2}{\lambda C_\alpha}T^{\sigma-1} + \frac{C_g C_\alpha}{1-\sigma}T^{-\sigma} + \left(\frac{L_w G_\theta}{\lambda}\right)^2\left(\frac{T^{-2(v-\sigma)}}{1-2(v-\sigma)}\right) + \left(\frac{2(C_r + 3C_w)}{\lambda}\right)^2\right)$$

$$(\because (a+b+c)^2 \leq 3(a^2 + b^2 + c^2))$$

$$= \mathcal{O}\left(\frac{1}{T^{1-\sigma}}\right) + \mathcal{O}\left(\frac{1}{T^\sigma}\right) + \mathcal{O}\left(\frac{1}{T^{2(v-\sigma)}}\right) + \mathcal{O}(1)$$

②  (similar to ①):

$$\left( \sqrt{\frac{2(C_r + 2C_w)^2}{C_\alpha}T^{\sigma-1} + \frac{C_s C_\alpha}{1-\sigma}T^{-\sigma}} + L_p G_\theta\left(\frac{1}{T}\sum_{t=0}^{T-1}(\frac{\gamma_t}{\alpha_t})^2\right)^{1/2} + 4C_w\right)^2$$

$$= \mathcal{O}\left(\frac{1}{T^{1-\sigma}}\right) + \mathcal{O}\left(\frac{1}{T^\sigma}\right) + \mathcal{O}\left(\frac{1}{T^{2(v-\sigma)}}\right) + \mathcal{O}(1)$$

Combining ① and ②:

$$\frac{1}{T}\sum_{t=0}^{T-1} E||\Delta w_t||^2 = \mathcal{O}\left(\frac{1}{T^{1-\sigma}}\right) + \mathcal{O}\left(\frac{1}{T^\sigma}\right) + \mathcal{O}\left(\frac{1}{T^{2(v-\sigma)}}\right) + \mathcal{O}(1) \qquad (A.17)$$

Using lemma 3 and equation A.17

$$\frac{1}{T}\sum_{t=0}^{T-1} E||\nabla_\theta \rho(\theta_t)||^2 = \mathcal{O}\left(\frac{1}{T^{1-v}}\right) + \mathcal{O}\left(\frac{1}{T^v}\right) + \mathcal{O}\left(\frac{1}{T^{1-\sigma}}\right) + \mathcal{O}\left(\frac{1}{T^\sigma}\right)$$

$$+ \mathcal{O}\left(\frac{1}{T^{2(v-\sigma)}}\right) + \mathcal{O}(1)$$

$$\implies \min_{0\leq t\leq T-1} E||\nabla_\theta \rho(\theta_t)||^2 = \mathcal{O}\left(\frac{1}{T^{1-v}}\right) + \mathcal{O}\left(\frac{1}{T^v}\right) + \mathcal{O}\left(\frac{1}{T^{1-\sigma}}\right) + \mathcal{O}\left(\frac{1}{T^\sigma}\right)$$

$$+ \mathcal{O}\left(\frac{1}{T^{2(v-\sigma)}}\right) + \mathcal{O}(1)$$

$$\left(\because \min_t E||\nabla_\theta \rho(\theta_t)||^2 \leq \frac{1}{T}\sum_{t=0}^{T-1} E||\nabla_\theta \rho(\theta_t)||^2\right)$$

By setting $v = 3/5$ and $\sigma = 2/5$, we obtain:

$$\min_{0\leq t\leq T-1} E||\nabla_\theta \rho(\theta_t)||^2 = \mathcal{O}\left(\frac{1}{T^{0.4}}\right) + \mathcal{O}(1)$$

$$\mathcal{O}\left(\frac{1}{T^{0.4}}\right) \leq \epsilon$$

Hence, the sample complexity of on-policy average reward actor-critic algorithm is $\Omega(\epsilon^{-2.5})$. □

**Lemma 6.** *The optimal critic parameter* $w(\theta_t)^*$ *as a function of actor parameter* $\theta_t$ *is Lipchitz continuous with constant* $L_w$. *Note:* $w_t^* := w(\theta_t)^*$.

$$||w_t^* - w_{t+1}^*|| \leq L_w ||\theta_{t+1} - \theta_t||$$

*Proof.* $\eta$ is the l2-regularisation coefficient from Algorithm 2 and $\eta > \lambda_{max}^{all}$, where $\lambda_{max}^{all}$ is defined in Lemma 11. Because of carefully setting the value of $\eta$, $A(\theta_t)$ is negative definite. Thus, for on-policy TD(0) with l2-regularization and target estimators, the following condition holds true for optimal critic parameter $w_t^*$:

$$E[(R^\pi(s) - \rho_t^*)\phi^\pi(s) + (\phi^\pi(s)(E[\phi^\pi(s')] - \phi^\pi(s))^\intercal - \eta I)w_t^*] = 0$$

$$b(\theta_t) := E[(R^\pi(s) - \rho_t^*)\phi^\pi(s)]$$
$$A(\theta_t) := E[(\phi^\pi(s)(E[\phi^\pi(s')] - \phi^\pi(s))^\intercal - \eta I)]$$

$$\therefore b(\theta_t) + A(\theta_t)w_t^* = 0 \implies w_t^* = -A(\theta_t)^{-1}b(\theta_t)$$

$$
\begin{aligned}
||w_t^* - w_{t+1}^*|| &= ||A(\theta_t)^{-1}b(\theta_t) - A(\theta_{t+1})^{-1}b(\theta_{t+1})|| \\
&\leq ||A(\theta_t)^{-1}b(\theta_t) - A(\theta_{t+1})^{-1}b(\theta_t) + A(\theta_{t+1})^{-1}b(\theta_t) - A(\theta_{t+1})^{-1}b(\theta_{t+1})|| \\
&\leq ||A(\theta_t)^{-1} - A(\theta_{t+1})^{-1}|| \, ||b(\theta_t)|| \quad ① \\
&\quad + ||A(\theta_{t+1})^{-1}|| \, ||b(\theta_t) - b(\theta_{t+1})|| ②
\end{aligned}
$$
(A.18)

From equation A.18:

①:

$$
\begin{aligned}
||A(\theta_t)^{-1} - A(\theta_{t+1})^{-1}|| &= ||A(\theta_t)^{-1}A(\theta_{t+1})A(\theta_{t+1})^{-1} - A(\theta_t)^{-1}A(\theta_t)^{A}(\theta_{t+1})^{-1}|| \\
&\leq ||A(\theta_t)^{-1}|| \, ||A(\theta_t) - A(\theta_{t+1})|| \, ||A(\theta_{t+1})^{-1}||
\end{aligned}
$$
(A.19)

From equation A.19:

Here, $\pi'$ and $\pi$ represents the policy with parameter $\theta_{t+1}$ and $\theta_t$ respectively.

$$
\begin{aligned}
||A(\theta_t) - A(\theta_{t+1})|| &\leq || \int d^{\pi'}(s)(\phi^{\pi'}(s)(\int P^{\pi'}(s'|s)\phi^{\pi'}(s')\,ds' - \phi^{\pi'}(s))^\intercal - \eta I)\,ds \\
&\quad - \int d^\pi(s)(\phi^\pi(s)(\int P^\pi(s'|s)\phi^\pi(s')\,ds' - \phi^\pi(s))^\intercal - \eta I)\,ds|| \\
&\leq || \int d^{\pi'}(s)(\phi^{\pi'}(s)(\int P^{\pi'}(s'|s)\phi^{\pi'}(s')\,ds')^\intercal)\,ds \\
&\quad - \int d^\pi(s)(\phi^\pi(s)(\int P^\pi(s'|s)\phi^\pi(s')\,ds')^\intercal)\,ds|| \quad ① \\
&\leq || \int d^\pi(s)(\phi^\pi(s)(\phi^\pi(s))^\intercal)\,ds - \int d^{\pi'}(s)(\phi^{\pi'}(s)(\phi^{\pi'}(s))^\intercal)\,ds|| \quad ②
\end{aligned}
$$
(A.20)

From equation A.20:

①:

$$||\int d^{\pi'}(s)(\phi^{\pi'}(s)(\int P^{\pi'}(s'|s)\phi^{\pi'}(s')\,ds')^\intercal)\,ds - \int d^{\pi}(s)(\phi^{\pi}(s)(\int P^{\pi}(s'|s)\phi^{\pi}(s')\,ds')^\intercal)\,ds||$$

$$\leq ||\int (d^{\pi'}(s) - d^{\pi}(s))\phi^{\pi'}(s)(\int P^{\pi'}(s'|s)\phi^{\pi'}(s')\,ds')^\intercal\,ds||$$

$$+ ||\int d^{\pi}(s)(\phi^{\pi'}(s) - \phi^{\pi}(s))(\int P^{\pi'}(s'|s)\phi^{\pi'}(s')\,ds')^\intercal\,ds||$$

$$+ ||\int d^{\pi}(s)\phi^{\pi}(s)(\int (P^{\pi'}(s'|s) - P^{\pi}(s'|s))\phi^{\pi'}(s')\,ds')^\intercal\,ds||$$

$$+ ||\int d^{\pi}(s)\phi^{\pi}(s)(\int P^{\pi}(s'|s)(\phi^{\pi'}(s') - \phi^{\pi}(s'))\,ds'^\intercal)\,ds$$

$$\leq L_d||\theta_{t+1} - \theta_t|| \qquad \text{(lemma 12)}$$
$$+ L_\phi||\theta_{t+1} - \theta_t|| \qquad \text{(assumption 8)}$$
$$+ L_t||\theta_{t+1} - \theta_t|| \qquad \text{(assumption 9)}$$
$$+ L_\phi||\theta_{t+1} - \theta_t|| \qquad \text{(assumption 8)}$$

$$||\int d^{\pi'}(s)(\phi^{\pi'}(s)(\int P^{\pi'}(s'|s)\phi^{\pi'}(s')\,ds')^\intercal)\,ds - \int d^{\pi}(s)(\phi^{\pi}(s)(\int P^{\pi}(s'|s)\phi^{\pi}(s')\,ds')^\intercal)\,ds||$$
$$\leq (L_d + L_t + 2L_\phi)||\theta_{t+1} - \theta_t||$$

$$(A.21)$$

From equation A.20:

②:

$$||\int d^{\pi}(s)(\phi^{\pi}(s)(\phi^{\pi}(s))^\intercal)\,ds - \int d^{\pi'}(s)(\phi^{\pi'}(s)(\phi^{\pi'}(s))^\intercal)\,ds||$$

$$\leq ||\int (d^{\pi}(s) - d^{\pi'}(s))\phi^{\pi}(s)(\phi^{\pi}(s))^\intercal\,ds||$$

$$+ ||\int d^{\pi'}(s)(\phi^{\pi}(s) - \phi^{\pi'}(s))(\phi^{\pi}(s))^\intercal\,ds|| \qquad (A.22)$$

$$+ ||\int d^{\pi'}(s)\phi^{\pi'}(s)(\phi^{\pi}(s) - \phi^{\pi'}(s))^\intercal\,ds||$$

$$\leq (L_d + 2L_\phi)||\theta_{t+1} - \theta_t||$$

Using equation A.21 and equation A.22 in equation A.20

$$||A(\theta_t) - A(\theta_{t+1})|| \leq (2L_d + 4L_\phi + L_t)||\theta_{t+1} - \theta_t|| \qquad (A.23)$$

From equation A.18:

②:

$$||b(\theta_t) - b(\theta_{t+1})|| = ||\int d^{\pi'}(s)((R^{\pi'}(s) - \rho_{t+1}^*)\phi^{\pi'}(s)\,ds - \int d^{\pi}(s)(R^{\pi}(s) - \rho_t^*)\phi^{\pi}(s)\,ds||$$

$$\leq ||\int d^{\pi'}(s)(R^{\pi'}(s)\phi^{\pi'}(s)\,ds - \int d^{\pi}(s)R^{\pi}(s)\phi^{\pi}(s)\,ds||$$

$$+ ||\int d^{\pi'}(s)\rho_{t+1}^*\phi^{\pi'}(s)\,ds - \int d^{\pi}(s)\rho_t^*\phi^{\pi}(s)\,ds||$$

$$\leq ||\int (d^{\pi'}(s) - d^{\pi}(s))R^{\pi'}(s)\phi^{\pi'}(s)\,ds||$$

$$+ ||\int d^{\pi}(s)(R^{\pi'}(s) - R^{\pi}(s))\phi^{\pi'}(s)\,ds||$$

$$+ ||\int d^{\pi}(s)R^{\pi}(s)(\phi^{\pi'}(s) - \phi^{\pi}(s)\,ds||$$

$$+ ||\int (d^{\pi'}(s) - d^{\pi}(s))\rho_{t+1}^*\phi^{\pi'}(s)\,ds||$$

$$+ ||\int d^{\pi}(s)(\rho_{t+1}^* - \rho_t^*)\phi^{\pi'}(s)\,ds||$$

$$+ ||\int d^{\pi}(s)\rho_t^*(\phi^{\pi'}(s) - \phi^{\pi}(s)\,ds||$$

$$\leq C_r L_d||\theta_{t+1} - \theta_t|| \quad \text{( Assumption 5, Lemma 12)}$$
$$+ L_r||\theta_{t+1} - \theta_t|| \quad \text{( Assumption 10)}$$
$$+ C_r L_\phi||\theta_{t+1} - \theta_t|| \quad \text{(Assumption 5,Assumption 8)}$$
$$+ C_r L_d||\theta_{t+1} - \theta_t|| \quad \text{( Assumption 5, Lemma 12)}$$
$$+ L_p||\theta_{t+1} - \theta_t|| \quad \text{(Lemma 14)}$$
$$+ C_r L_\phi||\theta_{t+1} - \theta_t|| \quad \text{(Assumption 5,Assumption 8)}$$

$$\implies ||b(\theta_t) - b(\theta_{t+1})|| \leq (2L_d C_r + 2C_r L_\phi + L_r + L_p)||\theta_{t+1} - \theta_t|| \qquad \text{(A.24)}$$

Using equation A.19, equation A.23 and equation A.24 in equation A.18:

$$||w_t^* - w_{t+1}^*|| \leq ||A(\theta_t)^{-1} - A(\theta_{t+1})^{-1}|| \, ||b(\theta_t)|| + ||A(\theta_{t+1})^{-1}|| \, ||b(\theta_t) - b(\theta_{t+1})||$$
$$\leq ||A(\theta_t)^{-1}|| \, ||A(\theta_t) - A(\theta_{t+1})|| \, ||A(\theta_{t+1})^{-1}|| \, ||b(\theta_t)||$$
$$+ ||A(\theta_{t+1})^{-1}|| \, ||b(\theta_t) - b(\theta_{t+1})||$$
$$\leq (2L_d + 4L_\phi + L_t)||A(\theta_t)^{-1}|| \, ||A(\theta_{t+1})^{-1}|| \, ||b(\theta_t)|| \, ||\theta_{t+1} - \theta_t||$$
$$+ (2L_d C_r + 2C_r L_\phi + L_r + L_p)||A(\theta_{t+1})^{-1}|| \, ||\theta_{t+1} - \theta_t||$$

Note:

- $||b(\theta_t)|| = ||\int d^{\pi}(s)(\phi^{\pi}(s)(\phi^{\pi}(s))^{\intercal})\,ds|| \leq C_r$ (Using Assumption 5)

- From Assumption 12, $\lambda_{min}$ is the lower bound on eigen values of $A(\theta)$ for all $\theta$.

$$\therefore ||w_t^* - w_{t+1}^*|| \leq \frac{C_r(2L_d + 4L_\phi + L_t)}{\lambda_{min}^2}||\theta_{t+1} - \theta_t||$$
$$+ \frac{(2L_d C_r + 2C_r L_\phi + L_r + L_p)}{\lambda_{min}}||\theta_{t+1} - \theta_t||$$
$$\leq L_w||\theta_{t+1} - \theta_t||$$

where,

$$L_w = \frac{C_r(2L_d + 4L_\phi + L_t)}{\lambda_{min}^2} + \frac{(2L_d C_r + 2C_r L_\phi + L_r + L_p)}{\lambda_{min}}$$

$\square$

**Lemma 7.** *$Q_{diff}^w$ is the approximate differential Q-value function parameterized by $w$. Then there exist a constant $G_\theta$, independent of policy parameter $\theta$, such that:*

$$||\frac{1}{M}\sum_{i=0}^{M-1}\nabla_a Q_{diff}^w(s,a)|_{a=\pi(s)}\nabla_\theta \pi(s)|| \le G_\theta$$

*Proof.*

$$||Q_{diff}^w(s,a_1) - Q_{diff}^w(s,a_2)|| \le L_a||a_1 - a_2|| \quad \text{(Assumption 6)}$$
$$\implies ||\nabla_a Q_{diff}^w(s,a)|| \le L_a \tag{A.25}$$
$$\implies ||\nabla_a Q_{diff}^w(s,a)|_{a=\pi(s)}|| \le L_a$$

$$||\pi(s,\theta_1) - \pi(s,\theta_2)|| \le L_\pi||\theta_1 - \theta_2|| \quad \text{(Assumption 7)}$$
$$\implies ||\nabla_\theta \pi(s)|| \le L_\pi \tag{A.26}$$

Using equation A.25 and equation A.26:

$$||\frac{1}{M}\sum_{i=0}^{M-1}\nabla_a Q_{diff}^w(s,a)|_{a=\pi(s)}\nabla_\theta \pi(s)\nabla_\theta \pi(s)||$$

$$\le \frac{1}{M}\sum_{i=0}^{M-1}||\nabla_a Q_{diff}^w(s,a)|_{a=\pi(s)}\nabla_\theta \pi(s)\nabla_\theta \pi(s)||$$

$$\le L_a L_\pi = G_\theta$$

$\square$

**Lemma 8.** *The average reward estimate $\rho_t$ is bounded.*
$$\forall t > 0 \quad |\rho_t| \le C_r + 2C_w$$

*Here, $C_w$ is the upper bound on critic parameter $w_t$ (Algorithm 2, step 8), $C_r$ is the upper bound on rewards (Assumption 5).*

*Proof.*
$$|\rho_0| \le C_r + 2C_w \quad \text{(Assumption 11)}$$

For t = 1:

$$\rho_1 = \rho_0 + \alpha_0\Big(\frac{1}{M}\sum_{i=0}^{M-1}R_\pi(s_{0,i}) + \phi^\pi(s'_{0,i})^\intercal\bar{w}_0 - \phi^\pi(s_{0,i})^\intercal\bar{w}_0 - \rho_0\Big)$$

$$= (1 - \alpha_0)\rho_0 + \alpha_0\Big(\frac{1}{M}\sum_{i=0}^{M-1}R^\pi(s_{0,i}) + \phi^\pi(s'_{0,i})^\intercal\bar{w}_0 - \phi^\pi(s_{0,i})^\intercal\bar{w}_0\Big)$$

$$|\rho_1| \le (1 - \alpha_0)|\rho_0| + \alpha_0||\Big(\frac{1}{M}\sum_{i=0}^{M-1}R^\pi(s_{0,i}) + \phi^\pi(s'_{0,i})^\intercal\bar{w}_0 - \phi^\pi(s_{0,i})^\intercal\bar{w}_0\Big)||$$

$$\le (1 - \alpha_0)|\rho_0| + \alpha_0\Big(\frac{1}{M}\sum_{i=0}^{M-1}|R^\pi(s_{0,i})| + ||\phi^\pi(s'_{0,i})|| \, ||\bar{w}_0|| + ||\phi^\pi(s_{0,i})|| \, ||\bar{w}_0||\Big)$$

$$\le (1 - \alpha_0)(C_r + 2C_w) + (\alpha_0)(C_r + 2C_w) = (C_r + 2C_w) \quad \text{(Assumption 11)}$$

Therefore the bound hold for t = 1.
Let the bound hold for t = k. We will prove that the bound will also hold for k+1

$$\rho_{k+1} = \rho_k + \alpha_k \Big( \frac{1}{M} \sum_{i=0}^{M-1} R_\pi(s_{k,i}) + \phi^\pi(s'_{k,i})^\intercal \bar{w}_k - \phi^\pi(s_{k,i})^\intercal \bar{w}_k - \rho_k \Big)$$

$$= (1 - \alpha_k)\rho_k + \alpha_k \Big( \frac{1}{M} \sum_{i=0}^{M-1} R^\pi(s_{k,i}) + \phi^\pi(s'_{k,i})^\intercal \bar{w}_k - \phi^\pi(s_{k,i})^\intercal \bar{w}_k \Big)$$

$$|\rho_{k+1}| \le (1 - \alpha_k)|\rho_k| + \alpha_k || \Big( \frac{1}{M} \sum_{i=0}^{M-1} R^\pi(s_{k,i}) + \phi^\pi(s'_{k,i})^\intercal \bar{w}_k - \phi^\pi(s_{k,i})^\intercal \bar{w}_k \Big) ||$$

$$\le (1 - \alpha_k)|\rho_k| + \alpha_k \Big( \frac{1}{M} \sum_{i=0}^{M-1} |R^\pi(s_{k,i})| + ||\phi^\pi(s'_{k,i})|| \, ||\bar{w}_k|| + ||\phi^\pi(s_{k,i})|| \, ||\bar{w}_k|| \Big)$$

$$\le (1 - \alpha_k)(C_r + 2C_w) + (\alpha_k)(C_r + 2C_w) = (C_r + 2C_w)$$

The bound hold for t = k+1 as well. Hence by the principle of mathematical induction :

$$\forall t > 0 \ \ |\rho_t| \le C_r + 2C_w$$

$\square$

**Lemma 9.** *The norm of target critic estimator $\bar{w}_t$ is bounded*
$$\forall t > 0 \ \ ||\bar{w}_t|| \le C_w$$

*Here, $C_w$ is the upper bound on critic parameter $w_t$ (Algorithm 2, step 8).*

*Proof.* For t=1:
$$\bar{w}_1 = (1 - \beta_0)\bar{w}_0 + \beta_0 w_1$$
$$||\bar{w}_1|| \le (1 - \beta_0)||\bar{w}_0|| + \beta_0 ||w_1||$$
$$||\bar{w}_1|| \le (1 - \beta_0)C_w + \beta_0 C_w \quad \text{(Assumption 11)}$$
$$||\bar{w}_1|| \le C_w$$

The bound hold for t=1.
Let the bound hold for t = k. We will prove that the bound will also hold for k+1

$$\bar{w}_{k+1} = (1 - \beta_k)\bar{w}_k + \beta_k w_{k+1}$$
$$||\bar{w}_{k+1}|| \le (1 - \beta_k)||\bar{w}_k|| + \beta_k ||w_{k+1}||$$
$$||\bar{w}_{k+1}|| \le (1 - \beta_k)C_w + \beta_k C_w \quad \text{(Assumption 11)}$$
$$||\bar{w}_{k+1}|| \le C_w$$

The bound hold for t = k+1 as well. Hence by the principle of mathematical induction :
$$\forall t > 0 \ \ ||\bar{w}_t|| \le C_w$$

$\square$

**Lemma 10.** *The norm of target average reward estimator $\bar{\rho}_t$ is bounded*
$$\forall t > 0 \ \ ||\bar{\rho}_t|| \le C_r + 2C_w$$

*Here, $C_w$ is the upper bound on critic parameter $w_t$ (Algorithm 2, step 8), $C_r$ is the upper bound on rewards (Assumption 5).*

*Proof.* For t=1:

$$\bar{\rho}_1 = (1 - \beta_0)\bar{\rho}_0 + \beta_0\rho_1$$
$$||\bar{\rho}_1|| \le (1 - \beta_0)||\bar{\rho}_0|| + \beta_0||\rho_1||$$
$$||\bar{\rho}_1|| \le (1 - \beta_0)(C_r + 2C_w) + \beta_0(C_r + 2C_w) \quad \text{(Assumption 11)}$$
$$||\bar{\rho}_1|| \le C_r + 2C_w$$

The bound hold for t=1.
Let the bound hold for t = k. We will prove that the bound will also hold for k+1

$$\bar{\rho}_{k+1} = (1 - \beta_k)\bar{\rho}_k + \beta_k\rho_{k+1}$$
$$||\bar{\rho}_{k+1}|| \le (1 - \beta_k)||\bar{\rho}_k|| + \beta_k||\rho_{k+1}||$$
$$||\bar{\rho}_{k+1}|| \le (1 - \beta_k)(C_r + 2C_w) + \beta_k(C_r + 2C_w) \quad \text{(Assumption 11)}$$
$$||\bar{\rho}_{k+1}|| \le C_r + 2C_w$$

The bound hold for t = k+1 as well. Hence by the principle of mathematical induction :

$$\forall t > 0 \;\; ||\bar{\rho}_t|| \le C_r + 2C_w$$

$\square$

**Lemma 11.** *The $A(\theta)$ matrix defined below is negative definite for all values of $\theta$ ($\theta$ is the policy parameter).*

$$A(\theta) = \int d^\pi(s)(\phi^\pi(s)(\int P^\pi(s'|s)\phi^\pi(s') \, ds' - \phi^\pi(s))^\intercal - \eta I) \, ds$$

$$\forall x \quad x^\intercal A(\theta)x \le -\lambda ||x||^2, \quad \lambda > 0$$

*$\eta$ is the l2-regularisation coefficient from Algorithm 2 and $\eta > \lambda_{max}^{all}$, where $\lambda_{max}^{all}$ is defined in the proof below.*

*Proof.* Let:

$$A'(\theta) = \int d^\pi(s)(\phi^\pi(s)(\int P^\pi(s'|s)\phi^\pi(s') \, ds' - \phi^\pi(s))^\intercal) \, ds = A(\theta) + \eta I \tag{A.27}$$

Here, $\eta$ is the l2-regularization coefficient from Algorithm 2.

$$x^\intercal A'(\theta)x = x^\intercal \left( \frac{A'(\theta)^\intercal + A'(\theta)}{2} \right) x \le \lambda_{max}(\theta)||x||^2$$

Here, $\left( \dfrac{A'(\theta)^\intercal + A'(\theta)}{2} \right)$ is a symmetric matrix and $\lambda_{max}(\theta)$ is the maximum eigen value of the $\left( \dfrac{A'(\theta)^\intercal + A'(\theta)}{2} \right)$. Using $\lambda_{max}^{all}$ from Assumption 13:

$$\implies x^\intercal A'(\theta)x \le \lambda_{max}^{all}||x||^2$$
$$x^\intercal(A'(\theta) - \eta I)x \le (\lambda_{max}^{all} - \eta)||x||^2$$
$$x^\intercal A(\theta)x \le (\lambda_{max}^{all} - \eta)||x||^2 \;\; \text{(using A.37)}$$

Here, if we take $\eta > \lambda_{max}^{all}$ then we can set $\lambda = \eta - \lambda_{max}^{all}$.

$$\implies \forall x \quad x^\intercal A(\theta)x \le -\lambda ||x||^2, \quad \lambda > 0$$

$\square$

**Lemma 12.** *Let $\theta_1$ and $\theta_2$ be the policy parameter for $\pi'$ and $\pi$ respectively. $d^{\pi'}(\cdot)$ and $d^{\pi}(\cdot)$ be the stationary state distribution for $\pi'$ and $\pi$ respectively. Here, $D_{TV}$ denotes the total variation distance between two probability distribution function. We have:*

$$\int |d^{\pi'}(s) - d^{\pi}(s)| \, ds = 2D_{TV}(d^{\pi'}, d^{\pi'}) \leq L_d ||\theta_1 - \theta_2||$$

*Here, $L_d = 2^{m+1}(\lceil \log_\kappa a^{-1} \rceil + 1/\kappa)L_t$. $L_t$ is the Lipchitz constant for the transition probability density function (Assumption 9). Constants $a$ and $\kappa$ are from Assumption 2, $m$ is the dimension of state space.*

*Proof.*

$$\int |d^{\pi'}(s) - d^{\pi}(s)| \, ds = 2D_{TV}(d^{\pi'}, d^{\pi}) = 2D_{TV}(\mu_1, \mu_2)$$

Let $\mu_1$ and $\mu_2$ be the stationary state probability measure for $\pi'$ and $\pi$ respectively. Then we have :

$$d\mu_1 = d^{\pi'}(s) \, ds$$
$$d\mu_2 = d^{\pi}(s) \, ds$$

Using the result of Theorem 3.1 of Mitrophanov (2005):

$$2D_{TV}(\mu_1, \mu_2) \leq 2\Big(\lceil \log_\kappa a^{-1} \rceil + \frac{1}{\kappa}\Big)||K_1 - K_2|| \tag{A.28}$$

where $K_1$ and $K_2$ are probability transition kernel for markov chain induced by policy $\pi'$ and $\pi$.

From equation A.28:

$$||K_1 - K_2|| \leq \sup_{||g||_{TV}=1} ||\int g(ds)(K_1(\cdot|s) - K_2(\cdot|s))||_{TV}$$

$$
\begin{aligned}
||\int g(ds)(K_1(\cdot|s) - K_2(\cdot|s))||_{TV} &\leq \sup_{|f|\leq 1} |\iint f(s')(K_1 - k_2)(ds'|s)g(ds)| \\
&\leq \sup_{|f|\leq 1} |\iint f(s')(P^{\pi'}(s'|s) - P^{\pi}(s'|s))(s'|s)g(ds)ds'| \\
&\leq \sup_{|f|\leq 1} \iint |f(s')| \, |(P^{\pi'}(s'|s) - P^{\pi}(s'|s)|g(ds)ds' \\
&\leq L_t ||\theta_1 - \theta_2|| \int g(ds) \int ds' \\
&\leq 2^m L_t ||\theta_1 - \theta_2||
\end{aligned}
$$

$$\implies ||K_1 - K_2|| \leq 2^m L_t ||\theta_1 - \theta_2|| \tag{A.29}$$

From equation A.28 and equation A.29:

$$
\begin{aligned}
\int |d^{\pi'}(s) - d^{\pi}(s)| \, ds = 2D_{TV}(d^{\pi'}, d^{\pi'}) &\leq 2^{m+1}(\lceil \log_\kappa a^{-1} \rceil + \frac{1}{\kappa})L_t ||\theta_1 - \theta_2|| \\
&\leq L_d ||\theta_1 - \theta_2||
\end{aligned}
$$

$$\square$$

**Lemma 13.** *The optimal critic parameter $w_\epsilon^*$ according to compatible function approximation Lemma (2) is bounded by constant $C_{w_\epsilon^*}$.*

$$||w_\epsilon^*|| \le C_{w_\epsilon^*}$$

*Proof.* From Lemma2:

$$\nabla_\theta \rho(\pi) = \int_S d^\pi(s) \nabla_a Q_{diff}^\pi(s, a)|_{a=\pi(s)} \nabla_\theta \pi(s, \theta) \, ds$$

$$= \int_S d^\pi(s) \nabla_a Q_{diff}^w(s, a)|_{a=\pi(s)} \nabla_\theta \pi(s, \theta) \, ds$$

$$= \int_S d^\pi(s) \nabla_\theta \pi(s, \theta) \nabla_\theta \pi(s, \theta)^\intercal w_\epsilon^* \, ds$$

$$= E[\nabla_\theta \pi(s, \theta) \nabla_\theta \pi(s, \theta)^\intercal] w_\epsilon^*$$

Here,

$$H_\theta = E[\nabla_\theta \pi(s, \theta) \nabla_\theta \pi(s, \theta)^\intercal]$$

$$\nabla_\theta \rho(\pi) = H_\theta w_\epsilon^*$$
$$\implies w_\epsilon^* = H_\theta^{-1} \nabla_\theta \rho(\pi)$$
$$\implies ||w_\epsilon^*|| \le ||H_\theta^{-1}|| \, ||\nabla_\theta \rho(\pi)||$$

By using Assumption 14, the lower bound on minimum eigenvalue of $H_\theta$ for all $\theta$ is $\lambda_{min}^\epsilon$ and using Assumption 6 and 7 :

$$||w_\epsilon^*|| \le \frac{L_a L_\pi}{\lambda_{min}^\epsilon} = C_{w_\epsilon^*}$$

$\square$

**Lemma 14.** *The average reward performance metric, defined in 3, $\rho(\pi)(\rho(\theta))$ is Lipchitz continuous wrt to the policy (actor) parameter $\theta$.*

$$||\rho(\theta_1) - \rho(\theta_2)|| \le L_p ||\theta_1 - \theta_2||$$

*Proof.* Let $\theta_1$ and $\theta_2$ be the policy parameters of policy $\pi'$ and $\pi$.

$$||\rho(\theta_1) - \rho(\theta_2)|| = ||\rho(\pi') - \rho(\pi)||$$

$$= || \int_S d^{\pi'}(s) R^{\pi'}(s) \, ds - \int_S d^\pi(s) R^\pi(s) \, ds||$$

$$\le || \int_S (d^{\pi'}(s) - d^\pi(s)) R^{\pi'}(s) \, ds|| + || \int_S d^\pi(s)(R^{\pi'}(s) - R^\pi(s)) \, ds||$$

$$\le L_d ||\theta_1 - \theta_2|| \quad \text{(Lemma 12)}$$
$$+ L_r ||\theta_1 - \theta_2|| \quad \text{(Assumption 10)}$$
$$\le (L_d + L_r) ||\theta_1 - \theta_2|| = L_p ||\theta_1 - \theta_2|| \quad (L_d + L_r = L_p)$$

$\square$

**Lemma 15.** *The optimal critic parameter $w(\theta_t)^*$ as a function of actor parameter $\theta_t$ is Lipchitz continuous with constant $L_v$ for off-policy case. Note: $w_t^* = w(\theta_t)^*$. $\mu$ is the behaviour policy.*

$$||w_t^* - w_{t+1}^*|| \le L_v ||\theta_{t+1} - \theta_t||$$

*Proof.* $\eta$ is the l2-regularisation coefficient from Algorithm 3 and $\eta > \chi_{max}^{all}$, where $\chi_{max}^{all}$ is defined in Lemma 16. Because of carefully setting the value of $\eta$, $A(\theta_t)$ is negative definite. Thus, for on-policy TD(0) with l2-regularization and target estimators, the following condition holds true for optimal critic parameter $w_t^*$:

$$E[(R^\mu(s) - \rho_t^*)\phi^\pi(s) + (\phi^\pi(s)(E[\phi^\pi(s')] - \phi^\pi(s))^\intercal - \eta I)w_t^*] = 0$$

$$b(\theta_t) := E[(R^\mu(s) - \rho_t^*)\phi^\pi(s)]$$
$$A(\theta_t) := E[(\phi^\pi(s)(E[\phi^\pi(s')] - \phi^\pi(s))^\intercal - \eta I)]$$

$$\therefore b(\theta_t) + A(\theta_t)w_t^* = 0 \implies w_t^* = -A(\theta_t)^{-1}b(\theta_t)$$

Expectation above is with respect to stationary state distribution $d^\mu(\cdot)$ of policy $\mu$. Please note the abuse of notation here, $A(\theta_t)$ is actually same as $A_{off}^\mu(\theta_t)$ of Lemma 16.

$$
\begin{aligned}
||w_t^* - w_{t+1}^*|| &= ||A(\theta_t)^{-1}b(\theta_t) - A(\theta_{t+1})^{-1}b(\theta_{t+1})|| \\
&\leq ||A(\theta_t)^{-1}b(\theta_t) - A(\theta_{t+1})^{-1}b(\theta_t) + A(\theta_{t+1})^{-1}b(\theta_t) - A(\theta_{t+1})^{-1}b(\theta_{t+1})|| \\
&\leq ||A(\theta_t)^{-1} - A(\theta_{t+1})^{-1}|| \, ||b(\theta_t)|| \quad \text{①} \\
&\quad + ||A(\theta_{t+1})^{-1}|| \, ||b(\theta_t) - b(\theta_{t+1})|| \quad \text{②}
\end{aligned}
$$
$$\text{(A.30)}$$

From equation A.30:

①:

$$
\begin{aligned}
||A(\theta_t)^{-1} - A(\theta_{t+1})^{-1}|| &= ||A(\theta_t)^{-1}A(\theta_{t+1})A(\theta_{t+1})^{-1} - A(\theta_t)^{-1}A(\theta_t)^A(\theta_{t+1})^{-1}|| \\
&\leq ||A(\theta_t)^{-1}|| \, ||A(\theta_t) - A(\theta_{t+1})|| \, ||A(\theta_{t+1})^{-1}||
\end{aligned}
$$
$$\text{(A.31)}$$

From equation A.31:

Here, $\pi'$ and $\pi$ represents the policy with parameter $\theta_{t+1}$ and $\theta_t$ respectively and $\mu$ be the behaviour policy .

$$
\begin{aligned}
||A(\theta_t) - A(\theta_{t+1})|| &\leq || \int d^\mu(s)(\phi^{\pi'}(s)(\int P^\mu(s'|s)\phi^{\pi'}(s') \, ds' - \phi^{\pi'}(s))^\intercal - \eta I) \, ds \\
&\quad - \int d^\mu(s)(\phi^\pi(s)(\int P^\mu(s'|s)\phi^\pi(s') \, ds' - \phi^\pi(s))^\intercal - \eta I) \, ds|| \\
&\leq || \int d^\mu(s)(\phi^{\pi'}(s)(\int P^\mu(s'|s)\phi^{\pi'}(s') \, ds')^\intercal) \, ds \\
&\quad - \int d^\mu(s)(\phi^\pi(s)(\int P^\mu(s'|s)\phi^\pi(s') \, ds')^\intercal) \, ds|| \quad \text{①} \\
&\leq || \int d^\mu(s)(\phi^\pi(s)(\phi^\pi(s))^\intercal) \, ds - \int d^\mu(s)(\phi^{\pi'}(s)(\phi^{\pi'}(s))^\intercal) \, ds|| \quad \text{②}
\end{aligned}
$$
$$\text{(A.32)}$$

From equation A.32:

①:

$$||\int d^\mu(s)(\phi^{\pi'}(s)(\int P^\mu(s'|s)\phi^{\pi'}(s')\,ds')^\intercal)\,ds - \int d^\mu(s)(\phi^\pi(s)(\int P^\mu(s'|s)\phi^\pi(s')\,ds')^\intercal)\,ds||$$

$$\leq ||\int (d^\mu(s) - d^\mu(s))\phi^{\pi'}(s)(\int P^\mu(s'|s)\phi^{\pi'}(s')\,ds')^\intercal\,ds||$$

$$+ ||\int d^\mu(s)(\phi^{\pi'}(s) - \phi^\pi(s))(\int P^\mu(s'|s)\phi^{\pi'}(s')\,ds')^\intercal\,ds||$$

$$+ ||\int d^\mu(s)\phi^\pi(s)(\int (P^\mu(s'|s) - P^\mu(s'|s))\phi^{\pi'}(s')\,ds')^\intercal\,ds||$$

$$+ ||\int d^\mu(s)\phi^\pi(s)(\int P^\mu(s'|s)(\phi^{\pi'}(s') - \phi^\pi(s'))\,ds'^\intercal)\,ds$$

$$\leq L_\phi||\theta_{t+1} - \theta_t|| \qquad \text{(Assumption 8)}$$
$$+ L_\phi||\theta_{t+1} - \theta_t|| \qquad \text{(Assumption 8)}$$

$$||\int d^\mu(s)(\phi^{\pi'}(s)(\int P^\mu(s'|s)\phi^{\pi'}(s')\,ds')^\intercal)\,ds - \int d^\mu(s)(\phi^\pi(s)(\int P^\mu(s'|s)\phi^\pi(s')\,ds')^\intercal)\,ds||$$
$$\leq 2L_\phi||\theta_{t+1} - \theta_t||$$

$$\text{(A.33)}$$

From equation A.32:

②:

$$||\int d^\mu(s)(\phi^\pi(s)(\phi^\pi(s))^\intercal)\,ds - \int d^\mu(s)(\phi^{\pi'}(s)(\phi^{\pi'}(s))^\intercal)\,ds||$$
$$\leq ||\int d^\mu(s)(\phi^\pi(s) - \phi^{\pi'}(s))(\phi^\pi(s))^\intercal\,ds||$$
$$+ ||\int d^\mu(s)\phi^{\pi'}(s)(\phi^\pi(s) - \phi^{\pi'}(s))^\intercal\,ds||$$
$$\leq 2L_\phi||\theta_{t+1} - \theta_t||$$

$$\text{(A.34)}$$

Using equation A.33 and equation A.34 in equation A.32

$$||A(\theta_t) - A(\theta_{t+1})|| \leq (4L_\phi + L_t)||\theta_{t+1} - \theta_t|| \tag{A.35}$$

From equation A.30:

②:

$$||b(\theta_t) - b(\theta_{t+1})|| = ||\int d^\mu(s)((R^\mu(s) - \rho^*_{t+1})\phi^{\pi'}(s)\,ds - \int d^\mu(s)(R^\mu(s) - \rho^*_t)\phi^\pi(s)\,ds||$$

$$\leq ||\int d^\mu(s)(R^\mu(s)\phi^{\pi'}(s)\,ds - \int d^\mu(s)R^\mu(s)\phi^\pi(s)\,ds||$$

$$+ ||\int d^\mu(s)\rho^*_{t+1}\phi^{\pi'}(s)\,ds - \int d^\mu(s)\rho^*_t\phi^\pi(s)\,ds||$$

$$\leq ||\int d^\mu(s)(R^\mu(s) - R^\mu(s))\phi^{\pi'}(s)\,ds||$$

$$+ ||\int d^\mu(s)R^\mu(s)(\phi^{\pi'}(s) - \phi^\pi(s)\,ds||$$

$$+ ||\int d^\mu(s)(\rho^*_{t+1} - \rho^*_t)\phi^{\pi'}(s)\,ds||$$

$$+ ||\int d^\mu(s)\rho^*_t(\phi^{\pi'}(s) - \phi^\pi(s)\,ds||$$

$$\leq C_r L_\phi ||\theta_{t+1} - \theta_t|| \quad \text{(Assumption 5)}$$
$$+ L_p ||\theta_{t+1} - \theta_t|| \quad \text{(Lemma 14)}$$
$$+ C_r L_\phi ||\theta_{t+1} - \theta_t|| \quad \text{(Assumption 5)}$$

$$\implies ||b(\theta_t) - b(\theta_{t+1})|| \leq (2C_r L_\phi + L_p)||\theta_{t+1} - \theta_t|| \tag{A.36}$$

Using equation A.31, equation A.35 and equation A.36 in equation A.30:

$$||w^*_t - w^*_{t+1}|| \leq ||A(\theta_t)^{-1} - A(\theta_{t+1})^{-1}||\,||b(\theta_t)|| + ||A(\theta_{t+1})^{-1}||\,||b(\theta_t) - b(\theta_{t+1})||$$

$$\leq ||A(\theta_t)^{-1}||\,||A(\theta_t) - A(\theta_{t+1})||\,||A(\theta_{t+1})^{-1}||\,||b(\theta_t)||$$
$$+ ||A(\theta_{t+1})^{-1}||\,||b(\theta_t) - b(\theta_{t+1})||$$

$$\leq 4L_\phi ||A(\theta_t)^{-1}||\,||A(\theta_{t+1})^{-1}||\,||b(\theta_t)||\,||\theta_{t+1} - \theta_t||$$
$$+ (2C_r L_\phi + L_p)||A(\theta_{t+1})^{-1}||\,||\theta_{t+1} - \theta_t||$$

Note:

- $||b(\theta_t)|| = ||\int d^\mu(s)(\phi^\pi(s)(\phi^\pi(s))^\intercal)\,ds|| \leq C_r$ (Assumption 5)
- Let $\lambda_{min}$ is the lower bound on eigen values of $A(\theta)$ for all $\theta$.

$$\therefore ||w^*_t - w^*_{t+1}|| \leq \frac{C_r(4L_\phi)}{\lambda^2_{min}}||\theta_{t+1} - \theta_t||$$
$$+ \frac{(2C_r L_\phi + L_p)}{\lambda_{min}}||\theta_{t+1} - \theta_t||$$
$$\leq L_v ||\theta_{t+1} - \theta_t||$$

where,

$$L_v = \frac{4C_r L_\phi}{\lambda^2_{min}} + \frac{C_r L_\phi}{\lambda_{min}}$$

$\square$

**Lemma 16.** *The $A^\mu_{off}(\theta)$ matrix defined below is negative definite for all values of $\theta$ ($\theta$ is the policy parameter). $\theta^\mu$ is the policy parameter for behaviour policy $\mu$.*

$$A^\mu_{off}(\theta) := \int d^\mu(s)(\phi^\pi(s)(\int P^\pi(s'|s)\phi^\pi(s')\,ds' - \phi^\pi(s))^\intercal - \eta I)\,ds$$

$$\forall x \quad x^\intercal A^\mu_{off}(\theta)x \leq -\lambda||x||^2, \quad \lambda > 0$$

*$\eta$ is the l2-regularisation coefficient from Algorithm 3 and $\eta > \chi^{all}_{max}$, where $\chi^{all}_{max}$ is defined in the proof below.*

*Proof.* Let:

$$A^\mu_{off}{}'(\theta) = \int d^\mu(s)(\phi^\pi(s)(\int P^\pi(s'|s)\phi^\pi(s')\, ds' - \phi^\pi(s))^\intercal)\, ds = A^\mu_{off}(\theta) + \eta I \quad \text{(A.37)}$$

Here, $\eta$ is the l2-regularization coefficient from Algorithm 2.

$$x^\intercal A^\mu_{off}{}'(\theta)x = x^\intercal \Big( \frac{A^\mu_{off}{}'(\theta)^\intercal + A^\mu_{off}{}'(\theta)}{2} \Big)x \leq \chi_{max}(\theta)||x||^2$$

Here, $\Big( \dfrac{A^\mu_{off}{}'(\theta)^\intercal + A^\mu_{off}{}'(\theta)}{2} \Big)$ is a symmetric matrix and $\chi_{max}(\theta)$ is the maximum eigen value of the $\Big( \dfrac{A^\mu_{off}{}'(\theta)^\intercal + A^\mu_{off}{}'(\theta)}{2} \Big)$. Using $\chi^{all}_{max}$ from Assumption 15:

$$\implies x^\intercal A^\mu_{off}{}'(\theta)x \leq \chi^{all}_{max}||x||^2$$
$$x^\intercal (A^\mu_{off}{}'(\theta) - \eta I)x \leq (\chi^{all}_{max} - \eta)||x||^2$$
$$x^\intercal A^\mu_{off}(\theta)x \leq (\chi^{all}_{max} - \eta)||x||^2 \quad \text{(using A.37)}$$

Here, if we take $\eta > \chi^{all}_{max}$ then we can set $\lambda = \eta - \chi^{all}_{max}$.

$$\implies \forall x \quad x^\intercal A^\mu_{off}(\theta)x \leq -\lambda||x||^2, \quad \lambda > 0$$

$\square$

**Lemma 17.** *Let the cumulative error of off-policy actor be $\sum_{t=0}^{T-1} E||\widehat{\nabla_\theta \rho}(\theta_t)||^2$ and cumulative error of critic be $\sum_{t=0}^{T-1} E||\Delta w_t||^2$. $\theta_t$ and $w_t$ are the actor and linear critic parameter at time t. $\theta^\mu$ is the policy parameter for behavior policy $\mu$. Bound on the cumulative error of off-policy actor with behaviour policy $\mu$ is proven using cumulative error of critic as:*

$$\frac{1}{T}\sum_{t=0}^{T-1} E||\widehat{\nabla_\theta \rho}(\theta_t)||^2 \leq 4\frac{C_r}{C_\gamma}T^{v-1} + 6C_\pi^4\Big(\frac{1}{T}\sum_{t=0}^{T-1} E||\Delta w_t||^2\Big) + 6C_\pi^4\Big(\tau^2 + \frac{4}{M}C_{w_\epsilon^*}^2\Big)$$
$$+ 2\frac{C_\gamma L_J G_\theta^2}{1-v}T^{-v} + \frac{Z}{T}\sum_{t=0}^{T-1} E||\theta^\mu - \theta^t||^2$$

*Here, $C_r$ is the upper bound on rewards (Assumption 5), $C_\gamma$, v are constants used for step size $\gamma_t$ (Assumption 3, $||\nabla_\theta \pi(s)|| \leq C_\pi$ (Assumption 7), $\Delta w_t = w_t - w_t^*, \tau = \max_t ||w_t^* - w_{\epsilon,t}^*||$, $w_\epsilon^*$ is the optimal critic parameter according to Lemma2. $w_t^*$ is the optimal parameters given by TD(0) algorithm corresponding to policy parameter $\theta_t$. Constant $C_{w^*}$ is defined in Lemma 13. $L_J$ is the coefficient used in smoothness condition of the non convex function $\rho(\theta)$. Constant $G_\theta$ is defined in Lemma 7. M is the size of batch of samples used to update parameters. $Z = 2^{m+1}C(\lceil \log_\kappa a^{-1} \rceil + 1/\kappa)L_t$ with $L_t$ being the Lipchitz constant for the transition probability density function (Assumption 9). Constants a and $\kappa$ are from Assumption 2, m is the dimension of state space, and $C = \max_s ||\nabla_a Q^\pi_{diff}(s,a)|_{a=\pi(s)} \nabla_\theta \pi(s,\theta)||$.*

*Proof.*

$$\frac{1}{T}\sum_{t=0}^{T-1}E||\widehat{\nabla_\theta\rho}(\theta_t)||^2 = \frac{1}{T}\sum_{t=0}^{T-1}E||\nabla_\theta\rho(\theta_t) + \widehat{\nabla_\theta\rho}(\theta_t) - \nabla_\theta\rho(\theta_t)||^2$$

$$\leq \frac{1}{T}\sum_{t=0}^{T-1}E||\nabla_\theta\rho(\theta_t)||^2 + \frac{1}{T}\sum_{t=0}^{T-1}E||\widehat{\nabla_\theta\rho}(\theta_t) - \nabla_\theta\rho(\theta_t)||^2$$

Using Theorem 2 and Lemma 3:

$$\frac{1}{T}\sum_{t=0}^{T-1}E||\widehat{\nabla_\theta\rho}(\theta_t)||^2 \leq 4\frac{C_r}{C_\gamma}T^{v-1} + 6C_\pi^4(\frac{1}{T}\sum_{t=0}^{T-1}E||\Delta w_t||^2) + 6C_\pi^4(\tau^2 + \frac{4}{M}C_{w_\epsilon^*}^2)$$

$$+ 2\frac{C_\gamma L_J G_\theta^2}{1-v}T^{-v} + \frac{Z}{T}\sum_{t=0}^{T-1}E||\theta^\mu - \theta^t||^2$$

$\square$

**Theorem 4.** *The off-policy average reward actor critic algorithm (Algorithm 3) with behavior policy $\mu$ obtains an $\epsilon$-accurate optimal point with sample complexity of $\Omega(\epsilon^{-2.5})$. Here $\theta_\mu$ refers to the behavior policy parameter and $\theta_t$ refers to the target or current policy parameter. We obtain*

$$\min_{0\leq t\leq T-1}E||\widehat{\nabla_\theta\rho}(\theta_t)||^2 = \mathcal{O}\left(\frac{1}{T^{0.4}}\right) + \mathcal{O}(1) + \mathcal{O}(W_\theta^2)$$

$$\leq \epsilon + \mathcal{O}(1) + \mathcal{O}(W_\theta^2)$$

*where $W_\theta := \max_t ||\theta_\mu - \theta_t||$.*

*Proof.* Lemma 4 and Lemma 5 will hold in the case of off-policy update. Lemma 4 will require Lemma 15 instead of Lemma 6.

Using Lemma 4 and Lemma 5 and using the procedure followed in Theorem 3 to obtain asymptotic notations, we have:

$$\frac{1}{T}\sum_{t=0}^{T-1}E||\Delta w_t||^2 = \mathcal{O}\left(\frac{1}{T^{1-\sigma}}\right) + \mathcal{O}\left(\frac{1}{T^\sigma}\right) + \mathcal{O}\left(\frac{1}{T^{2(v-\sigma)}}\right) + \mathcal{O}(1) \qquad \text{(A.38)}$$

Using Lemma 17 and equation A.38:

$$\min_{0\leq t\leq T-1}E||\widehat{\nabla_\theta\rho}(\theta_t)||^2 = \mathcal{O}\left(\frac{1}{T^{1-v}}\right) + \mathcal{O}\left(\frac{1}{T^v}\right) + \mathcal{O}\left(\frac{1}{T^{1-\sigma}}\right) + \mathcal{O}\left(\frac{1}{T^\sigma}\right)$$

$$+ \mathcal{O}\left(\frac{1}{T^{2(v-\sigma)}}\right) + \mathcal{O}(1) + \frac{Z}{T}\sum_{t=0}^{T-1}E||\theta^\mu - \theta^t||^2$$

By setting $v = 3/5$ and $\sigma = 2/5$, we obtain:

$$\min_{0 \leq t \leq T-1} E||\widehat{\nabla_\theta \rho}(\theta_t)||^2 = \mathcal{O}\left(\frac{1}{T^{0.4}}\right) + \mathcal{O}(1) + \frac{Z}{T}\sum_{t=0}^{T-1} E||\theta^\mu - \theta^t||^2$$

$$= \mathcal{O}\left(\frac{1}{T^{0.4}}\right) + \mathcal{O}(1) + ZN_\theta^2$$

$$= \mathcal{O}\left(\frac{1}{T^{0.4}}\right) + \mathcal{O}(1) + \mathcal{O}(N_\theta^2).$$

Further,

$$\mathcal{O}\left(\frac{1}{T^{0.4}}\right) \leq \epsilon.$$

Hence, the sample complexity of off-policy average reward actor-critic algorithm is $\Omega(\epsilon^{-2.5})$. $\quad\square$

## A.2 BOUNDEDNESS OF CRITIC PARAMETER

In this section we prove the critic parameter $w$ used in Algorithm 2 and 3 is bounded even without using projection operator $\Gamma_{C_w}$ defined as $\Gamma_{C_w} : \mathbb{R}^k \to B$, where $B(\subset \mathbb{R}^k)$ is a compact convex set. Let policy $\pi$ is parameterized by $\theta$.

For simplicity of proof we are assuming the batch size M to be 1. Critic parameter $w_t \in \mathbb{R}^k$, $\phi^\pi(s) \in \mathbb{R}^k$ and $\rho_t$ is a scalar. Let the update rules used for critic parameter and average reward estimator be as follows:

$$w_{t+1} = w_t + \alpha_t\left(R^\pi(s_t) - \bar{\rho}_t + \phi^\pi(s'_t)^\mathsf{T}\bar{w}_t - \phi^\pi(s_t)^\mathsf{T}w_t\right)\phi^\pi(s_t) - \alpha_t\eta w_t$$

$$\rho_{t+1} = \rho_t + \alpha_t\left(R^\pi(s_t) - \rho_t + \phi^\pi(s'_t)^\mathsf{T}\bar{w}_t - \phi^\pi(s_t)^\mathsf{T}\bar{w}_t\right) \qquad (A.39)$$

$$\overline{w}_{t+1} = \overline{w}_t + \beta_t(w_{t+1} - \overline{w}_{t+1})$$

$$\overline{\rho}_{t+1} = \overline{\rho}_t + \beta_t(\rho_{t+1} - \overline{\rho}_{t+1})$$

Let us define $z_t$ as $[w_t \ \rho_t]^\mathsf{T}$ and $\bar{z}_t$ as $[\bar{w}_t \ \bar{\rho}_t]^\mathsf{T}$. $\mathbf{0}$ is a vector in $\mathbb{R}^k$ and $I_0$ is an identity matrix in $\mathbb{R}^{(k+1)\times(k+1)}$ with $I_0[k][k] = 0$ (assuming indexing starts from 0).

$$\begin{bmatrix} w_{t+1} \\ \rho_{t+1} \end{bmatrix} = \begin{bmatrix} w_t \\ \rho_t \end{bmatrix} + \alpha_t\left(R^\pi(s_t)\begin{bmatrix} \phi^\pi(s_t) \\ 1 \end{bmatrix} + \begin{bmatrix} \phi^\pi(s_t)\phi^\pi(s'_t)^\mathsf{T} & -\phi^\pi(s_t) \\ \phi^\pi(s'_t)^\mathsf{T} - \phi^\pi(s_t)^\mathsf{T} & 0 \end{bmatrix}\begin{bmatrix} \bar{w}_t \\ \bar{\rho}_t \end{bmatrix}\right.$$

$$\left. - \begin{bmatrix} \phi^\pi(s_t)\phi^\pi(s_t)^\mathsf{T} & \mathbf{0} \\ \mathbf{0}^\mathsf{T} & 1 \end{bmatrix}\begin{bmatrix} w_t \\ \rho_t \end{bmatrix} - \eta I_0 \begin{bmatrix} w_t \\ \rho_t \end{bmatrix}\right) \qquad (A.40)$$

$$\begin{bmatrix} \bar{w}_{t+1} \\ \bar{\rho}_{t+1} \end{bmatrix} = \begin{bmatrix} \bar{w}_t \\ \bar{\rho}_t \end{bmatrix} + \beta_t\left(\begin{bmatrix} w_{t+1} \\ \rho_{t+1} \end{bmatrix} - \begin{bmatrix} \bar{w}_t \\ \bar{\rho}_t \end{bmatrix}\right)$$

Here, $R^\pi(s_t)\begin{bmatrix} \phi^\pi(s_t) \\ 1 \end{bmatrix} = R_\phi^\pi(s_t)$, $A_\phi(s_t, s'_t) = \begin{bmatrix} \phi^\pi(s_t)\phi^\pi(s'_t)^\mathsf{T} & -\phi^\pi(s_t) \\ \phi^\pi(s'_t)^\mathsf{T} - \phi^\pi(s_t)^\mathsf{T} & 0 \end{bmatrix}$ and $B_\phi(s_t) = \begin{bmatrix} \phi^\pi(s_t)\phi^\pi(s_t)^\mathsf{T} & \mathbf{0} \\ \mathbf{0}^\mathsf{T} & 1 \end{bmatrix}$

$$z_{t+1} = z_t + \alpha_t\left(R_\phi^\pi(s_t) + A_\phi(s_t, s'_t)\bar{z}_t - (B_\phi(s_t) + \eta I_0)z_t\right)$$

$$\bar{z}_{t+1} = \bar{z}_t + \beta_t(z_{t+1} - \bar{z}_t) \qquad (A.41)$$

Now, we will use the extension of stability criteria for iterates given Borkar & Meyn (2000) to two timescale stochastic approximation scheme (Lakshminarayanan & Bhatnagar, 2017) to show the

boundedness of the critic parameter and average reward estimator together. Let us write A.41 in the standard form of stochastic approximation scheme.

$$z_{t+1} = z_t + \alpha_t\big(h(z_t, \bar{z}_t) + \mathcal{M}^1_{t+1}\big)$$

Let, $\bar{R}^\pi_\phi = \int_S d^\pi(s_t) R^\pi_\phi(s_t)\, ds_t$, $\bar{A}_\phi = \int_S d^\pi(s_t) \int_S P^\pi(s'_t|s_t) A_\phi(s_t, s'_t)\, ds'_t\, ds_t$, $\bar{B}_\phi = \int_S d^\pi(s_t) B_\phi(s_t)\, s_t$

Here,

$$
\begin{aligned}
h(z_t, \bar{z}_t) &= \int_S d^\pi(s_t)\big(R^\pi_\phi(s_t) + A_\phi(s_t, s'_t)\bar{z}_t - (B_\phi(s_t) + \eta I_0)z_t\big)\, ds_t \\
&= \bar{R}^\pi_\phi + \bar{A}_\phi \bar{z}_t - (\bar{B}_\phi + \eta I_0)z_t \\
\mathcal{M}^1_{t+1} &= R^\pi_\phi(s_t) + A_\phi(s_t, s'_t)\bar{z}_t - (B_\phi(s_t) + \eta I_0)z_t - h(z_t, \bar{z}_t)
\end{aligned}
$$

$$\bar{z}_{t+1} = \bar{z}_t + \beta_t\big(g(z_t, \bar{z}_t) + \mathcal{M}^2_{t+1} + \epsilon(n)\big)$$

Here,

$$
\begin{aligned}
g(z_t, \bar{z}_t) &= \lambda(\bar{z}_t) - \bar{z}_t \\
\mathcal{M}^2_{t+1} &= 0 \\
\lambda(\bar{z}_t) &= (B_\phi + \eta I_0)^{-1}(R^\pi_\phi + A_\phi \bar{z}_t) \\
\epsilon(n) &= z_{t+1} - \lambda(\bar{z}_t)
\end{aligned}
$$

$\lambda(\bar{z}_t)$ is the unique globally asymptotically stable equilibrium point of the ODE $\dot{z} = h(z(t), \bar{z})$. $\lambda$ used here has no relation to usage of $\lambda$ in any other section of the paper. Using Lemma 1 of Chapter 6 of (Borkar, 2009), we have $\|z_{t+1} - \lambda(\bar{z}_t)\| \to 0$. Hence $\epsilon(n) = o(1)$. Therefore we can use the conclusion of (Lakshminarayanan & Bhatnagar, 2017).

We will now satisfy condition A1 till condition A5 of (Lakshminarayanan & Bhatnagar, 2017) to prove the boundedness of the critic parameter:

**Condition A1**:
$$
\begin{aligned}
\|h(z_1, \bar{z}_1) - h(z_2, \bar{z}_2)\| &= \|\bar{A}_\phi(\bar{z}_1 - \bar{z}_2) - (\bar{B}_\phi + \eta I_0)(z_1 - z_2)\| \\
&\le \|\bar{A}_\phi\|\|\bar{z}_1 - \bar{z}_2\| + \|\bar{B}_\phi + \eta I_0\|\|z_1 - z_2\| \\
&\le \max(\|\bar{A}_\phi\|, \|\bar{B}_\phi + \eta I_0\|)(\|\bar{z}_1 - \bar{z}_2\| + \|\|z_1 - z_2\|) \\
&= L_h(\|\bar{z}_1 - \bar{z}_2\| + \|\|z_1 - z_2\|)  \quad (L_h = \max(\|\bar{A}_\phi\|, \|\bar{B}_\phi + \eta I_0\|))
\end{aligned}
$$
(A.42)

Therefore, $h(z, \bar{z})$ is Lipchitz continuous with constant $L_h$.

$$
\begin{aligned}
\|g(z_1, \bar{z}_1) - g(z_2, \bar{z}_2)\| &= \|((\bar{B}_\phi + \eta I_0)A_\phi - I)(\bar{z}_1 - \bar{z}_2)\| \\
&\le \|((\bar{B}_\phi + \eta I_0)A_\phi - I)\|\|\bar{z}_1 - \bar{z}_2\| \\
&= L_g\|\bar{z}_1 - \bar{z}_2\|  \quad (L_g = \|((\bar{B}_\phi + \eta I_0)A_\phi - I)\|)
\end{aligned}
$$
(A.43)

Therefore, $g(z, \bar{z})$ is Lipchitz continuous with constant $L_g$.

Using A.42 and A.43, condition A1 is satisfied.

**Condition A2:**
Let us define an increasing sequence of $\sigma-$fields $\{\mathcal{F}_t\}$ as $\{z_m, \bar{z}_m, \mathcal{M}^1_m, \mathcal{M}^2_m, m \le t\}$.

$$
\begin{aligned}
E[\mathcal{M}^1_{t+1}|\mathcal{F}_t] &= E[R^\pi_\phi(s_t) + A_\phi(s_t, s'_t)\bar{z}_t - (B_\phi(s_t) + \eta I_0)z_t - h(z_t, \bar{z}_t)|\mathcal{F}_t] \\
&= \int_S d^\pi(s_t)\big(R^\pi_\phi(s_t) + A_\phi(s_t, s'_t)\bar{z}_t - (B_\phi(s_t) + \eta I_0)z_t\big)\, ds_t - h(z_t, \bar{z}_t) \\
&= 0
\end{aligned}
$$

$$E[\mathcal{M}_{t+1}^2|\mathcal{F}_t] = 0$$

Hence, $\{\mathcal{M}_t^1\}$ and $\{\mathcal{M}_t^2\}$ are martingale difference sequence.

$$\begin{aligned}
\|\mathcal{M}_{t+1}^1\|^2 &= \|(R_\phi^\pi(s_t) - \bar{R}_\phi) + (A_\phi(s_t, s_t') - \bar{A}_\phi)\bar{z}_t - (B_\phi(s_t) - \bar{B}_\phi(s_t))z_t\|^2 \\
&\leq 3(\|R_\phi^\pi(s_t) - \bar{R}_\phi\|^2 + \|(A_\phi(s_t, s_t') - \bar{A}_\phi)\|^2\|\bar{z}_t\|^2 + \|B_\phi(s_t) - \bar{B}_\phi(s_t))\|^2\|z_t\|^2) \\
&\leq K_1(1 + \|z_t\|^2 + \|\bar{z}_t\|^2)
\end{aligned}$$

Here, $K_1 = 6\max(\|R_\phi^\pi(s_t)\|, \|(A_\phi(s_t, s_t')\|, \|B_\phi(s_t)\|)$ and it follows from Assumption 4 and 5. We have, $E[\|\mathcal{M}_{t+1}^1\|^2||\mathcal{F}_t] \leq K_1(1+\|z_t\|^2+\|\bar{z}_t\|^2)$ and $E[\|\mathcal{M}_{t+1}^2\|^2||\mathcal{F}_t] \leq K_2(1+\|z_t\|^2+\|\bar{z}_t\|^2)$. $K_2$ can be any positive constant. Hence condition A2 is satisfied.

### Condition A3:

We have, $\sum_t \alpha_t = \sum_t \frac{C_\alpha}{(1+t)^\sigma} = \infty$, $\sum_t \beta_t = \sum_t \frac{C_\beta}{(1+t)^u} = \infty$ and $\sum_t(\alpha_t^2 + \beta_t^2) = \sum_t \left( (\frac{C_\alpha}{(1+t)^\sigma})^2 + (\frac{C_\beta}{(1+t)^u})^2 \right) < \infty$. We can carefully set the value of $\sigma$ and $u$ to satisfy the conditions on step sizes.

### Condition A4:

$$\begin{aligned}
h_c(z, \bar{z}) &:= \frac{h(cz, c\bar{z})}{c} \\
h_c(z, \bar{z}) &= \frac{\bar{R}_\phi^\pi + c\bar{A}_\phi\bar{z}_t - c(\bar{B}_\phi + \eta I_0)z_t}{c} \\
\lim_{c\to\infty} h_c(z, \bar{z}) &= \lim_{c\to\infty} \frac{\bar{R}_\phi^\pi + c\bar{A}_\phi\bar{z}_t - c(\bar{B}_\phi + \eta I_0)z_t}{c} \\
&= \bar{A}_\phi\bar{z}_t - (\bar{B}_\phi + \eta I_0)z_t
\end{aligned}$$

Let us define $h_\infty(z_t, \bar{z}_t) := \bar{A}_\phi\bar{z}_t - (\bar{B}_\phi + \eta I_0)z_t$. The ODE $\dot{z}(t) := h_\infty(z(t), \bar{z})$ has a unique globally asymptotically stable equilibrium point $\lambda_\infty(\bar{z}) = (\bar{B}_\phi + \eta I_0)^{-1}\bar{A}_\phi\bar{z}$ if $(\bar{B}_\phi + \eta I_0)$ is positive definite matrix. Let $C_\phi = \int_S d^\pi(s_t)\phi^\pi(s_t)\phi^\pi(s_t)^\intercal \, ds_t$.

$$\begin{aligned}
\bar{B}_\phi + \eta I_0 &= \begin{bmatrix} C_\phi + \eta I & \mathbf{0} \\ \mathbf{0}^\intercal & 1 \end{bmatrix} \\
[w^\intercal \quad \rho] \begin{bmatrix} C_\phi + \eta I & \mathbf{0} \\ \mathbf{0}^\intercal & 1 \end{bmatrix} \begin{bmatrix} w \\ \rho \end{bmatrix} &= w^\intercal(C_\phi + \eta I)w + \rho^2
\end{aligned}$$

If $\eta$ is strictly greater than negative of the minimum eigenvalue of $C_\phi$ then,

$$\begin{aligned}
\forall \begin{bmatrix} w \\ p \end{bmatrix} \neq \begin{bmatrix} 0 \\ 0 \end{bmatrix} \quad [w^\intercal \quad \rho] \begin{bmatrix} C_\phi + \eta I & \mathbf{0} \\ \mathbf{0}^\intercal & 1 \end{bmatrix} \begin{bmatrix} w \\ \rho \end{bmatrix} &> 0 \\
\forall \begin{bmatrix} w \\ p \end{bmatrix} \neq \begin{bmatrix} 0 \\ 0 \end{bmatrix} \quad [w^\intercal \quad \rho] \begin{bmatrix} \bar{B}_\phi + \eta I_0 \end{bmatrix} \begin{bmatrix} w \\ \rho \end{bmatrix} &> 0
\end{aligned} \tag{A.44}$$

Hence, for $\eta + \lambda_{min}(C_\phi) > 0$, $\bar{B}_\phi + \eta I_0$ is positive definite matrix. Therefore, the ODE $\dot{z}(t) := h_\infty(z(t), \bar{z})$ has a unique globally asymptotically stable equilibrium point $\lambda_\infty(\bar{z})$ and $\lambda_\infty(0) = 0$. Condition A4 is satisfied.

### Condition A5:

$$g_c(\bar{z}) := \frac{g(c\lambda_\infty(\bar{z}), c\bar{z})}{c}$$

$$g_c(\bar{z}) = \frac{(\bar{B}_\phi + \eta I_0)^{-1}(R_\phi^\pi + cA_\phi\bar{z}) - c\bar{z}}{c}$$

$$\lim_{c\to\infty} g_c(\bar{z}) = \lim_{c\to\infty} \frac{(\bar{B}_\phi + \eta I_0)^{-1}(R_\phi^\pi + cA_\phi\bar{z}) - c\bar{z}}{c}$$ \hspace{2cm} (A.45)

$$= (\bar{B}_\phi + \eta I_0)^{-1}A_\phi\bar{z} - \bar{z}$$

Let us define $g_\infty := ((\bar{B}_\phi + \eta I_0)^{-1}A_\phi - I)\bar{z}$. The ODE $\dot{\bar{z}}(t) = g_\infty(\bar{z}(t))$ has origin as its unique globally asymptotically stable equilibrium if $I - (\bar{B}_\phi + \eta I_0)^{-1}A_\phi$ is positive definite matrix.

$\|\cdot\|$ refers to L2-norm. $\lambda_i$ are the eigenvalues of the matrix $C_\phi$. Let us assume the following:

$$\max(1, \max_i(\frac{1}{\lambda_i + \eta})) = \|(\bar{B}_\phi + \eta I_0)^{-1}\| \leq \frac{1}{\|A_\phi\|}$$

$$\implies \|(\bar{B}_\phi + \eta I_0)^{-1}\|\|A_\phi\| < 1$$

$$\implies \|x\|\|(\bar{B}_\phi + \eta I_0)^{-1}\|\|A_\phi\|\|x\| < \|x\|^2$$ \hspace{2cm} (A.46)

$$\implies \|x^\intercal(\bar{B}_\phi + \eta I_0)^{-1}A_\phi x\| < \|x\|^2$$

$$\implies x^\intercal(\bar{B}_\phi + \eta I_0)^{-1}A_\phi x < \|x\|^2$$

$$\implies x^\intercal(I - (\bar{B}_\phi + \eta I_0)^{-1}A_\phi)x > 0$$

Hence, if $\max(1, \max_i(\frac{1}{\lambda_i+\eta})) < \frac{1}{\|A_\phi\|}$, then $I - (\bar{B}_\phi + \eta I_0)^{-1}A_\phi$ is positive definite matrix. Therefore, the ODE $\dot{\bar{z}}(t) = g_\infty(\bar{z}(t))$ has origin as its unique globally asymptotically stable. Condition A5 is satisfied.

Let us the consider the ODE $\dot{z}(t) = h(z(t), \bar{z})$. Here, $h(z(t), \bar{z}) = \bar{R}_\phi^\pi + \bar{A}_\phi\bar{z} - (\bar{B}_\phi + \eta I_0)z_t$ As earlier, for $\eta + \lambda_{min}(C_\phi) > 0$, $\bar{B}_\phi + \eta I_0$ is positive definite matrix. Therefore, the ODE $\dot{z}(t) := h(z(t), \bar{z})$ has a unique globally asymptotically stable equilibrium point $\lambda(\bar{z}) = (B_\phi + \eta I_0)^{-1}(R_\phi^\pi + A_\phi\bar{z}_t)$.

Conditions A1 to A5 are satisfied, therefore $\sup_t \|z_t\| < \infty$, which implies iterates are bounded. Hence critic parameter $w_t$ is bounded.

# B ALGORITHM AND HYPERPARAMETERS

## B.1 (OFF-POLICY) ARO-DDPG PRACTICAL ALGORITHM

---

**Algorithm 1** (Off-Policy) ARO-DDPG Practical Algorithm

---

Initialize actor parameter $\theta$ and critic parameters $w_1, w_2$. Initialize actor target parameter $\theta \to \overline{\theta}$
Initialize critic target parameters $w_1 \to \overline{w_1}, w_2 \to \overline{w_2}$. Initialize average reward parameter $\rho$.
Initialize target average reward parameter $\rho \to \overline{\rho}$. Initialize Replay buffer = { }

1: $t = 0$, $s_0$ = env.reset()
2: **while** $t \leq$ total steps **do**
3:     $a_t = \pi(s_t) + \epsilon$ {$\epsilon$ denotes the noise}
4:     $s_{t+1} \sim P(\cdot|s_t, a_t)$ and $r_t = R(s_t, a_t)$
5:     Store $\{s_t, a_t, s_{t+1}\}$ in the Replay Buffer
6:     **if** $t \% eval\_freq == 0$ **then**
7:         Evaluate(agent)
8:     **end if**
9:     **if** $t \% critic\_update\_freq == 0$ **then**
10:        Update critic according to (24) - (27)
11:     **end if**
12:     **if** $t \% actor\_update\_freq == 0$ **then**
13:        Update actor according to (28) - (29)
14:        Update target estimators according to (30) - (32)
15:     **end if**
16:     **if** $s_{t+1}$ is terminal **then**
17:        $s_t =$ env.reset()
18:     **else**
19:        $s_t = s_{t+1}$
20:     **end if**
21: **end while**

---

## B.2 FINITE TIME ANALYSIS ALGORITHM

Here we present the algorithm with linear function approximator for which finite time analysis was done. $\mathcal{B}_t$ denotes the batch of tuple of the form $\{s_i, a_i, s_i'\}$ sampled from the buffer at timestep $t$. $\Gamma_{C_w}$ is a projection operator defined as $\Gamma_{C_w} : \mathbb{R}^k \to B$, where $B(\subset \mathbb{R}^k)$ is a compact convex set. Here, the critic parameter $w \in \mathbb{R}^k$.

---

**Algorithm 2** On-policy AR-DPG with Linear FA

---

Initialize actor parameter $\theta$ and critic parameters $w$. Initialize actor target parameter $\theta \to \overline{\theta}$.
Initialize critic target parameters $w \to \overline{w}$.Initialize average reward parameter $\rho$
Initialize target average reward parameter $\rho \to \overline{\rho}$
Initialize buffer = { }

1: $t = 0$, $s_0$ = env.reset()
2: **while** $t \leq$ total steps **do**
3:     $a_t = \pi(s_t) + \epsilon$ {$\epsilon$ is the noise}
4:     $s_{t+1} \sim P(\cdot|s_t, a_t)$ and $r_t = R(s_t, a_t)$
5:     Store $\{s_t, a_t, s_{t+1}\}$ in the Buffer
6:     **if** $t \, \% \, critic\_update\_freq == 0$ **then**
7:         Sample $\mathcal{B}_t = \{s_i, a_i, s_i'\}_{i=0}^{M-1}$ from the Replay Buffer
8:         $w_{t+1} = \Gamma_{C_w}\Big(w_t + \frac{\alpha_t}{M}\sum_{i=0}^{M-1}\Big(R^\pi(s_i) - \bar{\rho}_t + \phi^\pi(s_i')^\intercal\bar{w}_t - \phi^\pi(s_i)^\intercal w_t\Big)\phi^\pi(s_i) - \alpha_t\eta w_t\Big)$

9:         $\rho_{t+1} = \rho_t + \frac{\alpha_t}{M}\sum_{i=0}^{M-1}\Big(R^\pi(s_i) - \rho_t + \phi^\pi(s_i')^\intercal\bar{w}_t - \phi^\pi(s_i)^\intercal\bar{w}_t\Big)$
10:        $\overline{w}_{t+1} = \overline{w}_t + \beta_t(w_{t+1} - \overline{w}_{t+1})$
11:        $\overline{\rho}_{t+1} = \overline{\rho}_t + \beta_t(\rho_{t+1} - \overline{\rho}_{t+1})$
12:        $\theta_{t+1} = \theta_t + \frac{\gamma_t}{M}\sum_{i=0}^{M-1}\nabla_a Q_{diff}^w(s_i, a)|_{a=\pi(s_i)}\nabla_\theta\pi(s_i)$
13:        buffer = { }
14:     **end if**
15:     **if** $s_{t+1}$ is terminal **then**
16:        $s_t$ = env.reset()
17:     **else**
18:        $s_t = s_{t+1}$
19:     **end if**
20: **end while**

---

**Algorithm 3** Off-policy AR-DPG with Linear FA

---

Initialize actor parameter $\theta$ and critic parameters $w$
Initialize actor target parameter $\theta \to \overline{\theta}$ and
Initialize critic target parameters $w \to \overline{w}$
Initialize average reward parameter $\rho$ and
Initialize target average reward parameter $\rho \to \overline{\rho}$
$\mu$ is the behavior policy
Initialize Replay buffer = { }

1: $t = 0$, $s_0$ = env.reset()
2: **while** $t \leq$ total steps **do**
3:     $a_t = \mu(s_t) + \epsilon$ {$\epsilon$ is the noise}
4:     $s_{t+1} \sim P(\cdot|s_t, a_t)$ and $r_t = R(s_t, a_t)$
5:     Store $\{s_t, a_t, s_{t+1}\}$ in the Replay Buffer
6:     Sample $\mathbb{B}_t = \{s_i, a_i, s_i'\}_{i=0}^{M-1}$ from the Replay Buffer
7:     $w_{t+1} = \Gamma_{C_w}\Big(w_t + \frac{\alpha_t}{M}\sum_{i=0}^{M-1}\Big(R^\mu(s_i) - \bar{\rho}_t + \phi^\pi(s_i')^\intercal\bar{w}_t - \phi^\pi(s_i)^\intercal w_t\Big)\phi^\pi(s_i) - \alpha_t\eta w_t\Big)$
8:     $\rho_{t+1} = \rho_t + \frac{\alpha_t}{M}\sum_{i=0}^{M-1}\Big(R^\mu(s_i) - \rho_t + \phi^\pi(s_i')^\intercal\bar{w}_t - \phi^\pi(s_i)^\intercal\bar{w}_t\Big)$
9:     $\overline{w}_{t+1} = \overline{w}_t + \beta_t(w_{t+1} - \overline{w}_{t+1})$
10:     $\overline{\rho}_{t+1} = \overline{\rho}_t + \beta_t(\rho_{t+1} - \overline{\rho}_{t+1})$
11:     $\theta_{t+1} = \theta_t + \frac{\gamma_t}{M}\sum_{i=0}^{M-1}\nabla_a Q_{diff}^w(s_i, a)|_{a=\pi(s_i)}\nabla_\theta\pi(s_i)$
12:     **if** $s_{t+1}$ is terminal **then**
13:        $s_t$ = env.reset()
14:     **else**
15:        $s_t = s_{t+1}$
16:     **end if**
17: **end while**

---

## B.3 HYPERPARAMETERS

| Hyperparameter | Value |
|---|---|
| Buffer Size | 1e6 |
| Total Environment Steps | 1e6 |
| Batch size | 256 |
| Evaluation Frequency | 5000 |
| Training Episode Length | 1000 |
| Evaluation Episode Length | 10000 |
| Activation Function | ReLU |
| Learning rate Actor | 3e-4 |
| Learning rate Critic | 3e-4 |
| Learning rate Average reward parameter | 3e-4 |
| No. of Hidden Layers | 2 |
| No. of Nodes in Hidden Layer | 128 |
| Update frequency | 10 steps |
| No. of Critic updates | 10 |
| No. of Actor updates | 5 |
| Polyak averaging constant | 0.995 |

