# OpenReview forum: "Off Policy Average Reward Actor Critic with Deterministic Policy Search"
_ICLR.cc/2023/Conference — Submitted to ICLR 2023_

### Official Review · Reviewer_xGda · 2022-10-17

**Confidence:** 5
**Clarity, Quality, Novelty And Reproducibility:** See above
**Correctness:** 1
**Technical Novelty And Significance:** 1
**Empirical Novelty And Significance:** Not applicable
**Recommendation:** 1

**Strength And Weaknesses:**

Strength:
The idea is straightforward and seems doable

Weaknesses:
The presentation is very confusing so I cannot evaluate the correctness of the proofs at all.
For example, in Algorithm 2, the critic is a state value function. However, in the update of the actor, it uses the **ground truth** action value function. How can we get the ground truth action value function? Moreover, in line 8 in Algorithm 2, there is projection in updating the critic. In the analysis of the critic, the first equation in page 19, the projection, however, disappears.

Overall, this work combines several existing ideas and I generally believe such a combination should work. But there are many details that need to be taken care of, which the authors fail.

**Summary Of The Paper:**

The paper extends the canonical deterministic policy gradient from the discounted setting to the average reward setting. Finite sample analysis is provided.

**Summary Of The Review:**

See above

---

> ### Author Response · Authors · 2022-11-18
> **Response to Reviewer xGda**
>
> Thank you for the review.
> 1. In Algorithm 2 the update rule for actor uses approximate q-value function. There was a mistake from our side in the code for Algorithm 2, line 12. We have corrected the mistake.
> 2. The boundedness of critic parameter $w_t$ can be ensured using the result on stability of iterates of the algorithm, given by Lakshminarayanan et al. (2017). We have added another section in the appendix, A.2, to prove the boundedness of the critic parameter using results of Lakshminarayanan et al. (2017), without using the projection operator. The projection operator is no longer required because the critic parameter is bounded by itself. Hence even without using the projection operator the finite time analysis will go through.
> 3. We have tried to improve the presentation of the paper by following the advices given by the reviewers. We have properly defined the constants used in the lemmas and theorem. We have also included explanation for assumptions taken and improved the explanation of the main results of the paper. We have modified suitably the notations used in the paper to avoid any confusion. We have tried our best to remove the typographical errors present in the paper.
>
> Chandrashekar Lakshminarayanan and Shalabh Bhatnagar. A stability criterion for two timescale stochastic approximation schemes. Automatica, 79:108–114, 2017.

---

> ### Author Response · Authors · 2022-11-30
> **Response to Reviewer xGda (Part 2)**
>
> We have also realised now that the finite time analysis is correct even with the projection operator in place (as was the case in the version of the paper we submitted initially) because since projection is a non-expansive operator, projection onto a convex set will not increase the norm. In fact, since $w_{t+1}^{\*}$ lies within the convex constraint set, the projection will only reduce the distance or keep it same (but will not increase it). We need to insert one extra step on page 21 to highlight the correctness of our claim.
> \begin{equation}
> \begin{split}
> &|| w_{t+1} - w_{t+1}^{\*}||^2 \\\\
> &= ||\Gamma_{C_{w}}(w_t + \alpha_t f(B_t, w_t, \theta_t))  - w_{t+1}^{\*}||^2 \\\\
> &\leq ||(w_t + \alpha_t f(B_t, w_t, \theta_t)  - w_{t+1}^{\*}||^2 \\\\
> &= ||(w_t - w_t^{\*}) + (w_t^{\*} - w_{t+1}^{\*}) + \alpha_t f(B_t, w_t, \theta_t)||^2
> \end{split}
> \end{equation}
> The inequality above has also been claimed in [1]-[2], with similar arguments as above.
>
> [1] S. Qiu, Z. Yang, J. Ye and Z. Wang, "On Finite-Time Convergence of Actor-Critic Algorithm," in IEEE Journal on Selected Areas in Information Theory, vol. 2, no. 2, pp. 652-664, June 2021, doi: 10.1109/JSAIT.2021.3078754.
>
> [2] https://arxiv.org/abs/2005.01350

---

### Official Review · Reviewer_eTNK · 2022-10-26

**Confidence:** 3
**Correctness:** 3
**Technical Novelty And Significance:** 3
**Empirical Novelty And Significance:** 2
**Recommendation:** 6

**Clarity, Quality, Novelty And Reproducibility:**

**Clarity:**
The paper is fairly clearly written and not difficult to understand.

**Quality:**
I have a few concerns about quality, mostly regarding the experiments.

**Novelty:**
To the best of my knowledge the proposed method is novel, but I am not extremely familiar with the average-reward literature.

**Reproducibility:**
Enough details are given to reproduce the experiments.

**Strength And Weaknesses:**

**Strengths:**
+ The proposed approximate off-policy policy gradient for continuous states and deterministic policies is an intriguing solution to a difficult problem.
+ The paper is written fairly clearly.
+ The proposed algorithm performs well in the experiments.

**Potential Weaknesses/Questions/Things to improve:**
- Spelling mistake in the title of the paper ("DETERMINSITIC").
- "In the case of recurrent Markov Decision Processes (MDPs), average reward happens to be the most selective optimization criterion." What does "most selective" mean in this sentence?
- "Moreover, an obvious discrepancy between the objective function and the evaluation metric, that exists for discounted reward setting, is resolved by opting for the average reward criterion." What discrepancy is the paper referring to?
- It's generally good to comment on the assumptions made just after making them. Are they necessary or just for convenience? If necessary, why? For example, Assumption 3 can always be made true by normalizing the feature vectors, and is therefore just for convenience, whereas Assumption 1 seems necessary for the steady-state visit distribution to be well-defined.
- "$P_t^\pi$ is the state distribution at instant t given by (9)." This should say Equation 10, right?
- Is Equation 9 correct? I've only seen the discounted state visit distribution defined as $\frac{1}{\sum_{t=0}^\infty \gamma^t} \sum_{t=0}^\infty \gamma^t P^\pi_t(s) = (1-\gamma) \sum_{t=0}^\infty \gamma^t P^\pi_t(s)$, because the denominator is a geometric series that converges to $\frac{1}{1-\gamma}$. Where did the $\frac{1}{\gamma}$ term come from?
- It would be good to comment further on the extra error term $\mathcal{O}(N_\theta^2)$ due to off-policy sampling. What situations could cause it to be small or large? Does it increase over time as the learned target policy deviates more and more from the behaviour policy?
- What are the shaded regions in the plots? Confidence intervals?
- The plots do not use colourblind-friendly colours. Consider removing the legend and instead using labels and arrows to show which line corresponds to each algorithm. Doing so would greatly reduce the reliance on colour to determine the performance of each algorithm.
- The paper only reports experiment hyper-parameters, but does not describe how they were chosen. Which values were checked for each of the methods? Do all of the methods use the same hyper-parameter settings (this is not good)? Are the chosen hyper-parameters representative of algorithm performance? Without conducting a grid search for each method, it's difficult to draw conclusions about the relative performance of each algorithm, and the experiments are more like demonstrations than rigorous comparisons.

**Minor comments/suggestions:**
- There seems to be a lot of vertical space near the equations on pages 2 and 3. Is "\allowdisplaybreaks[4]" in the preamble of the LaTeX file?
- It would be clearer to move equations 24-32 inside of Algorithm 1 instead of referring to them, and would save a lot of space.
- It would be good to clarify that the definition of the policy in Section 2 is for deterministic policies.
- It looks like $\rho^\pi(s)$ is used for discounted state visit distribution and $\rho(\pi)$ is used for average reward. Using $\rho$ for both makes the notation more confusing for the reader than it needs to be.
- The policy gradient theorem from Degris et al. (2012) is only valid for the tabular case. When using function approximation, the true off-policy policy gradient is given by Imani et al. (2018). However, the emphatic weightings rely on importance sampling, which as stated in Section 3.4 will not work with deterministic policies in continuous state spaces. The proposed approximate off-policy policy gradient expression is an interesting alternative.

**References:**
- Imani, E., Graves, E., & White, M. (2018). An off-policy policy gradient theorem using emphatic weightings. Advances in Neural Information Processing Systems, 31.

**Summary Of The Paper:**

The paper presents deterministic policy gradient theorems (both on- and off-policy) for the average reward setting. The paper then presents an Average Reward actor-critic algorithm based on DDPG (ARO-DDPG). A finite-sample analysis is conducted, and the algorithm is demonstrated empirically on several environments.

**Summary Of The Review:**

Despite liking the proposed algorithm, I must recommend rejection due to the concerns listed above, the experiments in particular. If the authors can address my concerns satisfactorily, I would be willing to recommend acceptance.

**Update:** The authors have addressed some—but not all—of my concerns, and I am increasing my score and recommendation in response.

---

> ### Author Response · Authors · 2022-11-10
> **Response to Reviewer eTNK**
>
> Thank you for the valuable comments. We agree we should have provided information about the hyperparameter search. We have tried to address the concern regarding method of hyperparameter tuning. Your comment regarding vertical spacing helped us save a lot of space throughout the paper.
> 1. Potential Weakness:
>     1. We have corrected the spelling mistake in the title of the paper.
>     2. Selectiveness pertains to the n-discount optimality criteria. In a unichain MDP average reward optimality (gain optimality) is underselective because it does not consider reward obtained in transient states but only the rewards in recurrent states. In recurrent MDP, there are no transient states and hence average reward optimality becomes the most selective optimality criteria. More details can be found in Dewanto & Gallagher (2021) and Mahadevan (1996) for n-discount optimality criteria.
>     3. The discrepancy is the following. In discounted reward algorithm the objective function depends on discounted summation of reward but in evaluation we consider the sum of undiscounted reward. In average reward criteria, objective function includes average of rewards and evaluation also includes calculation of average reward. Please refer section 5.4 of Dewanto & Gallagher (2021).
>     4. We have added explanation to each assumption made in the paper.
>     5. We have corrected the reference number for $P^{\pi}_t$.
>     6. We have corrected equation 9.
>     7. The extra term is now denoted by $W_\theta$ instead of $N_\theta$. We have now included an explanation for the extra term in Theorem 4. Replay buffer contains samples from policies similar to current policy. Hence the extra term will be small in case of using replay buffer. This explains why update rule provide by Theorem 2 works with replay buffer. We performed experiment using samples from a different policy and the algorithm did not learn anything. This was because the magnitude of the extra term was very high.
>     8. Shaded region in the plot represents standard deviation.
>     9. We have now plotted the results using colorblind friendly color. We have removed legends and are using arrows and labels.
>    10. In the hyperparameter section we have mentioned general sets of parameters which achieves the best performance for most of the environment and close to the best performance on the rest. The optimal parameters may differ a little bit for few environments from what is mentioned in the hyperparameter section.
> We tried around 60 set of parameters for all environments. As we gained confidence on certain parameter value, we then performed modifications on top of it.
> We could have done a grid search. But it is computationally expensive to perform grid search with several value of seeds. Each experiment takes 6-7 hours of training on GPU.
>
> 2. Minor comments:
>     1. We have removed the vertical space on page 2 and page 3. The vertical space was present because of empty lines in latex script.
>     2. We have move Algorithm 1 to appendix and equation 24-32 are in the main text.
>     3. We have made the clarification in section 2 about deterministic policy.
>     4. We have replaced $\rho^{\pi}$ with $\omega^{\pi}$.
>
> Vektor Dewanto and Marcus Gallagher. Examining average and discounted reward optimality criteria in reinforcement learning. CoRR, abs/2107.01348, 2021. URL https://arxiv.org/abs/
> 2107.01348\
> \
> Mahadevan, S. Average reward reinforcement learning: Foundations, algorithms, and empirical results. Mach Learn 22, 159–195 (1996). https://doi.org/10.1007/BF00114727

---

### Official Review · Reviewer_nHrk · 2022-10-26

**Confidence:** 3
**Correctness:** 3
**Technical Novelty And Significance:** 3
**Empirical Novelty And Significance:** 3
**Recommendation:** 6

**Clarity, Quality, Novelty And Reproducibility:**

The theory in the paper is clear, novel, and of good quality, but the paper needs major rewriting in order to properly present and explain the results to the reader. The experiments do not perform a hyperparameter search on the baseline, so it is difficult to assess their fairness. The authors include the hyperparameters used.

**Strength And Weaknesses:**


Strengths:

- The average reward problem is pretty unexplored, and to my knowledge this is the first work properly formalizing this setting for deterministic policies
- The theoretical results mostly follow in a straightforward way from previous work, and most of them are exactly how one would expect them. Nevertheless, it is important to have a good reference for such results, like it is in this paper.
- All the assumptions are well stated before each main result


Weaknesses:

- (major) The paper would benefit from some rewriting. The theory sometimes lacks a proper explanation. For instance, the results of Theorem 3 and 4 are not even commented on. Why should we care about these results? Are they surprising? For a theoretical paper like this, it is important to convey the result to the reader properly.
- Parts of the paper could be better organized. Sometimes definitions appear a bit after quantities are used for the first time (See Lemma 1 and then Eq. 5). The long list of equations (24)->(32) should be rewritten in a more organized way.

- Target estimators in Eq. 24 appear without any justification

- In the experimental results the authors reimplemented the baselines and used the original hyperparameters reported in the baseline papers. This seems pretty unfair, as it is well-known that slightly different implementations can have very different experimental results. Since the authors tuned their hyperparameters, but not the baselines, how can we consider the experiments fair?

Minor:

- There are too many repetitions of "came up with" throughout the paper
- In Eq 12 should there be $d^{\mu}$ instead of $\rho^{\pi}$?

Other questions:

- Some proofs in the Appendix use a projector operator. Is the operator used in the implementation?
- Could the authors clarify why Eq. 21 is not a good off-policy objective? It was not clear to me from the explanation given in the paper.
- Assumption 2 is stated without comments. Could the authors explain it in the paper?
- What happens if the training phase is 1000 and the evaluation phase is 1000? How does the proposed method compare to the baseline in this case?

**Summary Of The Paper:**

The paper introduces an off-policy deterministic policy gradient for the average reward setting. The authors follow a standard approach, first deriving the policy gradient, then finding a class of compatible value function approximators, and finally providing a finite time analysis for the on-policy and off-policy cases. Alternative approaches for deriving the off-policy policy gradient are proposed, with an explanation of why they could be less appealing to follow. Finally, an experimental comparison with existing on-policy actor-critic algorithms based on the average reward shows that the proposed method performs well. This confirms that off-policy algorithms might be more sample efficient than on-policy algorithms also in the average reward setting.

**Summary Of The Review:**

The paper introduces an average reward off-policy deterministic policy gradient algorithm. While the theory is sound and significant, the authors should improve the quality of the writing in order to explain better the importance of their results. Some concerns about the experiments need to be addressed before acceptance.

-----------------------------------
Edit: Updating my score after author's response

---

> ### Author Response · Authors · 2022-11-10
> **Response to Reviewer nHrk**
>
> Thank you for reviewing the paper. We agree there are issues in the presentation of the paper. We have tried to incorporate the suggestions regarding presentation and address concerns regarding hyperparameter tuning of the baselines.
> 1. Major
>     1. We have modified our paper to expound upon the significance of the results mentioned in Theorem 3 and Theorem 4.
>     2. We have modified the order in which Lemma 1 and equation 5 appears. We have included an assumption on step sizes (Assumption 3). Further we have provided explanation for the equations 24 to 32.
>     3. In section 3.5 we have now included the motivation for using target estimators.
>     4. We performed hyperparameter tuning for ATRPO and found that hyperparameters mentioned in the original paper are already optimal. Hence for ATRPO we have written that we are using hyperparameters from the original paper.
> For APO we performed the hyperparameter tuning and still found that our algorithm was better. We performed several experiments following the suggestions provided by the original paper.
>
> 2. Minor
>     1. We have removed the repetitions from the paper.
>     2. We think it should be $\rho^{\pi}$ because equation 12 mentions the update rule for discounted reward setting and it uses long term discounted state visitation probability density ($\rho^{\pi}$) instead of state-state probability distribution $d^{\pi}$. Note: We are now using $\omega^{\pi}$ in place of $\rho^{\pi}$ to denote long term discounted state visitation probability density.
>
> 3. Other questions:
>     1. Algorithm 2 and Algorithm 3 used projection operator. We have used Algorithm 1 for the practical implementation, and it does not use projection operator.
>     2. In equation 21, we are also considering derivative of reward function. Therefore, agent updated using equation 21, will only try to increase reward in each state but won’t try to change the steady state distribution much. In equation 18 we are considering derivative of differential Q-value function. If we notice the definition of differential Q-value function in equation 5, it includes long term summation of rewards. Hence using a derivative of differential Q-value function will increase the reward and at the same time modify the steady state distribution so that agent transition to states where it can get high rewards. Therefore, it is written in the manuscript that differential Q-value function encapsulates information regarding both reward and transition dynamics.
>    3. We have now provided more explanation to Assumption 2.
>    4. Episode length of 10000 is worst case scenario as compared to episode length of 1000. We performed the experiment with evaluation episode length 1000 and noticed our algorithm still performed better than the baselines and the variance in the plots reduced.

---

> > ### Comment · Reviewer_nHrk · 2022-11-16
> > **The authors addessed my concerns**
> >
> > I appreciate that the authors addressed my concerns, especially those related to the explanation of the results in the paper. I will update my score for the submission.

---

### Official Review · Reviewer_Fu5H · 2022-10-31

**Confidence:** 4
**Correctness:** 3
**Technical Novelty And Significance:** 3
**Empirical Novelty And Significance:** Not applicable
**Recommendation:** 5

**Clarity, Quality, Novelty And Reproducibility:**

Clearly written, method introduced is novel, some additional details needed for experiments.

**Strength And Weaknesses:**

Average-reward RL has only started gaining attention in the last 2-3 years and this work fills an important gap in that it provides a practical off-policy deep reinforcement learning algorithm that shows potential in high dimensional continuing control tasks. My detailed comments are below:
- The methods introduced in the paper can be seen as an average-reward extension to [1] and [2]. Intuitively, especially for off-policy settings, the deterministic policy gradient approach is probably easier to use and analyze. I also liked Section 3.4 where the authors provided comparisons to alternatives.
- One comment/question I have is regarding the step size. One thing that has always been tricky with average reward algorithm is the choice of learning rate since we need both the Q function and the average reward rho converge, this essentially gives us a multi-time scale problem which are notoriously hard to tune and can have complicated requirements in terms of their step size choice (an example is the proof for differential Q learning found in [3]. I was also quite surprised that in the practical algorithm, the authors used the same step size for the actor, critic, and average reward and the resulting algorithm actually works quite well.
- For the experiments, it seems that in the original papers for the two baselines used in the paper (ATRPO, PPO), both papers used OpenAI gym instead of DM control for their experimentation work. So it is unclear whether the original implementation/settings for the baselines would work just as well on DM control. I wonder if the authors accounted for this in their experiment work.
- Also concerning the choice of baselines, I wonder how your algorithm performs against discounted DDPG/SAC? That is, empirically, is there any evidence that the average reward algorithm outperforms the discounted version? Since it has been pretty well-known that off-policy algorithms generally perform better in terms of sample efficiency compared to on-policy algorithms, the results in the experiments are not surprising. I’m curious how your algorithm performs when compared to other off-policy methods.
- Another question I have concerning the experiments is the update frequency of the actor/critic/average reward value. You mentioned in the experiment section that you update the critic network with more frequency, however in the hyperparameter section the update frequency is listed as 10, is this the frequency for the actor or critic or the average reward? More importantly, could you comment on the different update frequency since this issue did not seem to come up in the theoretical analysis. Also related to my previous point concerning step sizes, some comments on the update frequency of the average reward would be much appreciated.

[1] David Silver, Guy Lever, Nicolas Heess, Thomas Degris, Daan Wierstra, and Martin A. Riedmiller. Deterministic Policy Gradient Algorithms. ICML 2014

[2] Thomas Degris, Martha White, and Richard S. Sutton. Linear Off-policy Actor-Critic. ICML 2012

[3] Yi Wan, Abhishek Naik, and Richard S Sutton. Learning and Planning in Average-Reward Markov Decision Processes. ICML 2021



**Summary Of The Paper:**

This paper extends the deterministic policy gradient algorithm to the average reward case and introduces an average reward version of DDPG. Finite-time analysis of linear function approximator version of the algorithm is also shown for both the on-policy and off-policy case. Empirically, the new algorithm is shown to outperform other on-policy average reward algorithms.

**Summary Of The Review:**

Overall, I think this paper does make an important contribution to the field but some important issues still need to be addressed, especially in the experiment section. As of now I am setting my score to borderline reject but I am open to changing my score upon a satisfactory response from the authors. I look forward to hearing back from the authors.

---

> ### Author Response · Authors · 2022-11-10
> **Response to Reviewer Fu5H**
>
> Thank you for the suggestions. We have tried to address your concerns regarding hyperparameter tuning of baselines.
> 1. Thank you for the comment.
> 2. The step sizes used are same for actor, critic and average reward estimator but the frequency of update is different. Average reward estimator and critic is updated with the same frequency. Actor is updated with half the frequency compared to critic. More detail is given in the $5^{th}$ point below.
> 3. We accounted for the difference between OpenAI gym and DM control. We tried tuning the parameters and using more number of nodes in the hidden layer of neural network. For APO we performed hyperparameter tuning and found the best parameters for the DM control tasks. Our algorithm still performed better than APO. For ATRPO we found that the parameters mentioned in the original paper were the best and we compared our algorithm with the default parameters of ATRPO.
> 4. We have not compared our algorithm with other discounted off-policy algorithms because the objective function for average reward itself is different from the objective function of discounted reward. Hence such comparison does not seem correct. Wan et al. (2021) also provides the same reasoning.\
> \
> We cannot assume that off-policy average reward algorithm would be obviously better than on-policy average reward algorithm. In average reward setting, we found that policy evaluation heavily depends on correct estimation of average reward estimator. There was every possibility that because of off-policy policy evaluation in average reward setting, the performance of off-policy algorithm could have been worse than on-policy algorithm. We had to use Adam optimizer for average reward estimator to make the algorithm work. Without this modification, our off-policy algorithm was crashing badly.\
> \
> Comparison with other off-policy methods: We could not compare our algorithm as our algorithm is the first off-policy average reward actor critic algorithm to the best of our knowledge.
>
> 5. Update frequency of 10 in the hyperparameter section means we initiate the update step of the algorithm after taking 10 environment steps. Each update step of the algorithm include critic, average reward estimator, actor, and target network update but with different frequency. “No of Critic Updates” in hyperparameter section means we update the critic network and average reward estimator 10 times in one update step of algorithm. “No of Actor Updates” in hyperparameter section means we update actor 5 times in one update step of the algorithm. So, update frequency of actor is half that of critic. Further we use polyak averaging to update target network. We use different update frequency and polyak averaging to enforce different timescale in the practical algorithm for actor, critic and average reward estimator, and target networks(estimator).
>
> Wan, Y., Naik, A., & Sutton, R. (2021). Average-Reward Learning and Planning with Options. Advances in Neural Information Processing Systems, 34, 22758-22769.

---

### Official Review · Reviewer_SgUA · 2022-11-03

**Confidence:** 4
**Clarity, Quality, Novelty And Reproducibility:** 1. The structure of the paper

(1) Th…
**Correctness:** 3
**Technical Novelty And Significance:** 2
**Empirical Novelty And Significance:** 3
**Recommendation:** 3

**Strength And Weaknesses:**

Strength: The paper makes a good progression from theory to practice. It has enough novel contributions and provides convincing experiment results.

Weakness: The paper is not clearly written and has a few mismatched mathematical notations. Furthermore, the theorems of non-asymptotic bounds have unstated assumptions and conditions. The proof is also hard to follow. Also, the bounds do not directly describe the properties of algorithms; the regret bound or the bound on the optimality of the policy would be better.

**Summary Of The Paper:**

The paper develops both on-policy and off-policy deterministic policy gradient theorems under the average reward setting; here, the off-policy theorem only gives an approximated form of the gradient but shows the error between the approximated gradient and the true gradient is bounded. Based on the theorems, the paper proposes an on-policy average reward policy gradient algorithm which defeats the other two average reward policy gradient algorithms in both learning speed and final performance. Meanwhile, the paper derives upper bounds for the smallest expected square norm of the gradients in linear on-policy and off-policy algorithms, respectively.

**Summary Of The Review:**

The paper may not be ready for publication and needs rewriting.

---

> ### Author Response · Authors · 2022-11-10
> **Response to Reviewer SgUA**
>
> Thank you for the review. We agree that the paper organisation and the notations were not refined. We have incorporated all the concerns and uploaded the new manuscript for your review. Following are our responses to your comments:
> 1. Structure of the Paper
>     1. We have modified last paragraph on page 1 to explain its relevance to our proposed algorithm
>     2. We have added description of the update rules.
>     3. There was a mistake in section 3.5. We are using off-policy policy gradient theorem for actor update. We have modified the title of algorithm 1 to reflect the same.
>     4. Algorithm 1 is three-timescale algorithm. Eq. 26-27 contains $\alpha_t$, eq. 29 contains $\gamma_t$, eq. 30-32 contains beta_t. Similarly, algorithm 2 and 3 contains 3 different step sizes $\alpha_t, \beta_t, \gamma_t$
>
> 2. Notations
>     1. We have corrected eq. 9 by replacing $\frac{1}{\gamma}$ with $1-\gamma$.
>     2. We have replaced equality with approximation in equation 11.
>     3. We have removed the approximation symbol in Theorem 2.
>     4. We have modified to notation to $\rho_{new}$ in equation 20 and 21.
>     5. We have corrected the equation 24 and 25.
>     6. We have modified assumption 5 to indicate that we are talking about estimated value of q-value function.
>     7. We have replaced N with M everywhere to denote the batch size.
>
> 3. Finite-time analysis
>     1. We have expounded upon the significance of bounds in Theorem 3 and Theorem 4 after stating theorems.
>     2. We are not assuming that differential q-value function for every policy can be linearly represented. In lemma 6, $w_{\theta}^{\*}$ represents the optimal critic parameter for a linear critic. We are not implying $Q_{diff}^{w_{\theta}^{\*}}$ = $Q_{diff}^{\pi^{\theta}}$.
>     3. We have now defined constants in all lemmas and theorems.
>     4. We are not sure which upper bound you are pointing to. Assuming it is the upper bound of Lemma 6 include $\eta$. $\eta$ is the regularization coefficient from Algorithm 2, line 8.
>         + $C_s$ has now been defined in Lemma 4.
>         + Because of the font size, $C_{\alpha}$ is appearing as $C_a$. $C_{\alpha}$ is now defined in Assumption 3.
>         + In Lemma 6, $\lambda_{min}$ is defined as minimum eigen value for all $\theta$. Hence $\lambda_{min}$ is a scalar which does not depend on $\theta$.
>         + In Lemma 13 we forgot to mention that $\lambda_{min}^{\epsilon}$ is minimum eigen value for all values of $\theta$. We have made the changes. Hence $\lambda_{min}^{\epsilon}$ is a scaler independent of $\theta$.
>      5. In equationA.10 $\eta w$ term is mentioned in the group 1 of terms(i.e first line). We have now made it part of group 2 of terms (line 2) in A.10 to clear any confusion. We have defined $g$ and $\bar{g}$ on page 19 and used them in equation A.11 on page 19.
>      6. In our paper, Q-value features don’t depend on policy. $\phi(s,a)$ is the feature vector for a state-action pair. We denote $\phi(s,\pi(s))$ as $\phi^{\pi}(s)$ because we are using deterministic policy. We have modified Assumption 4 to clarify this point. However, Konda et al. (2003) uses features dependent on policy.
>
>  Konda, V. R., & Tsitsiklis, J. N. (2003). Onactor-critic algorithms. SIAM journal on Control and   Optimization, 42(4), 1143-1166.

---

> > ### Comment · Reviewer_SgUA · 2022-11-12
> > **Questions on the theoretical result**
> >
> > Thanks for your clarification! But I still have a few questions about your results.
> >
> > 1. On page 7, after Theorem 3, you claimed that your algorithm converges to a stationary point as T goes to infinity which confuses me. How about that O(1) term? Will it diminish?
> >
> > 2. For your Lemma 6, I know the true gradient of the loss equals zero at the optimality. But I did not get why your semi-gradient can equal zero. Could you give a proof? This is where I thought you unstated an assumption.
> >
> > 3. The minimum eigenvalue cannot be treated as a constant. We do not know how small it can be for all policies. A lower bound I know is given in Merikoski and Virtanen (1997), which depends on the size of the matrix.
> >
> > Merikoski, J. K., & Virtanen, A. (1997). Bounds for eigenvalues using the trace and determinant. Linear algebra and its applications, 264, 101-108.

---

> > > ### Author Response · Authors · 2022-11-15
> > > **Response to Questions on the theoretical result**
> > >
> > > 1. The bound in Theorem 3 and Theorem 4 involves an $O(1)$ term, which will not go to zero as time increases. The $O(1)$ term is appearing because, using linear function approximator cannot learn true q-value function with zero error. This type of $O(1)$ term is present in other papers such as Xiong et al.(2022) (Theorem 1) and Wu et al. (2020) (Corollary 4.9). Xiong et al.(2022) also mentions in the paper that the $O(1)$ term will reduce upon using neural network instead of linear function approximator.
> > >
> > > 2. Let us consider the equation $E[(R^{\pi}(s)-\rho_t^{\*})\phi^{\pi}(s) + (\phi^{\pi}(s)(E[\phi^{\pi}(s')]-\phi^{\pi}(s))^{\intercal} - \eta I)w] = 0$. We can write the equation as $ b(\theta) + A(\theta)w = 0$, where definition of $ b(\theta)$ and $ A(\theta)$ is given in Lemma 6. The solution to the above equation, in terms of $w$, exists because  $ A(\theta)$ is negative definite matrix for all $\theta$ because of carefully setting $\eta$ (l2-regularization coefficient), which is proved in Lemma 11. Hence the solution of the above equation is the optimal critic parameter which we refer to as $w_t^{*}$ in Lemma 6. Therefore, the semi-gradient that you are referring is equal to zero. Further there are several papers in literature which equates the semi-gradient to zero for the optimal critic parameter. Please refer to section 3.4 of Tsitsiklis et al. (1999) and section 4.1 of Wu et al. (2020).
> > >
> > > 3. We apologize for the wrong usage of words in Lemma 13. By minimum eigenvalue for all $\theta$ we meant that there exist a uniform lower bound on the minimum eigenvalue of the matrix for all values of $\theta$. We are not claiming the uniform lower bound is an eigenvalue for some matrix. There are several references of uniform lower bound on parameterized matrix in the literature. Please refer to equation 1.12 second edition of Non-Linear Programming by Dimitri Bertsekas and Assumption 1 and 4 of Xiong et al. (2022).
> > >
> > >
> > >
> > > Xiong, H., Xu, T., Zhao, L., Liang, Y., & Zhang, W. (2022, February). Deterministic Policy Gradient: Convergence Analysis. In The 38th Conference on Uncertainty in Artificial Intelligence.
> > >
> > > Wu, Y. F., Zhang, W., Xu, P., & Gu, Q. (2020). A finite-time analysis of two time-scale actor-critic methods. Advances in Neural Information Processing Systems, 33, 17617-17628.
> > >
> > > Tsitsiklis, J. N., & Van Roy, B. (1999). Average cost temporal-difference learning. Automatica, 35(11), 1799-1808.

---

> > > > ### Comment · Reviewer_SgUA · 2022-11-18
> > > > **Further questions on the theoretical results**
> > > >
> > > > 1.  I am still confused about how the inclusion of a constant term in the gradient bound supports the claim of convergence. The paper makes an incorrect statement.
> > > >
> > > > 2. Thanks for your clarification! You are right. Now I understand this condition. But there are still two problems.
> > > >
> > > > The proof is not well written. This optimal critic parameter should be introduced before the use and this property should be pointed to when using.
> > > >
> > > > Also, here you set a condition for \eta. But it is not stated in any lemma or theorem. It is hidden in your proof.
> > > >
> > > > 3. Xiong et al. (2022) makes an assumption on the uniform lower bound. It is an assumption, not a theorem. If you make the same assumption, it should be stated. Otherwise, you need to show a bound.

---

> > > > > ### Author Response · Authors · 2022-11-18
> > > > > **Response to Further questions on the theoretical result**
> > > > >
> > > > >
> > > > > 1. We agree that the explanation given for Theorem 3 is bit inaccurate. The $O(1)$ error term will be present in the bound because of using linear function approximation. If the bounds on the gradient does not include $O(1)$ error term that would mean linear function approximator is sufficient to learn any Q-value function and by default satisfies compatible function approximator lemma (Lemma 2). If the $O(1)$ term is small, then the algorithm will get close to local maxima of the objective function as time increases. We have updated the explanation of Theorem 3 in the manuscript to clarify the role of $O(1)$ error term. We are not claiming the algorithm will find the exact local maxima.
> > > > >
> > > > > 2. We have updated the manuscript to include more explanation about optimal critic parameter. We have mentioned the condition for $\eta$ in all the lemma where we are using $\eta$.
> > > > >
> > > > > 3. The lower bound on minimum eigenvalue is an assumption. We have included three new assumptions 12, 13, and 14 to clearly state the assumptions taken.

---

### Decision · Program_Chairs · 2023-01-20

**Decision:**

Reject

**Justification For Why Not Higher Score:**

There are significant concerns about the theoretical results.

**Justification For Why Not Lower Score:**

N/A

**Metareview: Summary, Strengths And Weaknesses:**

This work addresses the important problem of developing an off-policy actor-critic method for the average reward setting. The work addresses the problem by developing on-policy and off-policy deterministic policy gradient theorems under the average reward setting and corresponding algorithms.

However, several reviewers pointed out that the exposition of the theoretical results requires significant improvement as many issues arose during the back-and-forth between the reviewers and the authors. The responses indicate that a thorough draft update, including the theoretical results and a fresh review of the theoretical results, is warranted. Specifically, the non-iid nature of data when sampling from a replay buffer, as well as the dropping of the projection, which was necessary for bounding the critic, should be addressed either to correct the result or to remove misunderstandings. We strongly encourage the authors to resubmit to the next venue after addressing these issues.